# Combining Recurrent, Convolutional, and Continuous-time Models with Linear State-Space Layers

**Albert Gu**[†], **Isys Johnson**[‡], **Karan Goel**[†], **Khaled Saab**[*], **Tri Dao**[†], **Atri Rudra**[‡], **Christopher Ré**[†]

[†] Department of Computer Science, Stanford University
[*] Department of Electrical Engineering, Stanford University
[‡] Department of Computer Science and Engineering, University at Buffalo, SUNY

`{albertgu,knrg,ksaab,trid}@stanford.edu, chrismre@cs.stanford.edu`
`{isysjohn,atri}@buffalo.edu`

## Abstract

Recurrent neural networks (RNNs), temporal convolutions, and neural differential equations (NDEs) are popular families of deep learning models for time-series data, each with unique strengths and tradeoffs in modeling power and computational efficiency. We introduce a simple sequence model inspired by control systems that generalizes these approaches while addressing their shortcomings. The Linear State-Space Layer (LSSL) maps a sequence $u \mapsto y$ by simply simulating a linear continuous-time state-space representation $\dot{x} = Ax + Bu, y = Cx + Du$. Theoretically, we show that LSSL models are closely related to the three aforementioned families of models and inherit their strengths. For example, they generalize convolutions to continuous-time, explain common RNN heuristics, and share features of NDEs such as time-scale adaptation. We then incorporate and generalize recent theory on continuous-time memorization to introduce a trainable subset of structured matrices $A$ that endow LSSLs with long-range memory. Empirically, stacking LSSL layers into a simple deep neural network obtains state-of-the-art results across time series benchmarks for long dependencies in sequential image classification, real-world healthcare regression tasks, and speech. On a difficult speech classification task with length-16000 sequences, LSSL outperforms prior approaches by 24 accuracy points, and even outperforms baselines that use hand-crafted features on 100x shorter sequences.

## 1 Introduction

A longstanding challenge in machine learning is efficiently modeling sequential data longer than a few thousand time steps. The usual paradigms for designing sequence models involve recurrence (e.g. RNNs), convolutions (e.g. CNNs), or differential equations (e.g. NDEs), which each come with tradeoffs. For example, RNNs are a natural stateful model for sequential data that require only constant computation/storage per time step, but are slow to train and suffer from optimization difficulties (e.g., the "vanishing gradient problem" [39]), which empirically limits their ability to handle long sequences. CNNs encode local context and enjoy fast, parallelizable training, but are not sequential, resulting in more expensive inference and an inherent limitation on the context length. NDEs are a principled mathematical model that can theoretically address continuous-time problems and long-term dependencies [37], but are very inefficient.

Ideally, a model family would combine the strengths of these paradigms, providing properties like parallelizable training (convolutional), stateful inference (recurrence) and time-scale adaptation

35th Conference on Neural Information Processing Systems (NeurIPS 2021).

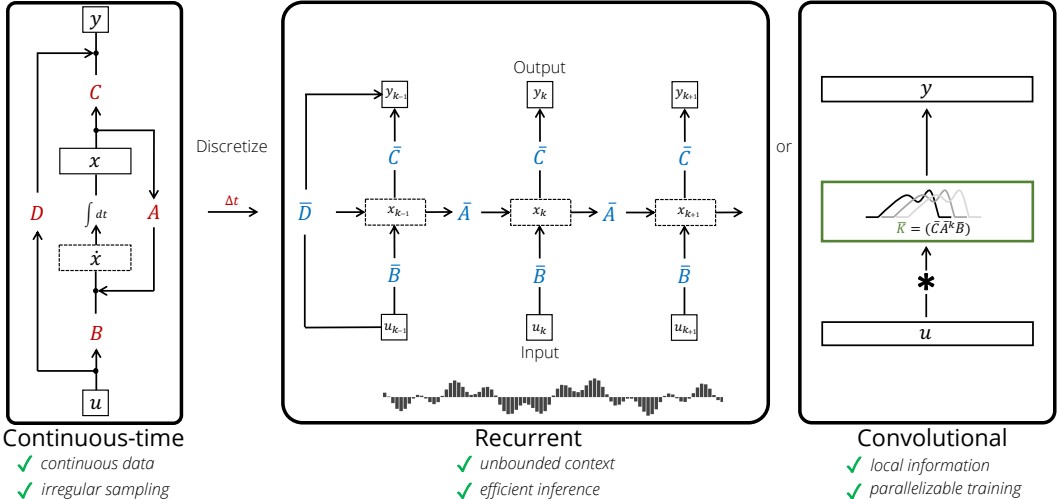

Figure 1: (**Three views of the LSSL**) A **Linear State Space Layer** layer is a map $u_t \in \mathbb{R} \to y_t \in \mathbb{R}$, where each feature $u_t \mapsto y_t$ is defined by discretizing a state-space model $A, B, C, D$ with a parameter $\Delta t$. The underlying state space model defines a discrete recurrence through combining the state matrix $A$ and timescale $\Delta t$ into a transition matrix $\overline{A}$. (**Left**) As an implicit continuous model, irregularly-spaced data can be handled by discretizing the same matrix $A$ using a different timescale $\Delta t$. (**Center**) As a recurrent model, inference can be performed efficiently by computing the layer *timewise* (i.e., one vertical slice at a time $(u_t, x_t, y_t), (u_{t+1}, x_{t+1}, y_{t+1}), \ldots$), by unrolling the linear recurrence. (**Right**) As a convolutional model, training can be performed efficiently by computing the layer *depthwise* in parallel (i.e., one horizontal slice at a time $(u_t)_{t \in [L]}, (y_t)_{t \in [L]}, \ldots$), by convolving with a particular filter.

(differential equations), while handling very long sequences in a computationally efficient way. Several recent works have turned to this question. These include the CKConv, which models a continuous convolution kernel [44]; several ODE-inspired RNNs, such as the UnICORNN [47]; the LMU, which speeds up a specific linear recurrence using convolutions [12, 58]; and HiPPO [24], a generalization of the LMU that introduces a theoretical framework for continuous-time memorization. However, these model families come at the price of reduced *expressivity*: intuitively, a family that is both convolutional and recurrent should be more restrictive than either.

Our first goal is to construct an expressive model family that combines all 3 paradigms while preserving their strengths. The **Linear State-Space Layer (LSSL)** is a simple sequence model that maps a 1-dimensional function or sequence $u(t) \mapsto y(t)$ through an implicit state $x(t)$ by simulating a linear continuous-time state-space representation in discrete-time

$$\dot{x}(t) = Ax(t) + Bu(t) \tag{1}$$
$$y(t) = Cx(t) + Du(t), \tag{2}$$

where $A$ controls the evolution of the system and $B, C, D$ are projection parameters. The LSSL can be viewed as an instantiation of each family, inheriting their strengths (Fig. 1):

- **LSSLs are recurrent.** If a discrete step-size $\Delta t$ is specified, the LSSL can be discretized into a linear recurrence using standard techniques, and simulated during inference as a stateful recurrent model with constant memory and computation per time step.

- **LSSLs are convolutional.** The linear time-invariant systems defined by (1)+(2) are known to be explicitly representable as a continuous convolution. Moreover, the discrete-time version can be parallelized during training using convolutions [12, 44].

- **LSSLs are continuous-time.** The LSSL itself is a differential equation. As such, it can perform unique applications of continuous-time models, such as simulating continuous processes, handling missing data [45], and adapting to different timescales.

Surprisingly, we show that LSSLs do not sacrifice expressivity, and in fact generalize convolutions and RNNs. First, classical results from control theory imply that all 1-D convolutional kernels can be approximated by an LSSL [59]. Additionally, we provide two results relating RNNs and ODEs

that may be of broader interest, e.g. showing that some RNN architectural heuristics (such as gating mechanisms) are related to the step-size $\Delta t$ and can actually be derived from ODE approximations. As corollaries of these results, we show that popular RNN methods are special cases of LSSLs.

The generality of LSSLs does come with tradeoffs. In particular, we describe and address two challenges that naive LSSL instantiations face when handling long sequences: (i) they inherit the limitations of both RNNs and CNNs at remembering long dependencies, and (ii) choosing the state matrix $A$ and timescale $\Delta t$ appropriately are critical to their performance, yet learning them is computationally infeasible. We simultaneously address these challenges by specializing LSSLs using a carefully chosen class of structured matrices $A$, such that (i) these matrices generalize prior work on continuous-time memory [24] and mathematically capture long dependencies with respect to a learnable family of measures, and (ii) with new algorithms, LSSLs with these matrices $A$ can be theoretically sped up under certain computation models, even while learning the measure $A$ and timescale $\Delta t$.

We empirically validate that LSSLs are widely effective on benchmark datasets and very long time series from healthcare sensor data, images, and speech.

- On benchmark datasets, LSSLs obtain SoTA over recent RNN, CNN, and NDE-based methods across sequential image classification tasks (e.g., by over 10% accuracy on sequential CIFAR) and healthcare regression tasks with length-4000 time series (by up to 80% reduction in RMSE).

- To showcase the potential of LSSLs to unlock applications with extremely long sequences, we introduce a new sequential CelebA classification task with length-38000 sequences. A small LSSL comes within 2.16 accuracy points of a specialized ResNet-18 vision architecture that has 10x more parameters and is trained directly on images.

- Finally, we test LSSLs on a difficult dataset of high-resolution speech clips, where usual speech pipelines pre-process the signals to reduce the length by 100x. When training on the *raw* length-16000 signals, the LSSL not only (i) outperforms previous methods by over 20 accuracy points in 1/5 the training time, but (ii) outperforms all baselines that use the pre-processed length-160 sequences, overcoming the limitations of hand-crafted feature engineering.

**Summary of Contributions**

- We introduce Linear State-Space Layers (LSSLs), a simple sequence-to-sequence transformation that shares the modeling advantages of recurrent, convolutional, and continuous-time methods. Conversely, we show that RNNs and CNNs can be seen as special cases of LSSLs (Section 3).

- We prove that a structured subclass of LSSLs can learn representations that solve continuous-time memorization, allowing it to adapt its measure and timescale (Section 4.1). We also provide new algorithms for these LSSLs, showing that they can be sped up computationally under an arithmetic complexity model Section 4.2.

- Empirically, we show that LSSLs stacked into a deep neural network are widely effective on time series data, even (or *especially*) on extremely long sequences (Section 5).

## 2  Technical Background

We summarize the preliminaries on differential equations that are necessary for this work. We first introduce two standard approximation schemes for differential equations that we will use to convert continuous-time models to discrete-time, and will be used in our results on understanding RNNs. We give further context on the step size or timescale $\Delta t$, which is a particularly important parameter involved in this approximation process. Finally, we provide a summary of the HiPPO framework for continuous-time memorization [24], which will give us a mathematical tool for constructing LSSLs that can address long-term dependencies.

**Approximations of differential equations.** Any differential equation $\dot{x}(t) = f(t, x(t))$ has an equivalent *integral equation* $x(t) = x(t_0) + \int_{t_0}^{t} f(s, x(s)) \, ds$. This can be numerically solved by storing some approximation for $x$, and keeping it fixed inside $f(t, x)$ while iterating the equation. For example, *Picard iteration* is often used to prove the existence of solutions to ODEs by iterating

the equation $x_{i+1}(t) := x_i(t_0) + \int_{t_0}^{t} f(s, x_i(s))\,ds$. In other words, it finds a sequence of functions $x_0(t), x_1(t), \ldots$ that approximate the solution $x(t)$ of the integral equation.

**Discretization.** On the other hand, for a desired sequence of discrete times $t_i$, approximations to $x(t_0), x(t_1), \ldots$ can be found by iterating the equation $x(t_{i+1}) = x(t_i) + \int_{t_i}^{t_{i+1}} f(s, x(s))\,ds$. Different ways of approximating the RHS integral lead to different discretization schemes. We single out a discretization method called the **generalized bilinear transform (GBT)** which is specialized to linear ODEs of the form (1). Given a *step size* $\Delta t$, the GBT update is

$$x(t + \Delta t) = (I - \alpha \Delta t \cdot A)^{-1}(I + (1 - \alpha)\Delta t \cdot A)x(t) + \Delta t(I - \alpha \Delta t \cdot A)^{-1}B \cdot u(t). \quad (3)$$

Three important cases are: $\alpha = 0$ becomes the classic *Euler method* which is simply the first-order approximation $x(t + \Delta t) = x(t) + \Delta t \cdot x'(t)$; $\alpha = 1$ is called the *backward Euler* method; and $\alpha = \frac{1}{2}$ is called the *bilinear* method, which preserves the stability of the system [61].

In Section 3.2 we will show that the backward Euler method and Picard iteration are actually related to RNNs. On the other hand, the bilinear discretization will be our main method for computing accurate discrete-time approximations of our continuous-time models. In particular, define $\overline{A}$ and $\overline{B}$ to be the matrices appearing in (3) for $\alpha = \frac{1}{2}$. Then the discrete-time state-space model is

$$x_t = \overline{A}x_{t-1} + \overline{B}u_t \quad (4)$$
$$y_t = Cx_t + Du_t. \quad (5)$$

$\Delta t$ **as a timescale.** In most models, the length of dependencies they can capture is roughly proportional to $\frac{1}{\Delta t}$. Thus we also refer to the step size $\Delta t$ as a *timescale*. This is an intrinsic part of converting a continuous-time ODE into a discrete-time recurrence, and most ODE-based RNN models have it as an important and non-trainable hyperparameter [24, 47, 58]. On the other hand, in Section 3.2 we show that the gating mechanism of classical RNNs is a version of learning $\Delta t$. Moreover when viewed as a CNN, the timescale $\Delta t$ can be viewed as controlling the width of the convolution kernel (Section 3.2). Ideally, all ODE-based sequence models would be able to automatically learn the proper timescales.

**Continuous-time memory.** Consider an input function $u(t)$, a fixed probability measure $\omega(t)$, and a sequence of $N$ basis functions such as polynomials. At every time $t$, the history of $u$ before time $t$ can be projected onto this basis, which yields a vector of coefficients $x(t) \in \mathbb{R}^N$ that represents an optimal approximation of the history of $u$ with respect to the provided measure $\omega$. The map taking the function $u(t) \in \mathbb{R}$ to coefficients $x(t) \in \mathbb{R}^N$ is called the **High-Order Polynomial Projection Operator (HiPPO)** with respect to the measure $\omega$. In special cases such as the uniform measure $\omega = \mathbb{I}\{[0, 1]\}$ and the exponentially-decaying measure $\omega(t) = \exp(-t)$, Gu et al. [24] showed that $x(t)$ satisfies a differential equation $\dot{x}(t) = A(t)x(t) + B(t)u(t)$ (i.e., (1)) and derived closed forms for the matrix $A$. Their framework provides a principled way to design memory models handling long dependencies; however, they prove only these few special cases.

## 3 Linear State-Space Layers (LSSL)

We define our main abstraction, a model family that generalizes recurrence and convolutions. Section 3.1 first formally defines the LSSL, then discusses how to compute it with multiple views. Conversely, Section 3.2 shows that LSSLs are related to mechanisms of the most popular RNNs.

### 3.1 Different Views of the LSSL

Given a fixed state space representation $A, B, C, D$, an LSSL is the sequence-to-sequence mapping defined by discretizing the linear state-space model (1) and (2).

Concretely, an LSSL layer has parameters $A, B, C, D$, and $\Delta t$. It operates on an input $u \in \mathbb{R}^{L \times H}$ representing a sequence of length $L$ where each timestep has an $H$-dimensional feature vector. Each feature $h \in [H]$ defines a sequence $(u_t^{(h)})_{t \in [L]}$, which is combined with a timescale $\Delta t_h$ to define an output $y^{(h)} \in \mathbb{R}^L$ via the discretized state-space model (4)+(5).

Computationally, the discrete-time LSSL can be viewed in multiple ways (Fig. 1).

**As a recurrence.** The recurrent state $x_{t-1} \in \mathbb{R}^{H \times N}$ carries the context of all inputs before time $t$. The current state $x_t$ and output $y_t$ can be computed by simply following equations (4)+(5). Thus the LSSL is a recurrent model with efficient and stateful inference, which can consume a (potentially unbounded) sequence of inputs while requiring fixed computation/storage per time step.

**As a convolution.** For simplicity let the initial state be $x_{-1} = 0$. Then (4)+(5) explicitly yields

$$y_k = C\left(\overline{A}\right)^k \overline{B}u_0 + C\left(\overline{A}\right)^{k-1}\overline{B}u_1 + \cdots + C\overline{AB}u_{k-1} + \overline{B}u_k + Du_k. \tag{6}$$

Then $y$ is simply the (non-circular) convolution $y = \mathcal{K}_L(\overline{A}, \overline{B}, C) * u + Du$, where

$$\mathcal{K}_L(A, B, C) = \left(CA^i B\right)_{i \in [L]} \in \mathbb{R}^L = (CB, CAB, \ldots, CA^{L-1}B). \tag{7}$$

Thus the LSSL can be viewed as a convolutional model where the entire output $y \in \mathbb{R}^{H \times L}$ can be computed at once by a convolution, which can be efficiently implemented with three FFTs.

**The computational bottleneck.** We make a note that the bottleneck of (i) the recurrence view is **matrix-vector multiplication (MVM)** by the discretized state matrix $\overline{A}$ when simulating (4), and (ii) the convolutional view is computing the **Krylov function** $\mathcal{K}_L$ (7). Throughout this section we assumed the LSSL parameters were fixed, which means that $\overline{A}$ and $\mathcal{K}_L(A, B, C)$ can be cached for efficiency. However, learning the parameters $\overline{A}$ and $\Delta t$ would involve repeatedly re-computing these, which is infeasible in practice. We revisit and solve this problem in Section 4.2.

## 3.2 Expressivity of LSSLs

For a model to be both recurrent and convolutional, one might expect it to be limited in other ways. Indeed, while [12, 44] also observe that certain recurrences can be replaced with a convolution, they note that it is not obvious if convolutions can be replaced by recurrences. Moreover, while the LSSL is a linear recurrence, popular RNN models are *nonlinear* sequence models with activation functions between each time step. We now show that LSSLs surprisingly do not have limited expressivity.

**Convolutions are LSSLs.** A well-known fact about state-space systems (1)+(2) is that the output $y$ is related to the input $u$ by a convolution $y(t) = \int h(\tau)u(t - \tau)d\tau$ with the *impulse response* $h$ of the system. Conversely, a convolutional filter $h$ that is a rational function of degree $N$ can be represented by a state-space model of size $N$ [59]. Thus, an arbitrary convolutional filter $h$ can be approximated by a rational function (e.g., by Padé approximants) and represented by an LSSL.

In the particular case of LSSLs with HiPPO matrices (Sections 2 and 4.1), there is another intuitive interpretation of how LSSL relate to convolutions. Consider the special case when $A$ corresponds to a uniform measure (in the literature known as the LMU [58] or HiPPO-LegT [24] matrix). Then for a fixed $dt$, equation (1) is simply memorizing the input within sliding windows of $\frac{1}{\Delta t}$ elements, and equation (2) extracts features from this window. Thus the LSSL can be interpreted as automatically learning convolution filters with a learnable kernel width.

**RNNs are LSSLs.** We show two results about RNNs that may be of broader interest. Our first result says that the ubiquitous *gating mechanism* of RNNs, commonly perceived as a heuristic to smooth optimization [28], is actually the analog of a step size or timescale $\Delta t$.

**Lemma 3.1.** *A (1-D) gated recurrence $x_t = (1 - \sigma(z))x_{t-1} + \sigma(z)u_t$, where $\sigma$ is the sigmoid function and $z$ is an arbitrary expression, can be viewed as the GBT($\alpha = 1$) (i.e.,* backwards-Euler*) discretization of a 1-D linear ODE $\dot{x}(t) = -x(t) + u(t)$.*

*Proof.* Applying a discretization requires a positive step size $\Delta t$. The simplest way to parameterize a positive function is via the exponential function $\Delta t = \exp(z)$ applied to any expression $z$. Substituting this into (3) with $A = -1, B = 1, \alpha = 1$ exactly produces the gated recurrence. □

While Lemma 3.1 involves approximating continuous systems using discretization, the second result is about approximating them using Picard iteration (Section 2). Roughly speaking, each layer of a deep *linear* RNN can be viewed as successive Picard iterates $x_0(t), x_1(t), \ldots$ approximating a function $x(t)$ defined by a *non-linear* ODE. This shows that we do not lose modeling power by using

linear instead of non-linear recurrences, and that the nonlinearity can instead be "moved" to the depth direction of deep neural networks to improve speed without sacrificing expressivity.

**Lemma 3.2.** *(Infinitely) deep stacked LSSL layers of order $N = 1$ with* position-wise *non-linear functions can approximate any non-linear ODE $\dot{x}(t) = -x + f(t, x(t))$.*

We note that many of the most popular and effective RNN variants such as the LSTM [28], GRU [14], QRNN [5], and SRU [33], involve a hidden state $x_t \in \mathbb{R}^H$ that involves independently "gating" the $H$ hidden units. Applying Lemma 3.1, they actually also approximate an ODE of the form in Lemma 3.2. Thus LSSLs and these popular RNN models can be seen to all approximate the same type of underlying continuous dynamics, by using Picard approximations in the depth direction and discretization (gates) in the time direction. Appendix C gives precise statements and proofs.

## 3.3 Deep LSSLs

The basic LSSL is defined as a sequence-to-sequence map from $\mathbb{R}^L \to \mathbb{R}^L$ on 1D sequences of length $L$, parameterized by parameters $A \in \mathbb{R}^{N \times N}, B \in \mathbb{R}^{N \times 1}, C \in \mathbb{R}^{1 \times N}, D \in \mathbb{R}^{1 \times 1}, \Delta t \in \mathbb{R}$. Given an input sequence with hidden dimension $H$ (in other words a feature dimension greater than 1), we simply broadcast the parameters $B, C, D, \Delta t$ with an extra dimension $H$. Each of these $H$ copies is learned independently, so that there are $H$ different versions of a 1D LSSL processing each of the input features independently. Overall, the standalone LSSL layer is a sequence-to-sequence map with the same interface as standard sequence model layers such as RNNs, CNNs, and Transformers.

The full LSSL architecture in a deep neural network is defined similarly to standard sequence models such as deep ResNets and Transformers, involving stacking LSSL layers connected with normalization layers and residual connections. Full architecture details are described in Appendix B, including the initialization of $A$ and $\Delta t$, computational details, and other architectural details.

# 4 Combining LSSLs with Continuous-time Memorization

In Section 3 we introduced the LSSL model and showed that it shares the strengths of convolutions and recurrences while also generalizing them. We now discuss and address its main limitations, in particular handling long dependencies (Section 4.1) and efficient computation (Section 4.2).

## 4.1 Incorporating Long Dependencies into LSSLs

The generality of LSSLs means they can inherit the issues of recurrences and convolutions at addressing long dependencies (Section 1). For example, viewed as a recurrence, repeated multiplication by $\overline{A}$ could suffer from the *vanishing gradients* problem [39, 44]. We confirm empirically that LSSLs with random state matrices $A$ are actually not effective (Section 5.4) as a generic sequence model.

However, one advantage of these mathematical continuous-time models is that they are theoretically analyzable, and specific $A$ matrices can be derived to address this issue. In particular, the HiPPO framework (Section 2) describes how to memorize a function in continuous time with respect to a measure $\omega$ [24]. This operator mapping a function to a continuous representation of its past is denoted hippo($\omega$), and was shown to have the form of equation (1) in three special cases. However, these matrices are non-trainable in the sense that no other $A$ matrices were known to be hippo operators.

To address this, we theoretically resolve the open question from [24], showing that hippo($\omega$) for *any measure $\omega$* [1] results in (1) with a structured matrix $A$.

**Theorem 1** (Informal). *For an arbitrary measure $\omega$, the optimal memorization operator* hippo($\omega$) *has the form $\dot{x}(t) = Ax(t) + Bu(t)$ (1) for a* low recurrence-width (LRW) *[17] state matrix $A$.*

For measures covering the classical orthogonal polynomials (OPs) [52] (in particular, corresponding to Jacobi and Laguerre polynomials), there is even more structure.

**Corollary 4.1.** *For $\omega$ corresponding to the classical OPs,* hippo($\omega$) *is 3-quasiseparable.*

Although beyond the scope of this section, we mention that LRW matrices are a type of structured matrix that have linear MVM [17]. In Appendix D we define this class and prove Theorem 1. Quasi-

---

[1]To be precise, the measures that correspond to orthogonal polynomials [52].

separable matrices are a related class of structured matrices with additional algorithmic properties. We define these matrices in Definition 4 and prove Corollary 4.1 in Appendix D.3.

Theorem 1 tells us that a LSSL that uses a state matrix $A$ within a particular class of structured matrices would carry the theoretical interpretation of continuous-time memorization. Ideally, we would be able to automatically learn the best $A$ within this class; however, this runs into computational challenges which we address next (Section 4.2). For now, we define the **LSSL-fixed** or **LSSL-f** to be one where the $A$ matrix is fixed to one of the HiPPO matrices prescribed by [24].

## 4.2 Theoretically Efficient Algorithms for the LSSL

Although $A$ and $\Delta t$ are the most critical parameters of an LSSL which govern the state-space (c.f. Section 4.1) and timescale (Sections 2 and 3.2), they are not feasible to train in a naive LSSL. In particular, Section 3.1 noted that it would require efficient *matrix-vector multiplication (MVM)* and *Krylov function* (7) for $\overline{A}$ to compute the recurrent and convolutional views, respectively. However, the former seems to involve a matrix inversion (3), while the latter seems to require powering $\overline{A}$ up $L$ times.

In this section, we show that the same restriction of $A$ to the class of quasiseparable (Corollary 4.1), which gives an LSSL the ability to theoretically remember long dependencies, simultaneously grants it computational efficiency.

First of all, it is known that quasiseparable matrices have efficient (linear-time) MVM [40]. We show that they also have fast Krylov functions, allowing efficient training with convolutions.

**Theorem 2.** *For any $k$-quasiseparable matrix $A$ (with constant $k$) and arbitrary $B, C$, the Krylov function $\mathcal{K}_L(A, B, C)$ can be computed in* quasi-linear *time and space* $\tilde{O}(N + L)$ *and* logarithmic *depth (i.e., is parallelizable). The operation count is in an exact arithmetic model, not accounting for bit complexity or numerical stability.*

We remark that Theorem 2 is non-obvious. To illustrate, it is easy to see that unrolling (7) for a general matrix $A$ takes time $LN^2$. Even if $A$ is extremely structured with linear computation, it requires $LN$ operations and linear depth. The depth can be reduced with the squaring technique (batch multiply by $A, A^2, A^4, \ldots$), but this then requires $LN$ intermediate storage. In fact, the algorithm for Theorem 2 is quite sophisticated (Appendix E) and involves a divide-and-conquer recursion over matrices of *polynomials*, using the observation that (7) is related to the power series $C(I - Ax)^{-1}B$.

Unless specified otherwise, the full **LSSL** refers to an LSSL with $A$ satisfying Corollary 4.1. In conclusion, learning within this structured matrix family simultaneously endows LSSLs with long-range memory through Theorem 1 and is theoretically computationally feasible through Theorem 2. We note the caveat that Theorem 2 is over exact arithmetic and not floating point numbers, and thus is treated more as a proof of concept that LSSLs can be computationally efficient in theory. We comment more on the limitations of the LSSL in Section 6.

# 5 Empirical Evaluation

We test LSSLs empirically on a range of time series datasets with sequences from length 160 up to 38000 (Sections 5.1 and 5.2), where they substantially improve over prior work. We additionally validate the computational and modeling benefits of LSSLs from generalizing all three main model families (Section 5.3), and analyze the benefits of incorporating principled memory representations that can be learned (Section 5.4).

**Baselines.**   Our tasks have extensive prior work and we evaluate against previously reported best results. We highlight our primary baselines, three very recent works explicitly designed for long sequences: CKConv (a continuous-time CNN) [44], UnICORNN (an ODE-inspired RNN) [47], and Neural Controlled/Rough Differential Equations (NCDE/NRDE) (a sophisticated NDE) [31, 37]. These are the only models we are aware of that have experimented with sequences of length >10k.

## 5.1 Image and Time Series Benchmarks

Table 1: (**Pixel-by-pixel image classification.**) (Top) our methods. (Middle) recurrent baselines. (Bottom) convolutional + other baselines.

| Model | sMNIST | pMNIST | sCIFAR |
|---|---|---|---|
| **LSSL** | **99.53** | **98.76** | **84.65** |
| **LSSL-fixed** | **99.50** | **98.60** | **81.97** |
| LipschitzRNN | 99.4 | 96.3 | 64.2 |
| LMUFFT [12] | - | 98.49 | - |
| UNIcoRNN [47] | - | 98.4 | - |
| HiPPO-RNN [24] | 98.9 | 98.3 | 61.1 |
| URGRU [25] | 99.27 | 96.51 | 74.4 |
| IndRNN [34] | 99.0 | 96.0 | - |
| Dilated RNN [8] | 98.0 | 96.1 | - |
| r-LSTM [56] | 98.4 | 95.2 | 72.2 |
| CKConv [44] | 99.32 | 98.54 | 63.74 |
| TrellisNet [4] | 99.20 | 98.13 | 73.42 |
| TCN [3] | 99.0 | 97.2 | - |
| Transformer [56] | 98.9 | 97.9 | 62.2 |

Table 2: (**Vital signs prediction.**) RMSE for predicting respiratory rate (RR), heart rate (HR), and blood oxygen (SpO2). * indicates our own runs to complete results for the strongest baselines.

| Model | RR | HR | SpO2 |
|---|---|---|---|
| **LSSL** | **0.350** | **0.432** | **0.141** |
| **LSSL-fixed** | **0.378** | **0.561** | **0.221** |
| UnICORNN [47] | 1.06 | 1.39 | 0.869* |
| coRNN [47] | 1.45 | 1.81 | - |
| CKConv | 1.214* | 2.05* | 1.051* |
| NRDE [37] | 1.49 | 2.97 | 1.29 |
| IndRNN [47] | 1.47 | 2.1 | - |
| expRNN [47] | 1.57 | 1.87 | - |
| LSTM | 2.28 | 10.7 | - |
| Transformer | 2.61* | 12.2* | 3.02* |
| XGBoost [55] | 1.67 | 4.72 | 1.52 |
| Random Forest [55] | 1.85 | 5.69 | 1.74 |
| Ridge Regress. [55] | 3.86 | 17.3 | 4.16 |

Table 3: (**Sequential CelebA Classification**.)

| | **LSSL-f** | **ResNet** |
|---|---|---|
| **Att.** | 78.89 | 81.35 |
| **MSO** | 92.36 | 93.92 |
| **Smil.** | 90.95 | 92.89 |
| **WL** | 90.57 | 93.25 |

We test on the sequential MNIST, permuted MNIST, and sequential CIFAR tasks (Table 1), popular benchmarks which were originally designed to test the ability of recurrent models to capture long-term dependencies of length up to 1k [2]. LSSL sets SoTA on sCIFAR by more than 10 points. We note that all results were achieved with at least 5x fewer parameters than the previous SoTA (Appendix F).

We additionally use the BDIMC healthcare datasets (Table 2), a suite of widely studied time series regression problems of length 4000 on estimating vital signs. LSSL reduces RMSE by more than two-thirds on all datasets.

## 5.2 Speech and Image Classification for *Very* Long Time Series

Raw speech is challenging for ML models due to high-frequency sampling resulting in very long sequences. Traditional systems involve complex pipelines that require feeding mixed-and-matched hand-crafted features into DNNs [42]. Table 4 reports results for the Speech Commands (SC) dataset [31] for classification of 1-second audio clips. Few methods have made progress on the raw speech signal, instead requiring pre-processing with standard mel-frequency cepstrum coefficients (MFCC). By contrast, LSSL sets SoTA on this dataset *while training on the raw signal*. We note that MFCC extracts sliding window frequency coefficients and thus is related to the coefficients $x(t)$ defined by LSSL-f (Section 2, Section 4.1, [24], Appendix D). Consequently, LSSL may be interpreted as automatically learning MFCC-type features in a trainable basis.

To stress-test the LSSL's ability to handle extremely long sequences, we create a challenging new sequential-CelebA task, where we classify $178 \times 218$ images = **38000-length** sequences for 4 facial attributes: Attractive (Att.), Mouth Slightly Open (MSO), Smiling (Smil.), Wearing Lipstick (WL) [36]. We chose the 4 most class-balanced attributes to avoid well-known problems with class imbalance. LSSL-f comes close to matching the performance of a specialized ResNet-18 image classification architecture that has $10\times$ the parameters (Table 3). We emphasize we are the first to demonstrate that this is possible to do with a generic sequence model.

## 5.3 Advantages of Recurrent, Convolutional, and Continuous-time Models

We validate that the generality of LSSLs endows it with the strengths of all three families.

**Convergence Speed.** As a recurrent and NDE model that incorporates new theory for continuous-time memory (Section 4.1), the LSSL has strong inductive bias for sequential data, and converges rapidly to SoTA results on our benchmarks. With its convolutional view, training can be parallelized and it is also computationally efficient in practice. Table 5 compares the time it takes the LSSL-f to

Table 4: (**Raw Speech Classification; Timescale Shift**.) (Top): Raw signals (length 16000); $1 \rightarrow f$ indicates test-time change in sampling rate by a factor of $f$. (Bottom): Pre-processed MFCC features used in prior work (length 161). ✗ denotes computationally infeasible.

|  | **LSSL** | **LSSL-f** | CKConv | UnICORNN | N(C/R)DE | ODE-RNN [45] | GRU-ODE [16] |
|---|---|---|---|---|---|---|---|
| $1 \rightarrow 1$ | **95.87** | 90.64 | 71.66 | 11.02 | 16.49 | ✗ | ✗ |
| $1 \rightarrow \frac{1}{2}$ | **88.66** | 78.01 | 65.96 | 11.07 | 15.12 | ✗ | ✗ |
| MFCC | 93.58 | 92.55 | **95.3** | 90.64 | 89.8 | 65.9 | 47.9 |

Table 5: (**Modeling and Computational Benefits of LSSLs.**) In each benchmark category, we compare the number of epochs (ep.) it takes a LSSL-f to reach the previous SoTA (PSoTA) results as well as a near-SoTA target. We also report the wall clock time it took to reach PSoTA relative to the previous best model.

|  | Permuted MNIST | | | BDIMC Heart Rate | | | Speech Commands RAW | | |
|---|---|---|---|---|---|---|---|---|---|
|  | 98% Acc. | PSoTA | Time | 1.5 RMSE | PSoTA | Time | 65% Acc. | PSoTA | Time |
| **LSSL-fixed** | 16 ep. | 104 ep. | 0.19× | 9 ep. | 10 ep. | 0.07× | 9 ep. | 10 ep. | 0.14× |
| CKConv | 118 ep. | 200 ep. | 1.0× | ✗ | ✗ | ✗ | 188 ep. | 280 ep. | 1.0× |
| UnICORNN | 75 ep. | ✗ | ✗ | 116 ep. | 467 ep. | 1.0× | ✗ | ✗ | ✗ |

achieve SoTA, in either sample (measured by epochs) or computational (measured by wall clock) complexity. In all cases, LSSLs reached the target in a fraction of the time of the previous model.

**Timescale Adaptation.** Table 4 also reports the results of continuous-time models that are able to handle unique settings such as missing data in time series, or test-time shift in timescale (we note that this is a realistic problem, e.g., when deployed healthcare models are tested on EEG signals that are sampled at a different rate [48, 49]). We note that many of these baselines were custom designed for such settings, which is of independent interest. On the other hand, LSSLs perform timescale adaptation by simply changing its $\Delta t$ values at inference time, while still outperforming the performance of prior methods with no shift. Additional results on the CharacterTrajectories dataset from prior work [31, 44] are in Appendix F, where LSSL is competitive with the best baselines.

### 5.4 LSSL Ablations: Learning the Memory Dynamics and Timescale

We demonstrate that the $\Delta t$ and $A$ parameters, which LSSLs are able to automatically learn in contrast to prior work, are indeed critical to the performance of these continuous-time models. We note that learning $\Delta t$ adds only $O(H)$ parameters and learning $A$ adds $O(N)$ parameters, adding less than 1% parameter count compared to the base models with $O(HN)$ parameters.

**Memory dynamics $A$.** We validate that vanilla LSSLs suffer from the modeling issues described in Section 4. We tested that LSSLs with *random $A$* matrices (normalized appropriately) perform very poorly (e.g., 62% on pMNIST). Further, we note the consistent increase in performance from LSSL-f to LSSL despite the negligible parameter difference. These ablations show that (i) incorporating the theory of Theorem 1 is actually *necessary* for LSSLs, and (ii) further training the structured $A$ is additionally helpful, which can be interpreted as learning the measure for memorization (Section 4.1).

**Timescale $\Delta t$.** Section 3.2 showed that LSSL's ability to learn $\Delta t$ is its direct generalization of the critical *gating mechanism* of popular RNNs, which previous ODE-based RNN models [12, 24, 47, 58] cannot learn. We note that on sCIFAR, LSSL-f with poorly-specified $\Delta t$ gets only $49.3\%$ accuracy. Additional results in Appendix F show that learning $\Delta t$ alone provides an orthogonal boost to learning $A$, and visualizes the noticeable change in $\Delta t$ over the course of training.

## 6 Discussion

In this work we introduced a simple and principled model (LSSL) inspired by a fundamental representation of physical systems. We showed theoretically and empirically that it generalizes and inherits the strengths of the main families of modern time series models, that its main limitations of long-term memory can be resolved with new theory on continuous-time memorization, and that it is empirically effective on difficult tasks with very long sequences.

**Related work.** The LSSL is related to several rich lines of work on recurrent, convolutional, and continuous-time models, as well as sequence models addressing long dependencies. Appendix A provides an extended related work connecting these topics.

**Tuning.** Our models are very simple, consisting of identical L(*inear*)SSL layers with simple position-wise non-linear modules between layers (Appendix B). Our models were able to train at much higher learning rates than baselines and were not sensitive to hyperparameters, of which we did light tuning primarily on learning rate and dropout. In contrast to previous baselines [4, 31, 44], we did not use hyperparameters for improving stability and regularization such as weight decay, gradient clipping, weight norm, input dropout, etc. While the most competitive recent works introduce at least one hyperparameter of critical importance (e.g. depth and step size [37], $\alpha$ and $\Delta t$ [47], $\omega_0$ [44]) that are difficult to tune, the LSSL-fixed has only $\Delta t$, which the full LSSL can even learn automatically (at the expense of speed).

**Limitations.** Sections 1 and 3 and Fig. 1 mention that a potential benefit of having the recurrent representation of LSSLs may endow it with efficient inference. While this is theoretically possible, this work did not experiment on any applications that leverage this. Follow-up work showed that it is indeed possible in practice to speed up some applications at inference time.

Theorem 2's algorithm is sophisticated (Appendix D) and was not implemented in the first version of this work. A follow-up to this paper found that it is not numerically stable and thus not usable on hardware. Thus the algorithmic contributions in Theorem 2 serve the purpose of a proof-of-concept that fast algorithms for the LSSL do exist in other computation models (i.e., arithmetic operations instead of floating point operations), and leave an open question as to whether fast, numerically stable, and practical algorithms for the LSSL exist.

As described in Appendix B, by freezing the $A$ matrix and $\Delta t$ timescale, the LSSL-fixed is able to be computed much faster than the full LSSL, and is comparable to prior models in practice (Table 5). However, beyond computational complexity, there is also a consideration of space efficiency. Both the LSSL and LSSL-fixed suffer from a large amount of space overhead (described in Appendix B) – using $O(NL)$ instead of $O(L)$ space when working on a 1D sequence of length $L$ – that essentially stems from using the latent state representation of dimension $N$. Consequently, the LSSL can be space inefficient and we used multi-GPU training for our largest experiments (speech and high resolution images, Tables 3 and 4).

These fundamental issues with computation and space complexity were revisited and resolved in follow-up work to this paper, where a new state space model (the Structured State Space) provided a new parameterization and algorithms for state spaces.

**Conclusion and future work.** Modern deep learning models struggle in applications with very long temporal data such as speech, videos, and medical time-series. We hope that our conceptual and technical contributions can lead to new capabilities with simple, principled, and less engineered models. We note that our pixel-level image classification experiments, which use no heuristics (batch norm, auxiliary losses) or extra information (data augmentation), perform similar to early convnet models with vastly more parameters, and is in the spirit of recent attempts at unifying data modalities with a generic sequence model [18]. Our speech results demonstrate the possibility of *learning* better features than hand-crafted processing pipelines used widely in speech applications. We are excited about potential downstream applications, such as training other downstream models on top of pre-trained state space features.

**Acknowledgments**

We thank Arjun Desai, Ananya Kumar, Laurel Orr, Sabri Eyuboglu, Dan Fu, Mayee Chen, Sarah Hooper, Simran Arora, and Trenton Chang for helpful feedback on earlier drafts. We thank David Romero and James Morrill for discussions and additional results for baselines used in our experiments. This work was done with the support of Google Cloud credits under HAI proposals 540994170283 and 578192719349. AR and IJ are supported under NSF grant CCF-1763481. KS is supported by the Wu Tsai Neuroscience Interdisciplinary Graduate Fellowship. We gratefully acknowledge the support of NIH under No. U54EB020405 (Mobilize), NSF under Nos. CCF1763315 (Beyond Sparsity), CCF1563078 (Volume to Velocity), and 1937301 (RTML); ONR under No. N000141712266 (Unifying Weak Supervision); ONR N00014-20-1-2480: Understanding and Applying Non-Euclidean Geometry in Machine Learning; N000142012275 (NEPTUNE); the Moore Foundation, NXP, Xilinx,

LETI-CEA, Intel, IBM, Microsoft, NEC, Toshiba, TSMC, ARM, Hitachi, BASF, Accenture, Ericsson, Qualcomm, Analog Devices, the Okawa Foundation, American Family Insurance, Google Cloud, Salesforce, Total, the HAI-AWS Cloud Credits for Research program, the Stanford Data Science Initiative (SDSI), and members of the Stanford DAWN project: Facebook, Google, and VMWare. The Mobilize Center is a Biomedical Technology Resource Center, funded by the NIH National Institute of Biomedical Imaging and Bioengineering through Grant P41EB027060. The U.S. Government is authorized to reproduce and distribute reprints for Governmental purposes notwithstanding any copyright notation thereon. Any opinions, findings, and conclusions or recommendations expressed in this material are those of the authors and do not necessarily reflect the views, policies, or endorsements, either expressed or implied, of NIH, ONR, or the U.S. Government.

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
