# A Related Work

We provide an extended related work comparing the LSSL to previous recurrent, convolutional, and continuous-time models.

**HiPPO** The LSSL is most closely related to the HiPPO framework for continuous-time memory [24] and its predecessor, the Legendre Memory Unit (LMU) [58]. The HiPPO-RNN and the LMU define dynamics of the form of equation (1), and incorporate it into an RNN architecture. A successor to the LMU, the LMU-FFT [12] keeps the original linear dynamics, allowing the LMU to be computed with a cached convolution kernel.

These methods all suffer from two main limitations. First, the state matrix $A$ and discretization timescale $\Delta t$ cannot be trained due to both limitations in theoretical understanding of which $A$ matrices are effective, as well as computational limitations. Second, (1) is a 1-D to $N$-D map, requiring states to be projected back down to 1-D. This creates an overall 1-D bottleneck in the state, limiting the expressivity of the model.

Compared to these, the LSSL does not use a conventional RNN architecture, instead keeping the linear recurrence (4) and downprojecting it with the second part of the state space representation (5). To avoid the 1-D feature bottlneck, it simply computes $H$ copies of this 1-D to 1-D independently, creating an overall $H$-dimensional sequence-to-sequence model. However, this exacerbates the computational issue, since the work is increased by a factor of $H$.

This work resolves the expressivity issue with new theory. Compared to HiPPO and the LMU, LSSL allows training the $A$ matrix by showing generalized theoretical results for the HiPPO framework, showing that there is a parameterized class of structured state spaces that are HiPPO operators.

The LSSL makes progress towards the second issue with new algorithms for these structured matrices (Theorem 2). However, as noted in Sections 4.2 and 6, the algorithm presented in Theorem 2 was later found to be not practical, and an improved representation and algorithm was found in subsequent work.

**Continuous-time CNNs.** The CKConv is the only example of a continuous-time CNN that we are aware of, and is perhaps the strongest baseline in our experiments. Rather than storing a finite sequence of weights for a convolution kernel, the CKConv parameterizes it as an implicit function from $[0, 1] \rightarrow \mathbb{R}$ which allows sampling it at any resolution. A successor to the CKConv is the FlexConv [43], which learns convolutional kernels with a flexible width. This is similar to the convolution interpretation of LSSL when using certain HiPPO bases (Section 3.2).

**Continuous-time RNNs.** The connection from RNNs to continuous-time models have been known since their inception, and recent years have seen an explosion of CT-RNN (continuous-time RNN) models based on dynamical systems or ODEs. We briefly mention a few classic and modern works along these lines, categorizing them into a few main topics.

First are theoretical works that analyze the expressivity of RNNs from a continuous-time perspective. The connection between RNNs and dynamical systems has been studied since the 90s [22], fleshing out the correspondence between different dynamical systems and RNN architectures [38]. Modern treatments have focused on analyzing the stability [62] and dynamics [29] of RNNs.

Second, a large class of modern RNNs have been designed that aim to combat vanishing gradients from a dynamical systems analysis. These include include the AntisymmetricRNN [7], iRNN [30], and LipschitzRNN [20], which address the exploding/vanishing gradient problem by reparatermizing the architecture or recurrent matrix based on insights from an underlying dynamical system.

Third is a class of models that are based on an explicit underlying ODE introduced to satisfy various properties. This category includes the UnICORNN [47] and its predecessor coRNN [46] which discretize a second-order ODE inspired by oscillatory systems. Other models include the Liquid Time-Constant Networks (LTC) [27] and successor CfC [26], which use underlying dynamical systems with varying time-constants with stable behavior and provable rates of expressivity measured by trajectory length. The LTC is based on earlier dynamic causal models (DCM) [21], which are a particular ODE related to state spaces with an extra bilinear term. Finally, the LMU [58] and HiPPO [24] also fall in this category, whose underlying ODEs are mathematically derived for continuous-time memorization.

Fourth, the recent family of neural ODEs [10], originally introduced as continuous-depth models, have been adapted to continuous-time, spawning a series of "ODE-RNN" models. Examples include the ODE-RNN [45], GRU-ODE-Bayes [16], and ODE-LSTM [32], which extend adjoint-based neural ODEs to the discrete input setting as an alternative to standard RNNs. Neural Controlled Differential Equations (NCDE) [31] and Neural Rough Differential Equations (NRDE) [37] are memory efficient versions that integrate observations more smoothly and can be extended to very long time series.

**Gating mechanisms.**   As a special case of continuous-time RNNs, some works have observed the relation between gating mechanisms and damped dynamical systems [54]. Some examples of continuous-time RNNs based on such damped dynamical systems include the LTC [27] and iRNN [30]. Compared to these, Lemma 3.1 shows a stronger result that sigmoid gates are not just motivated by being an arbitrary monotonic function with range $(0, 1)$, but the *exact formula* appears out of discretizing a damped ODE.

# B   Model Details

## B.1   (M)LSSL Computation

Section 3.1 noted that some of the computations for using the LSSL are expensive to compute. When the LSSL fixes the $A$ and $\Delta t$ parameters (e.g. when they are not trained, or at inference time), these computational difficulties can be circumvented by caching particular computations. In particular, this case applies to the LSSL-f. Note that in this case, the other state-space matrices $C$ and $D$ comprise the $O(HN)$ trainable parameters of the fixed-transition LSSL.

In particular, we assume that there is a black-box inference algorithm for this system, i.e. matrix-vector multiplication by $\overline{A}$ (an example of implementing this black box for a particular structured class is in Appendix E.2). We then compute and cache

- the transition matrix $\overline{A}$, which is computed by applying the black-box $\overline{A}$ MVM algorithm to the identity matrix $I$.

- the *Krylov matrix*

$$K(\overline{A}, \overline{B}) = (\overline{B}, \overline{AB}, (\overline{A})^2 \overline{B}, ...) \in \mathbb{R}^{N \times L}, \tag{8}$$

  which is computed in a parallelized manner by the squaring technique for exponentiation, i.e. batch multiply by $\overline{A}, (\overline{A})^2, (\overline{A})^4, \ldots$.

At inference time, the model can be unrolled recurrently with $\overline{A}$. At training time, the convolutional filter $K_L(\overline{A}, \overline{B}, C)$ (equation (7)) is computed with a matrix multiplication $C \cdot K(\overline{A}, \overline{B})$ before convolving with the input $u$.

Table 7 provides more detailed complexity of this version of the LSSL with fixed $A, \Delta t$.

Note that as mentioned in Section 6, this cached algorithm is fairly fast, but the main drawback is that materializing the Krylov matrix (8) requires $O(NL)$ instead of $O(L)$ space.

## B.2   Initialization of $A$

The LSSL initializes the $A$ parameter in (1) to the HiPPO-LegS operator, which was derived to solve a particular continuous-time memorization problem. This matrix $A \in \mathbb{R}^{N \times N}$ is

$$A_{nk} = \begin{cases} (2n + 1)^{1/2}(2k + 1)^{1/2} & \text{if } n > k \\ n + 1 & \text{if } n = k \ . \\ 0 & \text{if } n < k \end{cases}$$

Note that the LSSL-f is the LSSL with a non-trainable $A$ (and $\Delta t$), so that $A$ is fixed to the above matrix.

### B.3 Initialization of $\Delta t$

One distinction between the LSSL and the most related prior work is that the inclusion of the projection (2) makes the layer a 1-dimensional to 1-dimensional map, instead of 1-D to $N$-D [24, 58]. This enables us to concatenate $H$ copies of this map (at the expense of computation, cf. Section 4.2 and Appendix D). Even when $\Delta t$ is not trained as in the LSSL-f, these $H$ copies allow multiple timescales to be considered by setting $\Delta t$ differently for each copy.

In particular, we initialize $\Delta t$ log-uniformly in a range $\Delta t_{min}, \Delta t_{max}$ (i.e., $\Delta t$ is initialized within this range, such that $\log \Delta t$ is uniformly distributed). The maximum and minimum values were generally chosen to be a factor of $100$ apart such that the length of the sequences in the dataset are contained in this range. Specific values for each model and dataset are in Appendix F. We did not search over these as a hyperparameter, but we note that it can be tuned for additional performance improvements in our experiments.

### B.4 Deep Neural Network Architecture

The Deep LSSL models used in our experiments simply stack together LSSL layers in a simple deep neural network architecture. We note the following architecture details.

**Channels.** The state-space model (1)+(2) accepts a 1-dimensional input $u$, but does not strictly have to return a 1-dimensional output $y$. By making the matrices in (2) dimension $C \in \mathbb{R}^{M \times N}, D \in \mathbb{R}^{M \times 1}$, the output $y$ will be dimension $M$ instead of 1.

We call $M$ the number of *channels* in the model.

**Feedforward.** There are two drawbacks with the current definition of LSSL:

- They are defined by running $H$ independent copies of a state-space model, which means the $H$ input features do not interact at all.
- If the channel dimension is $M > 1$, then the LSSL is a map from dimension 1 to $M$, which means residuals cannot be applied.

These are both addressed by introducing a position-wise feedforward layer after the LSSL of shape $H \cdot M \rightarrow H$. This simultaneously mixes the hidden features, and projects the output back to dimension 1 if necessary. There is also an optional non-linearity in between the LSSL and this feedforward projection; we fix it to the GeLU activation function in our models.

We note that this factorization of parallel convolutions on the $H$ features followed by a position-wise linear map is very similar to depth-wise separable convolutions [13].

**Residuals and normalization.** To stack multiple layers of LSSLs together, we use very standard architectures for deep neural networks. In particular, we use residual connections and a layer normalization (either pre-norm or post-norm) in the style of standard Transformer architectures. Whether to use pre-norm or post-norm was chosen on a per-dataset basis, and depended on whether the model overfit; recent results have shown that pre-norm architectures are more stable [15, 35], so we used it on harder datasets with less overfitting. We note that we could have additionally inserted MLP modules in between LSSL layers, in the style of Transformers [57], but did not experiment with this.

**Parameter count.** The overall parameter count of an LSSL model is $M \cdot H \cdot (H + N)$.

We primarily used two model sizes in our experiments, which were chosen simply to produce round numbers of parameters:

- LSSL small ($\approx 200K$ parameters): 6 layers, $H = 128, N = 128, M = 1$.
- LSSL large ($\approx 2M$ parameters): 4 layers, $H = 256, N = 256, M = 4$.

We did not search over additional sizes, but for some datasets reduced the model size for computational reasons.

## C    LSSL Proofs

This section gives refinements of the statements in Section 3, additional results, and proofs of all results.

Appendix C.1 has a more detailed (and self-contained) summary of basic methods in ODE approximation which will be used in the results and proofs.

Appendix C.2 give more general statements and proofs of Lemma 3.1 and Lemma 3.2 in Lemma C.1 and Theorem 4, respectively.

### C.1    Approximations of ODEs

We consider the standard setting of a first-order initial value problem (IVP) ordinary differential equation (ODE) for a continuous function $f(t, x)$

$$\begin{aligned} \dot{x}(t) &= f(t, x(t)) \\ x(t_0) &= x_0 \end{aligned}. \tag{9}$$

This differential form has an equivalent integral form

$$x(t) = x_0 + \int_{t_0}^{t} f(s, x(s)) \, ds. \tag{10}$$

Appendices C.1.1 and C.1.2 overview the Picard theorem and first-order numerical integration methods, which apply to any IVP (9). Appendix C.1.3 then shows how to specialize it to linear systems as in equation (1).

At a high level, the basic approximation methods considered here use the integral form (10) and approximate the integral in the right-hand side by simple techniques.

### C.1.1    Picard Iteration

The **Picard-Lindelöf Theorem** gives sufficient conditions for the existence and uniqueness of solutions to an IVP. As part of the proof, it provides an iteration scheme to compute this solution.

**Theorem 3** (Picard-Lindelöf)**.** *In the IVP* (9)*, if there is an interval around $t_0$ such that $f$ is Lipschitz in its second argument, then there is an open interval $I \ni t_0$ such that there exists a unique solution $x(t)$ to the IVP in $I$. Furthermore, the sequence of **Picard iterates** $x^{(0)}, x^{(1)}, \dots$ defined by*

$$\begin{aligned} x^{(0)}(t) &= x_0 \\ x^{(\ell)}(t) &= x_0 + \int_{t_0}^{t} f(s, x^{(\ell-1)}(s)) \, ds \end{aligned}$$

*converges to $x$.*

The Picard iteration can be viewed as approximating (10) by holding the previous estimate of the solution $x^{(\ell-1)}$ fixed inside the RHS integral.

### C.1.2    Numerical Integration Methods

Many methods for numerical integration of ODEs exist, which calculate discrete-time approximations of the solution. We discuss a few of the simplest methods, which are first-order methods with local error $O(h^2)$ [6].

These methods start by discretizing (10) into the form

$$x(t_k) - x(t_{k-1}) = \int_{t_{k-1}}^{t_k} f(s, x(s)) \, ds. \tag{11}$$

Here we assume a sequence of discrete times $t_0, t_1, t_2, \dots$ is fixed. For convenience, let $x_k$ denote $x(t_k)$ and let $\Delta t_k := t_k - t_{k-1}$. The goal is now to approximate the integral in the RHS of (11).

**Euler method.** The Euler method approximates (11) by holding the left endpoint constant throughout the integral (i.e., the "rectangle rule" with left endpoint), $f(s, x(s)) \approx f(t_{k-1}, x(t_{k-1}))$. The discrete-time update becomes

$$
\begin{aligned}
x_k - x_{k-1} &= (t_k - t_{k-1}) f(t_{k-1}, x(t_{k-1})) \\
&= \Delta t_k f(t_{k-1}, x_{k-1}).
\end{aligned}
\tag{12}
$$

**Backward Euler method.** The backward Euler method approximates (11) by holding the right endpoint constant throughout the integral (i.e., the "rectangle rule" with right endpoint), $f(s, x(s)) \approx f(t_k, x(t_k))$. The discrete-time update becomes

$$
\begin{aligned}
x_k - x_{k-1} &= (t_k - t_{k-1}) f(t_k, x(t_k)) \\
&= \Delta t_k f(t_k, x_k).
\end{aligned}
\tag{13}
$$

### C.1.3 Discretization of State-Space Models

In the case of a linear system, the IVP is specialized to the case

$$
f(t, x(t)) = Ax(t) + Bu(t).
$$

Note that here $u$ is treated as a fixed external input, which is constant from the point of view of this ODE in $x$. Let $u_k$ denote the average value in each discrete time interval,

$$
u_k = \frac{1}{\Delta t_k} \int_{t_{k-1}}^{t_k} u(s)\, ds.
$$

The integral equation (11) can be specialized to this case, and more generally a convex combination of the left and right endpoints can be taken to approximate the integral, weighing them by $1 - \alpha$ and $\alpha$ respectively. Note that the case $\alpha = 0, 1$ are specializations of the forward and backward Euler method, and the case $\alpha = \frac{1}{2}$ is the classic "trapezoid rule" for numerical integration.

$$
\begin{aligned}
x(t_k) - x(t_{k-1}) &= \int_{t_{k-1}}^{t_k} Ax(s)\, ds + \int_{t_{k-1}}^{t_k} Bu(s)\, ds \\
&= \int_{t_{k-1}}^{t_k} Ax(s)\, ds + \Delta t_k Bu_k \\
&\approx \Delta t_k \left[ (1 - \alpha) Ax_{k-1} + \alpha Ax_k \right] + \Delta t_k Bu_k.
\end{aligned}
$$

Rearranging yields

$$
\begin{aligned}
(I - \alpha \Delta t_k \cdot A) x_k &= (I + (1 - \alpha) \Delta t_k \cdot A) x_{k-1} + \Delta t_k \cdot Bu_k \\
x_k &= (I - \alpha \Delta t_k \cdot A)^{-1} (I + (1 - \alpha) \Delta t_k \cdot A) x_{k-1} + (I - \alpha \Delta t_k \cdot A)^{-1} \Delta t_k \cdot Bu_k
\end{aligned}
$$

This derives the **generalized bilinear transform (GBT)** [61]. The **bilinear method** is the case $\alpha = \frac{1}{2}$ of special significance, and was numerically found to be better than the forward and backward Euler methods $\alpha = 0, 1$ both in synthetic function approximation settings and in end-to-end experiments [24, Figure 4].

### C.2 RNNs are LSSLs: Proof of Results in Section 3.2

We provide more detailed statements of Lemmas 3.1 and 3.2 from Section 3.2. In summary, LSSLs and popular families of RNN methods all approximate the same continuous-time dynamics

$$
\dot{x}(t) = -x + f(t, x(t))
\tag{14}
$$

by viewing them with a combination of two techniques.

We note that these results are about two of the most commonly used architecture modifications for RNNs. First, the gating mechanism is ubiquitous in RNNs, and usually thought of as a heuristic for smoothing optimization [28]. Second, many of the effective large-scale RNNs use linear (gated) recurrences and deeper models, which is usually thought of as a heuristic for computational efficiency [5]. Our results suggest that neither of these are heuristics after all, and arise from standard ways to approximate ODEs.

To be more specific, we show that:

Table 6: A summary of the characteristics of popular RNN methods and their approximation mechanisms for capturing the dynamics $\dot{x}(t) = -x(t) + f(t, x(t))$ (equation (14)). The LSSL entries are for the very specific case with order $N = 1$ and $A = -1, B = 1, C = 1, D = 0$; LSSLs are more general.

| Method | RNN | RNN | LSSL | LSSL |
|---|---|---|---|---|
| *Variant* | Gated | Gated, linear | Discrete (4)+(5) | Continuous (1)+(2) |
| *Special cases* | LSTM [28], GRU [14] | QRNN [5], SRU [33] | | |
| Deep? | Single-layer | Deep | Deep | Deep |
| Continuous? | Discrete-time | Discrete-time | Discrete-time | Continuous-time |
| Linear? | Non-linear | Linear | Linear | Linear |
| *Approximation* | | | | |
| **Depth-wise** | - | Picard iteration | Picard iteration | Picard iteration |
| **Time-wise** | Backwards Euler | GBT($\alpha = 1$) | GBT($\alpha = \frac{1}{2}$) (i.e. Bilinear) | - |

- Non-linear RNNs discretize the dynamics (14) by applying backwards Euler discretization to the linear term, which arises in the gating mechanism of RNNs (Appendix C.2.2, Lemma C.1).

- A special case of LSSLs approximates the dynamics (14) (in continuous-time) by applying Picard iteration to the non-linear term (Appendix C.2.3, Theorem 4).

- Deep linear RNNs approximate the dynamics (14) with both Picard iteration in the depth direction to linearize the non-linear term, and discretization (gates) in the time direction to discretize the equation (Appendix C.2.4, Corollary C.3).

A comparison is summarized in Table 6.

In the remainder of this section, we assume that there is an underlying function $x(t)$ that satisfies (14) on some interval for any initial condition, and that $f$ is continuous and Lipschitz in its second argument. Our goal is to show that several families of models approximate this in various ways.

### C.2.1 Intuition / Proof Sketches

We sketch the idea of how LSSLs capture popular RNNs. More precisely, we will show how approximating the dynamics (14) in various ways lead to types of RNNs and LSSLs.

The first step is to look at the simpler dynamics

$$\dot{x}(t) = -x(t) + u(t)$$

where there is some input $u(t)$ that is independent of $x$. (In other words, in (14), the function $f(t, x)$ does not depend on the second argument.)

By directing applying the GBT discretization with $\alpha = 1$, this leads to a gated recurrence (Lemma 3.1).

The second step is that by applying the backwards Euler discretization more directly to (14), this leads to a gated RNN where the input can depend on the state (Lemma C.1).

Alternatively, we can apply Picard iteration on (14), which says that the iteration

$$x^{(\ell)}(t) = x_0 + \int_{t_0}^{t} -x^{(\ell-1)}(s)\, ds + \int_{t_0}^{t} f(s, x^{(\ell-1)}(s))\, ds$$

converges to the solution $x(t)$.

However, the first integral term is simple and can be tightened. We can instead try to apply Picard iteration on only the second term, leaving the first integral in terms of $x^{(\ell)}$. Intuitively this should still converge to the right solution, since this is a weaker iteration; we're only using the Picard approximation on the second term.

$$x^{(\ell)}(t) = x_0 + \int_{t_0}^{t} -x^{(\ell)}(s)\, ds + \int_{t_0}^{t} f(s, x^{(\ell-1)}(s))\, ds$$

Differentiating, this equation is the ODE

$$\dot{x}^{(\ell)}(t) = -x^{(\ell)}(t) + f(t, x^{(\ell-1)}(t))$$

This implies that alternating point-wise functions with a simple linear ODE $\dot{x}^{(\ell)}(t) = -x^{(\ell)}(t) + u^{(\ell)}(t)$ also captures the dynamics (14). But this is essentially what an LSSL is.

To move to discrete-time, this continuous-time layer can be discretized with gates as in Lemma 3.1, leading to deep linear RNNs such as the QRNN, or with the bilinear discretization, leading to the discrete-time LSSL. We note again that in the discrete-time LSSL, $\overline{A}$ and $\overline{B}$ play the role of the gates $\sigma, 1 - \sigma$.

### C.2.2 Capturing gates through discretization

**Lemma C.1.** *Consider an RNN of the form*

$$x_k = (1 - \sigma(z_k))x_{k-1} + \sigma(z_k)\overline{f}(k, x_{k-1}), \tag{15}$$

*where $\overline{f}(k, x)$ is an arbitrary function that is Lipschitz in its second argument (e.g., it may depend on an external input $u_k$).*

*Then equation (15) is a discretization of the dynamics (14) with step sizes $\Delta t_k = \exp(z_k)$, i.e. $x_k \approx x(t_k)$ where $t_k = \sum_{i=1}^{k} \Delta t_i$.*

*Proof.* Apply the backwards Euler discretization (13) to equation (14) to get

$$x_k - x_{k-1} = \Delta t_k \left[ -x_k + f(t_k, x_k) \right]$$
$$(1 + \Delta t_k)x_k = x_{k-1} + \Delta t_k f(t_k, x_k)$$
$$x_k = \frac{1}{1 + \Delta t_k} x_{k-1} + \frac{\Delta t_k}{1 + \Delta t_k} f(t_k, x_k).$$

Note that $\frac{\Delta t_k}{1+\Delta t_k} = \frac{e^{z_k}}{1+e^{z_k}} = \frac{1}{1+e^{-z_k}}$ and $\frac{1}{1+\Delta t_k} = 1 - \frac{\Delta t_k}{1+\Delta t_k}$, thus

$$x_k = (1 - \sigma(z_k))x_{k-1} + \sigma(z_k)\overline{f}(k, x_{k-1}).$$

Here we are denoting $\overline{f}(k, x) = f(t_k, x)$ to be a discrete-time version of $f$ evaluatable at the given timesteps $t_k$. □

Note that a potential external input function $u(t)$ or sequence $u_k$ is captured through the abstraction $f(t, x)$. For example, a basic RNN could define $\overline{f}(k, x) = f(t_k, x) = \tanh(Wx + Uu_k)$.

### C.2.3 Capturing non-linearities through Picard iteration

The main result of this section is Theorem 4 showing that LSSLs can approximate the same dynamics as the RNNs in the previous section. This follows from a technical lemma.

**Lemma C.2.** *Let $f(t, x)$ be any function that satisfies the conditions of the Picard-Lindelöf Theorem (Theorem 3).*

*Define a sequence of functions $x^{(\ell)}$ by alternating the (point-wise) function $f$ with solving an ODE*

$$x^{(0)}(t) = x_0$$
$$u^{(\ell)}(t) = f(t, x^{(\ell-1)}(t))$$
$$\dot{x}^{(\ell)}(t) = Ax^{(\ell)}(t) + u^{(\ell)}(t).$$

*Then $x^{(\ell)}$ converges to a solution $x^{(\ell)}(t) \to x(t)$ of the IVP*

$$\dot{x}(t) = Ax(t) + f(t, x(t))$$
$$x(t_0) = x_0.$$

**Theorem 4.** *A (continuous-time) deep LSSL with order $N = 1$ and $A = -1, B = 1, C = 1, D = 0$ approximates the non-linear dynamics (14).*

*Proof.* Applying the definition of an LSSL (equations (1)+(2)) with these parameters results in a layer mapping $u(t) \mapsto y(t)$ where $y$ is defined implicitly through the ODE

$$\dot{y}(t) = -y(t) + u(t).$$

This can be seen since the choice of $C, D$ implies $y(t) = x(t)$ and the choice of $A, B$ gives the above equation.

Consider the deep LSSL defined by alternating this LSSL with position-wise (in time) non-linear functions

$$u^{(\ell)}(t) = f(t, y^{(\ell-1)}(t))$$
$$\dot{y}^{(\ell)}(t) = -y^{(\ell)}(t) + u^{(\ell)}(t).$$

But this is exactly a special case of Lemma C.2, so that we know $y^{(\ell)}(t) \to y(t)$ such that $y(t)$ satisfies

$$\dot{y}(t) = -y(t) + f(t, y(t))$$

as desired. $\qquad \square$

*Proof of Lemma C.2.* Let

$$z(t) = e^{-At} x(t)$$

(and $z_0 = z(t_0) = x(t_0) = x_0$). Note that

$$\begin{aligned}
\dot{z}(t) &= e^{-At} \left[ \dot{x}(t) - Ax(t) \right] \\
&= e^{-At} f(t, x(t)) \\
&= e^{-At} f(t, e^{At} z(t)).
\end{aligned}$$

Since $f$ satisfies the conditions of the Picard Theorem (i.e., is continuous in the first argument and Lipschitz in the second), so does the function $g$ where $g(t, x) := e^{-At} f(t, e^{At} x)$ for some interval around the initial time.

By Theorem 3, the iterates $z^{(\ell)}$ defined by

$$z^{(\ell)}(t) = z_0 + \int_{t_0}^{t} e^{-As} f(s, e^{As} z^{(\ell-1)}(s)) \, ds \tag{16}$$

converges to $z$.

Define $x^{(\ell)}(t) = e^{At} z^{(\ell)}(t)$. Differentiate (16) to get

$$\begin{aligned}
\dot{z}^{(\ell)}(t) &= e^{-At} f(t, e^{At} z^{(\ell-1)}(t)) \\
&= e^{-At} f(t, x^{(\ell-1)}(t)) \\
&= e^{-At} u^{(\ell)}(t).
\end{aligned}$$

But

$$\dot{z}^{(\ell)}(t) = e^{-At} \left[ \dot{x}^{(\ell)}(t) - Ax^{(\ell)}(t) \right],$$

so

$$\dot{x}^{(\ell)}(t) = Ax^{(\ell)}(t) + u^{(\ell)}(t).$$

Since $z^{(\ell)} \to z$ and $x^{(\ell)}(t) = e^{At} z^{(\ell)}(t)$ and $x(t) = e^{At} z(t)$, we have $x^{(\ell)} \to x$. $\qquad \square$

### C.2.4   Capturing Deep, Linear, Gated RNNs

We finally note that several types of RNNs exist which were originally motivated by approximating linearizing gated RNNs for speed. Although these were treated as a heuristic for efficiency reasons, they are explained by combining our two main technical results.

Lemma C.1 shows that a single-layer, discrete-time, non-linear RNN approximates the dynamics (14) through discretization, which arises in the gating mechanism.

Theorem 4 shows that a deep, continuous-time, linear RNN approximates (14) through Picard iteration, where the non-linearity is moved to the depth direction.

Combining these two results leads to Corollary C.3, which says that a deep, discrete-time, linear RNN can also approximate the same dynamics (14).

**Corollary C.3.** *Consider a deep, linear RNN of the form*

$$x_k^{(\ell)} = (1 - \sigma(z_k))x_{k-1}^{(\ell)} + \sigma(z_k)u_k^{(\ell)}$$
$$u_k^{(\ell)} = \overline{f}(k, x_k^{(\ell-1)}).$$

*This is a discretization of the dynamics* (14) *with step sizes* $\Delta t_k = \exp(z_k)$, *i.e.* $x_k \approx x(t_k)$ *where* $t_k = \sum_{i=1}^k \Delta t_i$.

*Proof.* By Lemma C.1, the first equation is a discretization of the continuous-time equation

$$\dot{x}^{(\ell)}(t) = -x^{(\ell)}(t) + u^{(\ell)}(t)$$

where

$$u^{(\ell)}(t) = f(t, x^{(\ell-1)}(t))$$

uses the continuous-time version $f$ of $\overline{f}$. But by Lemma C.2, this is an approximation of the dynamics (14) using Picard iteration. $\qquad\square$

Notable examples of this type of model include the Quasi-RNN or QRNN [5] and the Simple Recurrent Unit (SRU) [33], which are among the most effective models in practice. We remark that these are the closest models to the LSSL and suggest that their efficacy is a consequence of the results of this section, which shows that they are not heuristics.

We note that there are many more RNN variants that use a combination of these gating and linearization techniques that were not mentioned in this section, and can be explained similarly.

# D LSSL Proofs and Algorithms

This section proves the results in Section 4.1, and is organized as follows:

- Appendix D.1 gives a self-contained synopsis of the HiPPO framework [24].
- Appendix D.2 proves Theorem 1, which shows that the hippo operators for any measure lead to a simple linear ODE of the form of equation (1).
- Appendix D.3 proves Corollary 4.1, including a formal definition of quasiseparable matrices (i.e., how LSSL matrices are defined) in Definition 4.

**Notation** This section is technically involved and we adopt notation to simplify reasoning about the shapes of objects. In particular, we use bold capitals (e.g. $\mathbf{A}$) to denote matrices and bold lowercase (e.g. $\mathbf{b}$) to denote vectors. For example, equation (1) becomes $\dot{x} = \mathbf{A}x + \mathbf{b}u$. These conventions are adopted throughout Appendices D and E.

## D.1 Preliminaries: HiPPO Framework and Recurrence Width

This section summarizes technical preliminaries taken directly from prior work. We include this section so that this work is self-contained and uses consistent notation, which may deviate from prior work. For example, we use modified notation from Gu et al. [24] in order to follow conventions in control theory (e.g., we denote input by $u$ and state by $x$ as in (1)).

Appendix D.1.1 formally defines the HiPPO operator mathematically as in [24, Section 2.2], and Appendix D.1.2 overviews the steps to derive the HiPPO operator as in [24, Appendix C]. Appendix D.1.3 defines the class of Low Recurrence Width (LRW) matrices, which is the class of matrices that our generalization of the HiPPO results (Theorem 1) uses.

### D.1.1 Definition of HiPPO Operator

**Definition 1** ([24], Definition 1). *Given a time-varying measure $\mu^{(t)}$ supported on $(-\infty, t]$, an N-dimensional subspace $\mathcal{G}$ of polynomials, and a continuous function $u : \mathbb{R}_{\geq 0} \to \mathbb{R}$, HiPPO defines a* projection *operator* $\mathrm{proj}_t$ *and a* coefficient extraction *operator* $\mathrm{coef}_t$ *at every time t, with the following properties:*

1. $\mathrm{proj}_t$ *takes a function u restricted up to time t, $u_{\leq t} := u(x)|_{x \leq t}$, and maps it to a polynomial $g^{(t)} \in \mathcal{G}$, that minimizes the approximation error $\|u_{\leq t} - g^{(t)}\|_{L_2(\mu^{(t)})}$.*

2. $\mathrm{coef}_t : \mathcal{G} \to \mathbb{R}^N$ *maps the polynomial $g^{(t)}$ to the coefficients $c(t) \in \mathbb{R}^N$ of the basis of orthogonal polynomials defined with respect to the measure $\mu^{(t)}$.*

*The composition* $\mathrm{coef}_t \circ \mathrm{proj}_t$ *is called* hippo, *which is an operator mapping a function $u : \mathbb{R}_{\geq 0} \to \mathbb{R}$ to the optimal projection coefficients $c : \mathbb{R}_{\geq 0} \to \mathbb{R}^N$ (i.e $(\mathrm{hippo}(u))(t) = \mathrm{coef}_t(\mathrm{proj}_t(f))$.*

### D.1.2 HiPPO Framework for Deriving the HiPPO Operator

The main ingredients of HiPPO consists of an approximation measure and an orthogonal polynomial basis. We recall how they are defined in [24] (we note that compared to Gu et al. [24], our notation has changed from input $f(t)$ coefficients (state) $c(t)$ to input $u(t)$ and coefficients (state) $x(t)$, following conventions in controls).

**Approximation Measures** At every $t$, the approximation quality is defined with respect to a measure $\mu^{(t)}$ supported on $(-\infty, t]$. We assume that the measures $\mu^{(t)}$ have densities $\omega(t, Y) := \frac{d\mu^{(t)}}{dY}$. Note that this implies that integrating with respect to $d\mu^{(t)}$ is the same as integrating with respect to $\omega(t, Y) \, dY$.

**Orthogonal Polynomial basis** Let $\{P_n^{(t)}\}_{n \in \mathbb{N}}$ denote a sequence of orthogonal polynomials with respect to some time-varying measure $\mu^{(t)}$. Let $p_n^{(t)}$ be the normalized version of of orthogonal $P_n^{(t)}$, and define

$$p_n(t, Y) = p_n^{(t)}(Y).$$

In particular, the above implies that

$$\int_{-\infty}^t p_n^{(t)}(Y) \cdot p_m^{(t)}(Y) \omega(t, Y) \, dY = \delta_{m,n}.$$

In the general framework, HiPPO does not require an orthogonal polynomial basis as the selected basis. The choice of basis is generalized by tilting with $\chi$.

**Tilted measure and basis** For any scaling function $\chi(t, Y)$, the functions $p_n(t, Y)\chi(t, Y)$ are orthogonal with respect to the density $\frac{\omega}{\chi^2}$ at every time $t$. Define $\nu(t)$ to be the normalized measure with density proportional to $\frac{\omega}{\chi^2}$, with normalization constant $\zeta(t) = \int_0^t \frac{\omega(t, Y)}{\chi(t, Y)^2} dx$.

We express the coefficients $x_n(t)$ calculated by the HiPPO framework as:

$$x_n(t) = \frac{1}{\sqrt{\zeta(t)}} \int_0^t u(Y) p_n(t, Y) \frac{\omega(t, Y)}{\chi(t, Y)} \, dY. \tag{17}$$

To use this to derive $\dot{x}_n(t)$, let $h(t, Y) = u(Y) p_n(t, Y) \omega(t, Y)$. We see that

$$\dot{x}_n(t) = \frac{\mathrm{d}}{\mathrm{d}t} \int_0^t u(Y) p_n(t, Y) \omega(t, Y) \mathrm{d}Y$$

$$= \frac{\mathrm{d}}{\mathrm{d}t} \int_0^t h(t, Y) \mathrm{d}Y$$

$$= \int_0^t \frac{\partial}{\partial t} h(t, Y) \mathrm{d}Y + h(t, t)$$

$$= \int_0^t u(Y) \left( \frac{\partial}{\partial t} p_n(t, Y) \right) \omega(t, Y) \mathrm{d}Y + \int_0^t f(Y) p_n(t, Y) \left( \frac{\partial}{\partial t} \omega(t, Y) \right) \mathrm{d}Y$$

$$\quad + u(t) p_n(t, t) \omega(t, t).$$

This allows $\dot{x}_n(t)$ to be written as

$$\dot{x}_n(t) = u(t) p_n(t, t) \omega(t, t) + \int_0^t u(Y) \left( \frac{\partial}{\partial t} p_n(t, Y) \right) \omega(t, Y) \mathrm{d}Y$$

$$\quad + \int_0^t f(Y) p_n(t, Y) \left( \frac{\partial}{\partial t} \omega(t, Y) \right) \mathrm{d}Y. \tag{18}$$

Although Gu et al. [24] describe the framework in the full generality above and use $\chi$ as another degree of freedom, in their concrete derivations they always fix $\chi = \omega$. Our general results also use this setting. For the remainder of this section, we assume the "full tilting" case $\chi = \omega$. In particular, this means that in Eq. (18), we essentially substitute $\omega$ above with 1 and divide each term by the inverse square root of our normalization constant, $\zeta$, to get the coefficient dynamics that we will use in our arguments:

$$\dot{x}_n(t) = \frac{1}{\sqrt{\zeta(t)}} u(t) p_n(t, t) + \frac{1}{\sqrt{\zeta(t)}} \int_0^t u(Y) \left( \frac{\partial}{\partial t} p_n(t, Y) \right) \mathrm{d}Y \tag{19}$$

Now, if we can show that each of the integrated terms in (18) are linear combinations of $x_n(t)$, this would be the same as saying that $\dot{x}_n(t) = \mathbf{A}(t) x(t) + \mathbf{b}(t) u(t)$ for some $\mathbf{A}(t)$. Therefore, the incremental update operation would be bounded by the runtime of the matrix-vector operation $\mathbf{A}(t) x(t)$.

### D.1.3 Recurrence Width

Our final goal is to show that $\dot{x}_n(t) = \mathbf{A}(t) x(t) + \mathbf{b}(t) u(t)$ for some $\mathbf{A}(t)$ with constant recurrence width (see Definition 2). This will show Theorem 1, and also imply that the MVM $\mathbf{A}(t) x(t)$ can be computed in $\tilde{O}(N)$ time. To build this argument, we borrow the fact that OPs all have recurrence width 2 and results regarding matrix-vector multiplication of matrices with constant recurrence width along with their inverses.

**Definition 2** ([17]). *An $N \times N$ matrix $\mathbf{A}$ has recurrence width $t$ if the polynomials $a_i(X) = \sum_{j=0}^{N-1} \mathbf{A}[i, j] X^j$ satisfy $\deg(a_i) \leq i$ for $i < t$, and*

$$a_i(X) = \sum_{j=1}^t g_{i,j}(X) a_{i-j}(X)$$

*for $i \geq t$, where the polynomials $g_{i,j} \in \mathbb{R}[X]$ have degree at most $j$.*

**Theorem 5** ([17], Theorem 4.4). *For any $N \times N$ matrix $\mathbf{A}$ with constant recurrence width, any vector $\mathbf{x} \in \mathbb{R}^n$, $\mathbf{A}\mathbf{x}$ can be computed with $\tilde{O}(N)$ operations over $\mathbb{R}$.*

**Theorem 6** ([17], Theorem 7.1). *For any $N \times N$ matrix $\mathbf{A}$ with constant recurrence width, any vector $\mathbf{x} \in \mathbb{R}^n$, $\mathbf{A}^{-1}\mathbf{x}$ can be computed with $\tilde{O}(N)$ operations over $\mathbb{R}$.*

For the rest of the note we'll assume that any operation over $\mathbb{R}$ can be done in constant time. It would be useful for us to define $\mathbf{P} \in \mathbb{R}^{N \times N}$ such that the coefficients of the OP $p_i(X)$, i.e. $p_i(X) = \sum_{j=0}^{N-1} \mathbf{P}[i, j] X^j$.

### D.2 Proof of Theorem 1

This section proves Theorem 1, which is restated formally in Corollary D.4. Appendix D.2.1 proves some results relating orthogonal polynomials to recurrence width (Appendix D.1.3). Appendix D.2.2 proves Corollary D.4. Appendices D.2.3 and D.2.4 provides examples showing how Corollary D.4 can be specialized to exactly recover the HiPPO-LegT, HiPPO-LagT, HiPPO-LegS methods [24].

#### D.2.1 Relating Orthogonal Polynomials and Recurrence Width

Next we introduce the following lemma, which will be useful in our arguments:

**Lemma D.1.** *For any $n$, there exists ordered sets of coefficients $\alpha_n = \{\alpha_{n,i}\}, \beta_n = \{\beta_{n,i}\}$,*

*(i)* $p'_n(Z) = \sum_{i=0}^{n-1} \alpha_{n,i} p_i(Z)$

*(ii)* $Z p'_n(Z) = \sum_{i=0}^{n-1} \beta_{n,i} p_i(Z)$

*Proof.* Follows from the fact that $p_i(z)$ for $0 \le i < N$ forms a basis and the observation of the degrees of the polynomials on the LHS. $\square$

The following matrices will aid in showing verifying that matrix vector multiplication with a given matrix $\mathbf{A}$ can be computed in $O(\tilde{N})$ time.

**Definition 3.** $\mathbf{D}_1, \mathbf{D}_2 \in \mathbb{R}^{N \times N}$ *are the matrices such that*

$$p'_i(Z) = \sum_{j=0}^{N-1} \mathbf{D}_1[i,j] Z^j, \text{ and } Z p'_i(Z) = \sum_{j=0}^{N-1} \mathbf{D}_2[i,j] Z^j.$$

Let $\mathbf{S}$ be the "right shift" matrix, i.e. for any matrix $\mathbf{M}$, $\mathbf{MS}$ has the columns of $\mathbf{M}$ shifted to right by one. Note that $\mathbf{S}^T$ corresponds to the "left shift" matrix.

We now note that:

**Lemma D.2.** $\mathbf{D}_1 = \mathbf{P} \cdot \text{diag}(0, 1, \ldots, N-1) \cdot \mathbf{S}^T$ *and* $\mathbf{D}_2 = \mathbf{P} \cdot \text{diag}(0, 1, \ldots, N-1)$. *In particular,* $\mathbf{D}_1 \mathbf{z}$ *and* $\mathbf{D}_2 \mathbf{z}$ *can be computed in* $\tilde{O}(N)$ *time for any* $\mathbf{z} \in \mathbb{R}^N$.

*Proof.* Recall that $\mathbf{P}$ has the coefficients of the OP polynomials $p_0(Z), \ldots, p_{N-1}(Z)$ as its rows. Then note that

$$p'_n(Z) = \sum_{i=0}^{n-1} i \cdot \mathbf{P}[n,i] \cdot Z^{i-1}. \tag{20}$$

The claim on $\mathbf{D}_1 = \mathbf{P} \cdot \text{diag}(0, 1, \ldots, N-1) \cdot \mathbf{S}^T$ follows from the above. Recall that $\mathbf{D}$ has recurrence width of 2. The claim on the runtime of computing $\mathbf{D}_1 \mathbf{z}$ then follows from Theorem 5 and the fact that both $\text{diag}(0, 1, \ldots, N-1)$ and $\mathbf{S}^T$ is $n$-sparse.

From Eq. (20), it is easy to see that

$$Z p'_n(Z) = \sum_{i=0}^{n-1} i \cdot \mathbf{P}[n,i] \cdot Z^i.$$

The claim on the structure of $\mathbf{D}_2$ then follows from the above expression. The claim on runtime of computing $\mathbf{D}_2 \mathbf{z}$ follows from essentially the same argument as for $\mathbf{D}_1 \mathbf{z}$.

$\square$

Finally, we make the following observation:

**Lemma D.3.** *Let $\mathbf{A}'$ and $\mathbf{B}'$ be defined such that $\mathbf{A}'[n,i] = \alpha_{n,i}$ and $\mathbf{B}'[n,i] = \beta_{n,i}$. Then both $\mathbf{A}'$ and $\mathbf{B}'$ are both products of three matrices: two of which have recurrence width at most 2 and the third is the inverse of a matrix that has recurrence width 2.*

*Proof.* We note that since $\mathbf{P}$ expresses the orthogonal polynomials in standard basis, $\mathbf{P}^{-1}$ changes from OP basis to standard basis. This along with Lemma D.2 implies that $\mathbf{A} = \mathbf{P} \cdot \left(\mathrm{diag}(0, 1, \ldots, N-1) \cdot \mathbf{S}^T\right) \cdot \mathbf{P}^{-1}$. It is easy to check that $\mathrm{diag}(0, 1, \ldots, N-1) \cdot \mathbf{S}^T$ has recurrence width 1 and the claim on $\mathbf{A}'$ follows since $\mathbf{P}$ has recurrence width 2. A similar argument proves the claim on $\mathbf{B}'$. $\qquad\square$

### D.2.2 HiPPO for General Measures

Let $\theta : \mathbb{R}_{\geq 0} \mapsto \mathbb{R}_{\geq 0}$ be a function such that for all $t$, $\theta(t) \leq t$ and $\theta(t)$ is differentiable.

In what follows, define

$$z = \frac{2(Y - t)}{\theta(t)} + 1.$$

We note that

$$\frac{\mathrm{d}z}{\mathrm{d}Y} = \frac{2}{\theta(t)}. \tag{21}$$

Further, note that:

$$\begin{aligned}
\frac{\mathrm{d}z}{\mathrm{d}t} &= \frac{\mathrm{d}}{\mathrm{d}t}\left(\frac{2(Y - t)}{\theta(t)} + 1\right) \\
&= -\frac{2}{\theta(t)} - \frac{2(Y - t)\theta'(t)}{\theta^2(t)} \\
&= -\frac{2}{\theta^2(t)}\left(\theta(t) + (Y - t)\theta'(t)\right).
\end{aligned}$$

From the definition of $z$, we see that $Y - t = \frac{(z-1)\theta(t)}{2}$. Then

$$\begin{aligned}
\frac{\mathrm{d}z}{\mathrm{d}t} &= -\frac{2}{\theta(t)}\left(1 + \left(\frac{z - 1}{2}\right)\theta'(t)\right) \\
&= -\frac{2}{\theta(t)} - \frac{(z - 1)\theta'(t)}{\theta(t)}.
\end{aligned} \tag{22}$$

Additionally, given a measure $\omega$ on [-1,1] and OP family $p_0(Y)$, $p_1(Y)$, $\ldots$ such that for all $i \neq j$,

$$\int_{-1}^{1} p_i(Y)p_j(Y)\omega(Y)\mathrm{d}Y = \delta_{i,j},$$

define

$$\omega(Y, t) = \frac{2}{\theta(t)}\omega(z) \text{ and } p_n(Y, t) = p_n(z).$$

Then we can adjust (17) to:

$$x_n(t) = \int_{t-\theta(t)}^{t} u(Y)p_n(z)\frac{2}{\theta(t)}\mathrm{d}Y. \tag{23}$$

The Leibniz integral rule states that

$$\frac{\partial}{\partial t}\int_{\alpha(t)}^{\beta(t)} h(t, Y)\mathrm{d}Y = \int_{\alpha(t)}^{\beta(t)} \frac{\partial}{\partial t}h(t, Y)\mathrm{d}Y - \alpha'(t)h(\alpha(t), t) + \beta'(t)h(t, t).$$

If we let $\alpha(t) = t - \theta(t)$ and $\beta(t) = t$, then applying the Leibniz rule to (23) we get:

$$\dot{x}_n(t) = \int_{t-\theta(t)}^{t} u(Y)\frac{\partial}{\partial t}\left(p_n(z)\frac{2}{\theta(t)}\right)dY - (1-\theta'(t))u(t-\theta(t))p_n(t-\theta(t),t)\frac{2}{\theta(t)}$$
$$+ u(t)p_n(t,t)\frac{2}{\theta(t)}$$
$$= -(1-\theta'(t))u(t-\theta(t))p_n(-1)\frac{2}{\theta(t)} + u(t)p_n(1)\frac{2}{\theta(t)}$$
$$+ \int_{t-\theta(t)}^{t} u(Y)\frac{dz}{dt}p_n'(z)\frac{2}{\theta(t)}dY - \frac{\theta'(t)}{\theta(t)}\int_{t-\theta(t)}^{t} u(Y)p_n(z)\frac{2}{\theta(t)}dY.$$

From (22), it follows that

$$\dot{x}_n(t) = -\frac{2(1-\theta'(t))u(t-\theta(t))p_n(-1)}{\theta(t)} + \frac{2\cdot u(t)p_n(1)}{\theta(t)} - \frac{2}{\theta(t)}\int_{t-\theta(t)}^{t} u(Y)p_n'(z)\frac{2}{\theta(t)}dY -$$
$$\frac{\theta'(t)}{\theta(t)}\int_{t-\theta(t)}^{t} u(Y)(z-1)p_n'(z)\frac{2}{\theta(t)}dY - \frac{\theta'(t)}{\theta(t)}\int_{t-\theta(t)}^{t} u(Y)p_n(z)\frac{2}{\theta(t)}dY. \qquad (24)$$

Because $\deg(p_n'(z)) \le n-1$ and $\deg((z-1)p_n'(z)) \le n$, they can be written as a linear combination of $\{p_i'\}_{i\le n}$. Let us define $\{\alpha_{n,j}\}$, $\{\beta_{n,j}\}$ such that

$$p_n'(z) = \sum_{j=0}^{n-1} \alpha_{n,j}p_j(z) \text{ and } (z-1)p_n'(z) = \sum_{j=0}^{n} \beta_{n,j}p_j(z). \qquad (25)$$

Then by using (25) in (24), we get:

$$\dot{x}_n(t) = -\frac{2(1-\theta'(t))u(t+\theta(t))p_n(-1)}{\theta(t)} + \frac{2\cdot u(t)p_n(1)}{\theta(t)}$$
$$- \frac{2}{\theta(t)}\sum_{j=0}^{n-1}\alpha_{n,j}\int_{t-\theta(t)}^{t} u(Y)p_j(z)\frac{2}{\theta(t)}dY$$
$$- \frac{\theta'(t)}{\theta(t)}\sum_{j=0}^{n}\beta_{n,j}\int_{t-\theta(t)}^{t} u(Y)p_j(z)\frac{2}{\theta(t)}dY - \frac{\theta'(t)}{\theta(t)}\int_{t-\theta(t)}^{t} u(Y)p_n(z)\frac{2}{\theta(t)}dY$$
$$= -\frac{2(1-\theta'(t))u(t+\theta(t))p_n(-1)}{\theta(t)} + \frac{2\cdot u(t)p_n(1)}{\theta(t)}$$
$$- \frac{2}{\theta(t)}\sum_{j=0}^{n-1}\alpha_{n,j}x_j(t) - \frac{\theta'(t)}{\theta(t)}\sum_{j=0}^{n}\beta_{n,j}x_j(t) - \frac{\theta'(t)}{\theta(t)}x_n(t).$$

Thus, in vector form we get
**Theorem 7.**

$$\dot{x}_n(t) = -\frac{1}{\theta(t)}\mathbf{A}_1(t)x(t) - \frac{2}{\theta(t)}(1-\theta'(t))u(t-\theta(t))\begin{bmatrix}\vdots \\ p_n(-1) \\ \vdots\end{bmatrix} + \frac{2}{\theta(t)}u(t)\begin{bmatrix}\vdots \\ p_n(1) \\ \vdots\end{bmatrix}$$

*where* $\mathbf{A}_1(t)[n,k] = \begin{cases} 2\alpha_{n,k} + \theta'(t)\beta_{n,k} & \text{if } k < n \\ \theta'(t)\beta_{n,n} + \theta'(t) & \text{if } k = n \quad \text{for } \alpha_{n,k}, \beta_{n,k} \text{ as defined in (25).} \\ 0 & \text{otherwise} \end{cases}$

**Corollary D.4.** *The matrix* $\mathbf{A}_1$ *in Theorem 7 can be re-written as*
$$\mathbf{A}_1 = 2\cdot\mathbf{A}' + \theta'(t)\cdot\mathbf{B}' + \theta'(t)\cdot\mathbf{I}. \qquad (26)$$

*In particular, both* $\mathbf{A}'$ *and* $\mathbf{B}'$ *both products of three matrices: two of which have recurrence width at most 2 and the third is the inverse of a matrix that has recurrence width 2.*

*Proof.* Eq. (26) follows from Theorem 7 and defining $\mathbf{A}'$ and $\mathbf{B}'$ to contain the $\alpha_{n,k}$ and $\beta_{n,k}$ coefficients. □

### D.2.3 Translated HiPPO (Sliding Windows)

The case when $\theta(t) = \theta$ for all $t$ represents a constant-size sliding window, which Gu et al. [24] denote as the "Translated HiPPO" case with instantiations such as HiPPO-LegT (Translated Legendre) and HiPPO-LagT (Translated Laguerre).

We now state a corollary of Theorem 7 for the case of $\theta(t) = \theta$ for all t.

**Corollary D.5.** *Let $\theta(t) = \theta$ for all t. Then*

$$\dot{x}_n(t) = -\frac{1}{\theta}\mathbf{A}_1 x(t) - \frac{2}{\theta}u(t-\theta)\begin{bmatrix} \vdots \\ p_n(-1) \\ \vdots \end{bmatrix} + \frac{2}{\theta}u(t)\begin{bmatrix} \vdots \\ p_n(1) \\ \vdots \end{bmatrix}.$$

*where* $\mathbf{A}_1[n,j] = \begin{cases} 2\alpha_{n,k} & \text{if } k < n \\ 0 & \text{otherwise} \end{cases}$.

Next, we use the approximation

$$u(x) \approx \sum_{k=0}^{N-1} x_k(t)p_k(z).$$

to handle the $u(t-\theta)$ term in Corollary D.5.

**Corollary D.6.** *Let $\theta(t) = \theta$ for all t. Then*

$$\dot{x}_n(t) \approx -\frac{1}{\theta}\mathbf{A}x(t) + \frac{2}{\theta}u(t)\begin{bmatrix} \vdots \\ p_n(1) \\ \vdots \end{bmatrix}$$

*where* $\mathbf{A} = \mathbf{A}_1 + 2\mathbf{A}_2$ *for $\mathbf{A}_1$ as defined in Corollary D.5 and $\mathbf{A}_2[n,k] = p_n(-1)p_k(-1)$.*

*Proof.* To approximate $u(t-\theta)$, we note that when $Y = t - \theta$, $z = -1$. Then

$$u(t-\theta) \approx \sum_{k=0}^{N-1} x_k(t)p_k(-1).$$

Then by Corollary D.5,

$$\dot{x}_n(t) \approx -\frac{1}{\theta}\mathbf{A}_1 x(t) - \frac{2}{\theta}\left(\sum_{k=0}^{N-1} x_k(t)p_k(-1)\right)\begin{bmatrix} \vdots \\ p_n(-1) \\ \vdots \end{bmatrix} + \frac{2}{\theta}u(t)\begin{bmatrix} \vdots \\ p_n(-1) \\ \vdots \end{bmatrix}. \quad (27)$$

Let us define a matrix, $\mathbf{A}_2 \in \mathbb{R}^{N \times N}$ matrix such that $\mathbf{A}_2[n,k] = p_n(-1)p_k(-1)$. Then the claim follows.

□

We now show that the special case of Corollary D.6 for Legendre matches the results from [24].

**Corollary D.7.** *Let* $p_n(z) = \left(\frac{2n+1}{2}\right)^{1/2} P_n(z)$ *where* $P_n(z)$ *are the Legendre polynomials. Then*

$$\dot{x}_n(t) \approx \frac{1}{\theta}\mathbf{A}x(t) + \frac{2}{\theta}\mathbf{b}u(t)$$

*where*

$$\mathbf{A}[n, k] = (2n + 1)^{\frac{1}{2}} (2k + 1)^{\frac{1}{2}} \begin{cases} 1 & \text{if } k \leq n \\ (-1)^{n-k} & \text{if } k \geq n \end{cases},$$

*and* $\mathbf{b}[n] = \left(\frac{2n+1}{2}\right)^{\frac{1}{2}}$.

*Proof.* From Corollary D.6,

$$\dot{x}_n(t) \approx -\frac{1}{\theta}\mathbf{A}x(t) + \frac{2}{\theta}u(t) \begin{bmatrix} \vdots \\ p_n(1) \\ \vdots \end{bmatrix}$$

where $\mathbf{A} = \mathbf{A}_1 + 2\mathbf{A}_2$ for $\mathbf{A}_1$ as defined in Corollary D.5 and $\mathbf{A}_2[n, k] = p_n(-1)p_k(-1)$.
It is known from (7.21.1) and in [53] that

$$p_n(-1) = \left(\frac{2n + 1}{2}\right)^{\frac{1}{2}} P_n(-1) \text{ and } P_n(-1) = (-1)^n. \tag{28}$$

Further,

$$p_n(1) = \left(\frac{2n + 1}{2}\right)^{\frac{1}{2}} P_n(1) \text{ and } P_n(1) = 1. \tag{29}$$

Then $\mathbf{b}[n] = \left(\frac{2n+1}{2}\right)^{\frac{1}{2}}$ follows from Corollary D.6 and (29).
From the following recurrence relations [1, Chapter 12]:

$$(2n + 1)P_n(z) = P'_{n+1}(z) + P'_{n-1}(z)$$

implies that

$$P'_{n+1}(z) = (2n + 1)P_n(z) + (2n + 1)P_{n-2}(z) + \cdots +,$$

which in turn implies

$$P'_n = (2n - 1)P_{n-1}(z) + (2n - 5)P_{n-3}(z) + \dots.$$

Then

$$p'_n(z) = \left(\frac{2n + 1}{2}\right)^{\frac{1}{2}} \cdot P'_n(z)$$

$$= \left(\frac{2n + 1}{2}\right)^{\frac{1}{2}} \left((2n - 1)\left(\frac{2}{2n - 1}\right)^{\frac{1}{2}} P_{n-1}(z) + (2n - 5)\left(\frac{2}{2n - 5}\right)^{\frac{1}{2}} P_{n-3}(z) + \dots\right)$$

$$= (2n + 1)^{\frac{1}{2}} \left((2n - 1)^{\frac{1}{2}} P_{n-1}(z) + (2n - 5)^{\frac{1}{2}} P_{n-3}(z) + \dots\right).$$

Thus, we have

$$\alpha_{n,k} = \begin{cases} (2n + 1)^{\frac{1}{2}}(2k + 1)^{\frac{1}{2}} \text{ if } k < n \text{ and } n - k \text{ is odd,} \\ 0 \text{ is otherwise.} \end{cases},$$

Recalling that $\mathbf{A}_1[n,k] = 2\alpha_{n,k}$.

We note that from (28), $\mathbf{A}_2[n,k] = \left(\frac{2n+1}{2}\right)^{\frac{1}{2}} \left(\frac{2k+1}{2}\right)^{\frac{1}{2}} (-1)^n(-1)^k = \frac{(2n+1)^{\frac{1}{2}}(2k+1)^{\frac{1}{2}}}{2}(-1)^{n-k}$.
Recalling $\mathbf{A} = \mathbf{A}_1 + 2\mathbf{A}_2$, we get:

$$\mathbf{A}[n,k] = (2n+1)^{\frac{1}{2}}(2k+1)^{\frac{1}{2}} \begin{cases} 2 + (-1)^{n-k} & \text{if } k < n \text{ and } n-k \text{ is odd} \\ 0 + (-1)^{n-k} & \text{if } k < n \text{ and } n-k \text{ is even} \\ (-1)^{n-k} & \text{if } k \geq n \end{cases}.$$

Note that the above is the same as:

$$\mathbf{A}[n,k] = (2n+1)^{\frac{1}{2}}(2k+1)^{\frac{1}{2}} \begin{cases} 1 & \text{if } k \leq n \\ (-1)^{n-k} & \text{if } k \geq n \end{cases},$$

which completes our claim.

$\square$

### D.2.4 Scaled HiPPO: Recovering HiPPO-LegS

We now use Theorem 7 to recover the HiPPO-LegS instantiation for the "Scaled Legendre" measure, the main method from Gu et al. [24].

**Corollary D.8.** *Let* $p_n(z) = \left(\frac{2n+1}{2}\right)^{1/2} P_n(z)$ *where* $P_n(z)$ *are the Legendre polynomials and let* $\delta(t) = t$ *for all t. Then*

$$\dot{x}_n(t) = \frac{1}{t}\mathbf{A}x(t) + \frac{2}{t}\mathbf{b}u(t)$$

*where*

$$\mathbf{A}[n,k] = \begin{cases} (2n+1)^{\frac{1}{2}}(2k+1)^{\frac{1}{2}} & \text{if } k < n \\ n+1 & \text{if } k = n \\ 0 & \text{if } k > n \end{cases},$$

*and* $\mathbf{b}[n] = \left(\frac{2n+1}{2}\right)^{\frac{1}{2}}$.

*Proof.* Let $\theta(t) = t$. By Theorem 7 and noting that $\theta(t) = 1$, we get:

$$\dot{x}_n(t) = -\frac{1}{t}\mathbf{A}_1 x(t) + \frac{2}{t}u(t)\begin{bmatrix} \vdots \\ p_n(1) \\ \vdots \end{bmatrix}$$

where

$$\mathbf{A}_1(t)[n,k] = \begin{cases} 2\alpha_{n,k} + \beta_{n,k} & \text{if } k < n \\ \beta_{n,n} + 1 & \text{if } k = n \\ 0 & \text{otherwise} \end{cases} \tag{30}$$

for $\alpha_{n,k}, \beta_{n,k}$ as defined in (25).

Using the same arguments as in the proof of Corollary D.7, $\mathbf{b}[n] = \left(\frac{2n+1}{2}\right)^{\frac{1}{2}}$ follows from Corollary D.6 and (29). Also using similar arguments as the proof of Corollary D.7, we have

$$\alpha_{n,k} = \begin{cases} (2n+1)^{\frac{1}{2}}(2k+1)^{\frac{1}{2}} & \text{if } k < n \text{ and } n-k \text{ is odd,} \\ 0 & \text{is otherwise.} \end{cases}.$$

From (8) in [24], we know that
$$(z+1)P_n'(z) = nP_n(z) + (2n+1)P_{n-1}(z) + (2n-3)P_{n-2}(z) + \ldots.$$

Including the normalization constant $(2n+1)^{\frac{1}{2}}$, we note that $(z-1)p'_n(z) = (z+1)p'_n(z) - 2p'_n(z)$. Then we get

$$(z+1)p'_n(z) = np_n(z) - (2n+1)^{\frac{1}{2}}(2n-1)^{\frac{1}{2}}p_{n-1}(z) + (2n+1)^{\frac{1}{2}}(2n-3)^{\frac{1}{2}}p_{n-2}(z) - \dots .$$

In other words,

$$\beta_{n,k} = \begin{cases} -(2n+1)^{\frac{1}{2}}(2k+1)^{\frac{1}{2}} & \text{if } k < n \text{ and } n-k \text{ is odd,} \\ (2n+1)^{\frac{1}{2}}(2k+1)^{\frac{1}{2}} & \text{if } k < n \text{ and } n-k \text{ is even} \\ n & \text{if } n = k \\ 0 & \text{otherwise.} \end{cases}$$

Recalling that the definition for $\mathbf{A}_1$ from (30), we get:

$$\mathbf{A}[n,k] = \begin{cases} (2n+1)^{\frac{1}{2}}(2k+1)^{\frac{1}{2}} & \text{if } k < n \\ n+1 & \text{if } k = n \\ 0 & \text{if } k > n \end{cases},$$

which completes our claim. $\qquad\square$

## D.3 Proof of Corollary 4.1: HiPPO for Classical Orthogonal Polynomials

This section proves Corollary 4.1, showing that the HiPPO matrices for measures corresponding to classical families of orthogonal polynomials [11] are quasiseparable. We define quasi-separability in Appendix D.3.1. Theorem 8 proves the claimed result for Jacobi polynomials and Lemma D.11 proves the claimed result for Laguerre polynomials.

We note that there is a third family of classical OPs, the Hermite polynomials [11], which have a two-sided infinite measure. However, since HiPPO is about continuous-time memorization of a function's *history*, it requires a one-sided measure and therefore the Hermite polynomials are not appropriate.

### D.3.1 Quasiseparable Matrices

**Definition 4** (from [19]). *A matrix $\mathbf{R} \in \mathbb{R}^{N \times N}$ is $(p,q)$-quasiseparable if*

- *Every matrix contained strictly above the diagonal has rank at most $p$.*

- *Every matrix contained strictly below the diagonal has rank at most $q$.*

*A $(q,q)$-quasiseparable matrix is called $q$-quasiseparable.*

We are interested in showing the $\mathbf{A}$ matrices for a broad class of OPs in Corollary D.6 are $O(1)$-quasiseperable. We now state some properties of $q$-quasiseparable matrices:

**Lemma D.9.** *Let $\mathbf{Q}$ be $q$-quasiseparable. Then:*

(i) *For any $q'$-quasiseparable matrix $\mathbf{Q}' \in \mathbb{R}^{N \times N}$, $\mathbf{Q} \pm \mathbf{Q}'$ is $(q+q')$-quasiseparable.*

(ii) *For any $\mathbf{E} \in \mathbb{R}^{N \times N}$, $\mathbf{E}$ is $r$-quasiseparable where $r = \text{rank}(\mathbf{E})$.*

(iii) *For any two diagonal matrices $\mathbf{D}_1$, $\mathbf{D}_2 \in \mathbb{R}^{N \times N}$, $\mathbf{D}_1 \mathbf{Q} \mathbf{D}_2$ is $q$-quasiseparable.*

*Proof.* We argue each point separately:

(i) Any submatrix contained strictly below or above the diagonal in $\mathbf{Q}$ has rank $\leq q$ and its corresponding submatrix in $\mathbf{Q}'$ also has rank $\leq q'$. This implies that the corresponding submatrix in $\mathbf{Q} \pm \mathbf{Q}'$ has rank $\leq q + q'$. Therefore $\mathbf{Q} \pm \mathbf{Q}'$ is $(q+q')$-quasiseparable.

(ii) Let the $r = \text{rank}(\mathbf{E})$. Thus any submatrix in $\mathbf{E}$ has rank $\leq r$. Then $\mathbf{E}$ is $r$-quasiseparable.

(iii) Multiplication by diagonal matrices only scales the rows and columns, leaving the rank of each submatrix unchanged.

$\square$

### D.3.2 Jacobi Polynomials

The Jacobi polynomial of degree $n$ with parameters $\alpha, \beta > -1$ will be denoted $J_n^{\alpha,\beta}(z)$. The Jacobi polynomials are orthogonal with respect to measure $\omega(z) = (1-z)^\alpha(1+z)^\beta$. In particular, it is known from (eq. (4.3.3) from [53]) that

$$\int_{-1}^1 J_n^{\alpha,\beta}(z) J_m^{\alpha,\beta}(z) \omega(z) \mathrm{d}z = \frac{2^{\alpha+\beta+1}}{2n+\alpha+\beta+1} \cdot \frac{\Gamma(n+\alpha+1)\Gamma(n+\beta+1)}{\Gamma(n+\alpha+\beta+1)n!} \delta_{n,m},$$

where $\Gamma(\cdot)$ is the gamma function. Let

$$\lambda_n^{\alpha,\beta} = \left(\frac{2^{\alpha+\beta+1}}{2n+\alpha+\beta+1} \cdot \frac{\Gamma(n+\alpha+1)\Gamma(n+\beta+1)}{\Gamma(n+\alpha+\beta+1)n!}\right)^{\frac{1}{2}}$$

be our normalization constant. We note that the normalized Jacobi polynomials

$$p_n^{\alpha,\beta}(z) = \frac{J_n^{\alpha,\beta}(z)}{\lambda_n^{\alpha,\beta}} \tag{31}$$

form an orthonormal OP family.

We now discuss some useful properties of Jacobi polynomials. It is known that ([50], eq. (3.100)):

$$J_n^{\alpha,\beta}(z) = \frac{1}{n+\alpha+\beta}\left[(n+\beta)J_n^{\alpha,\beta-1}(z) + (n+\alpha)J_n^{\alpha-1,\beta}(z)\right]. \tag{32}$$

From (4.21.7) in [53], it is known that the derivative of $J_n^{\alpha,\beta}(z)$ is proportional to $J_{n-1}^{\alpha+1,\beta+1}(z)$:

$$\frac{\partial}{\partial z} J_n^{\alpha,\beta}(z) = \frac{1}{2}(n+\alpha+\beta+1)J_{n-1}^{\alpha+1,\beta+1}(z). \tag{33}$$

From (32) and (33), it follows that

$$\frac{\partial}{\partial z} J_n^{\alpha,\beta}(z) = \frac{1}{2} \cdot \left((n+\beta)J_{n-1}^{\alpha+1,\beta}(z) + (n+\alpha)J_{n-1}^{\alpha,\beta+1}(z)\right). \tag{34}$$

Additionally, the Jacobi polynomials $J_{n-1}^{\alpha+1,\beta}(z)$ and $J_{n-1}^{\alpha,\beta+1}(z)$ can be written as sums of $J_{n-1}^{\alpha,\beta}(z)$ polynomials. In particular from [50] (3.112) and (3.115),

$$J_{n-1}^{\alpha+1,\beta}(z) = \frac{\Gamma(n+\beta)}{\Gamma(n+\alpha+\beta+1)} \cdot \sum_{k=0}^{n-1} \frac{(2k+\alpha+\beta+1)\Gamma(k+\alpha+\beta+1)}{\Gamma(k+\beta+1)} J_k^{\alpha,\beta}(z), \tag{35}$$

and

$$J_{n-1}^{\alpha,\beta+1}(z) = \frac{\Gamma(n+\alpha)}{\Gamma(n+\alpha+\beta+1)} \cdot \sum_{k=0}^{n-1}(-1)^{n-k-1}\frac{(2k+\alpha+\beta+1)\Gamma(k+\alpha+\beta+1)}{\Gamma(k+\alpha+1)} J_k^{\alpha,\beta}(z). \tag{36}$$

Using (35) and (36) in (34) allows us to write $\frac{\partial}{\partial z} J_n^{\alpha,\beta}(z)$ as a sum of $\left\{J_k^{\alpha,\beta}(z)\right\}_{k \leq n}$ as follows:

$$\frac{\partial}{\partial z} J_n^{\alpha,\beta}(z) = \frac{n+\beta}{2}\left(\frac{\Gamma(n+\beta)}{\Gamma(n+\alpha+\beta+1)}\sum_{k=0}^{n-1}\frac{(2k+\alpha+\beta+1)\Gamma(k+\alpha+\beta+1)}{\Gamma(k+\beta+1)}J_k^{\alpha,\beta}(z)\right)$$
$$-\frac{n+\alpha}{2}\left(\frac{\Gamma(n+\alpha)}{\Gamma(n+\alpha+\beta+1)}\sum_{k=0}^{n-1}(-1)^{n-k}\frac{(2k+\alpha+\beta+1)\Gamma(k+\alpha+\beta+1)}{\Gamma(k+\alpha+1)}J_k^{\alpha,\beta}(z)\right).$$

(37)

We use these properties to write $\frac{\partial}{\partial z}p_n^{\alpha,\beta}(z)$ as a sum of $\left\{p_k^{\alpha,\beta}(z)\right\}_{k\le n}$:

**Corollary D.10.** *Let $p_n^{\alpha,\beta}(z)$ and $\lambda_n^{\alpha,\beta}$ be as defined in (31).*

*Then*

$$\frac{\partial}{\partial z}\lambda_n^{\alpha,\beta}p_n^{\alpha,\beta}(z) = \frac{(n+\beta)}{2}\cdot$$
$$\left(\frac{\Gamma(n+\beta)}{\Gamma(n+\alpha+\beta+1)}\sum_{k=0}^{n-1}\frac{(2k+\alpha+\beta+1)\Gamma(k+\alpha+\beta+1)}{\Gamma(k+\beta+1)}\lambda_k^{\alpha,\beta}p_k^{\alpha,\beta}(z)\right)$$
$$-\frac{(n+\alpha)}{2}\cdot$$
$$\left(\frac{\Gamma(n+\alpha)}{\Gamma(n+\alpha+\beta+1)}\sum_{k=0}^{n-1}(-1)^{n-k}\frac{(2k+\alpha+\beta+1)\Gamma(k+\alpha+\beta+1)}{\Gamma(k+\beta+1)}\lambda_k^{\alpha,\beta}p_k^{\alpha,\beta}(z)\right)$$

*Proof.* Recall that $J_n^{\alpha,\beta}(z) = \lambda_n^{\alpha,\beta}p_n^{\alpha,\beta}$. Then the claim follows from (37).

$\square$

### D.3.3 HiPPO for Jacobi Polynomials

**Theorem 8.** *Let $p_n^{\alpha,\beta}(z)$ be defined as in (31) and $\omega(z) = (1-z)^\alpha(1+z)^\beta$. Then*

$$\dot{x}_n(t) \approx -\frac{1}{\theta}\mathbf{A}x(t) + \frac{2}{\theta}\mathbf{b}u(t)$$

*where $\mathbf{A}$ is 3-quasiseperable.*

*Proof.* From Corollary D.6,

$$\dot{x}_n(t) \approx -\frac{1}{\theta}\mathbf{A}x(t) + \frac{2}{\theta}u(t)\begin{bmatrix}\vdots\\p_n^{\alpha,\beta}(1)\\\vdots\end{bmatrix}$$

where $\mathbf{A} = \mathbf{A}_1 + 2\mathbf{A}_2$ for $\mathbf{A}_1$ as defined in Corollary D.5 and $\mathbf{A}_2[n,k] = p_n^{\alpha,\beta}(-1)p_n^{\alpha,\beta}(-1)$. From Corollary D.10, we observe that

$$\mathbf{A}_1[n,k] = 2\cdot\begin{cases}\frac{(n+\beta)}{2\,\lambda_n^{\alpha,\beta}}\cdot\frac{\Gamma(n+\beta)}{\Gamma(n+\alpha+\beta+1)}\cdot\left(\frac{(2k+\alpha+\beta+1)\Gamma(k+\alpha+\beta+1)}{\Gamma(k+\beta+1)}\lambda_k^{\alpha,\beta}\right)-\\\frac{(n+\alpha)}{2\,\lambda_n^{\alpha,\beta}}\cdot\frac{\Gamma(n+\alpha)}{\Gamma(n+\alpha+\beta+1)}\cdot\left((-1)^{n-k}\frac{(2k+\alpha+\beta+1)\Gamma(k+\alpha+\beta+1)}{\Gamma(k+\alpha+1)}\lambda_k^{\alpha,\beta}\right) & \text{if } k<n\\0 & \text{otherwise}\end{cases}\cdot$$

(38)

Then we note that,
$$\mathbf{A}_1 = \mathbf{D}_{11}\mathbf{Q}_1\mathbf{D}_{12} - \mathbf{D}_{21}\mathbf{Q}_1\mathbf{D}_{22},$$

(39)

where $\mathbf{D}_{11}, \mathbf{D}_{12}, \mathbf{D}_{21}, \mathbf{D}_{22}$ are the diagonal matrices such that

$$\mathbf{D}_{11}[n, n] = \frac{1}{\lambda_n^{\alpha,\beta}} \cdot \frac{\Gamma(n+\beta+1)}{\Gamma(n+\alpha+\beta+1)},$$

$$\mathbf{D}_{12}[k, k] = \frac{(2k+\alpha+\beta+1)\Gamma(k+\alpha+\beta+1)}{\Gamma(k+\beta+1)}\lambda_k^{\alpha,\beta},$$

$$\mathbf{D}_{21}[n, n] = (-1)^n \cdot \frac{(1}{\lambda_n^{\alpha,\beta}} \cdot \frac{\Gamma(n+\alpha+1)}{\Gamma(n+\alpha+\beta+1)}$$

$$\mathbf{D}_{22}[k, k] = (-1)^k \cdot \frac{(2k+\alpha+\beta+1)\Gamma(k+\alpha+\beta+1)}{\Gamma(k+\alpha+1)}\lambda_k^{\alpha,\beta},$$

and

$$\mathbf{Q}_1[n, k] = \begin{cases} 1 & \text{if } k < n \\ 0 & \text{otherwise.} \end{cases}$$

(39) makes use of the fact that $(-1)^{n+k} = (-1)^{n-k}$ along with the definitions above.

Any submatrix of $\mathbf{Q}_1$ below the diagonal contains all 1s, and submatrix of $\mathbf{Q}_1$ above the diagonal contains all 0s. Then any submatrix above or below the diagonal has rank 1. Therefore $\mathbf{Q}_1$ is 1-quasiseparable. Since $\mathbf{Q}_1$ is 1-quasiseparable and $\mathbf{D}_{11}, \mathbf{D}_{12}, \mathbf{D}_{21}, \mathbf{D}_{22}$ are all diagonal matrices, part (iii) of Lemma D.9 implies that the matrices $\mathbf{D}_{11}\mathbf{Q}_1\mathbf{D}_{12}$ and $\mathbf{D}_{21}\mathbf{Q}_1\mathbf{D}_{22}$ are both 1-quasiseparable. Therefore part (i) of Lemma D.9 implies that $\mathbf{A}_1$ is 2-quasiseparable.

From (4.1.1) and (4.1.4) in [53], it is known that

$$p_n^{\alpha,\beta}(1) = \frac{1}{\lambda_n^{\alpha,\beta}}\binom{n+\alpha}{n} \text{ and } p_n^{\alpha,\beta}(1) = \frac{(-1)^n}{\lambda_n^{\alpha,\beta}}\binom{n+\beta}{n}$$

where

$$\binom{z}{n} = \begin{cases} \frac{\Gamma(z+1)}{\Gamma(n+1)\Gamma(z-n+1)} & \text{if } n \geq 0 \\ 0 & \text{if } n < 0 \end{cases}.$$

Then $\mathbf{A}_2$ can be written $\mathbf{D}_3\mathbf{Q}_2\mathbf{D}_4$ where $\mathbf{D}_3, \mathbf{D}_4$ are the diagonal matrices such that

$$\mathbf{D}_3[n, n] = \frac{(-1)^n}{\lambda_n^{\alpha,\beta}}\binom{n+\beta}{n}, \mathbf{D}_4[k, k] = \frac{(-1)^k}{\lambda_k^{\alpha,\beta}}\binom{k+\beta}{k},$$

where $\mathbf{Q}_2[n, k] = 1$ for all $0 \leq n, k < N$. $\mathbf{Q}_2$ has rank 1, and $\mathbf{D}_3, \mathbf{D}_4$ are diagonal matrices. Hence by part (ii) and (iii) Lemma D.9, $\mathbf{A}_2$ is 1-quasiseparable.

Since $\mathbf{A}_1$ is 2-quasiseparable and $\mathbf{A}_2$ is 1-quasiseparable, part (i) of Lemma D.9 implies that $\mathbf{A} = \mathbf{A}_1 + 2\mathbf{A}_2$ is 3-quasiseparable and the claim follows. $\square$

### D.3.4   HiPPO-LagT

The Laguerre polynomial of degree $n$ with parameters $\alpha > -1$ will be denoted $L_n^\alpha(z)$. The Laguerre polynomials are orthogonal with respect to measure $z^\alpha e^{-z}$. In particular, from (5.1.1) in [53] we know that

$$\int_{-1}^\infty L_n^\alpha(z) L_m^\alpha(z) z^\alpha e^{-z}\mathrm{d}z = \frac{\Gamma(n+\alpha+1)!}{\Gamma(n+1)}\delta_{n,m}.$$

Let $\lambda_n = \left(\frac{\Gamma(n+1)}{\Gamma(n+\alpha+1)}\right)^{\frac{1}{2}}$ be our normalization constant. We note that the normalized Laguerre polynomials

$$p_n(z) = \lambda_n L_n^\alpha(t-Y) \tag{40}$$

form an orthonormal OP family with respect to measure $\omega = (t - Y)^\alpha e^{-(t-Y)} \mathbb{1}_{(-\infty, t)}$ for a fixed $\alpha$ and tilting $\chi = (t - Y)^\alpha \exp\left(-\frac{1-\beta}{2}(t - Y)\right) \mathbb{1}_{(-\infty, t)}$ for a fixed $\beta$.

We use the following result from [24]:

**Theorem 9.** *Let $p_n(z)$ be defined as in* $(40)$. *Then*

$$\dot{x}_n(t) = -\mathbf{A}x(t) + \mathbf{b}u(t)$$

*where*

$$\mathbf{A}[n, k] = \begin{cases} \frac{1+\beta}{2} & \text{if } k = n \\ 1 & \text{if } k < n \\ 0 & \text{otherwise} \end{cases},$$

$$\mathbf{b}[n] = \lambda_n \binom{n + \alpha}{n},$$

We now show that $\mathbf{A}$ as defined in Theorem 9 is 1-quasiseperable.

**Lemma D.11.** *Let $\mathbf{A}$ be defined as in Theorem 9. Then $\mathbf{A}$ is 1-quasiseperable.*

*Proof.* From Theorem 9, we know that

$$\mathbf{A}[n, k] = \begin{cases} \frac{1+\beta}{2} & \text{if } k = n \\ 1 & \text{if } k < n \\ 0 & \text{otherwise} \end{cases},$$

$$\mathbf{b}[n] = \lambda_n \binom{n + \alpha}{n}.$$

Below the diagonal, all entries $\mathbf{A}[n, k] = 1$. Then any submatrix below the diagonal has rank 1. Similarly, above the diagonal, all entries $\mathbf{A}[n, k] = 0$. Then any submatrix above the diagonal also has rank 1. Then by Definition 4, the claim follows.

$\square$

# E    LSSL Algorithms

- Appendix E.1 proves Theorem 2, providing an algorithm to compute the Krylov function efficiently for LSSLs.

- Appendix E.2 shows a further simplification of Corollary 4.1, presenting an even simpler class of structured matrices that we use in our implementation of LSSL.

- Appendix E.3 provides technical details of the implementation of LSSL, in particular for computing the MVM black box (multiplication by $\overline{A}$) and for computing gradients during backpropagation.

## E.1    Proof of Theorem 2

This section addresses the computational aspects of the LSSL. In particular, we prove Theorem 2 for the computational speed of computing the Krylov function (7) for quasiseparable matrices $A$, by providing a concrete algorithm in Appendix E.1.1.

We restate the Krylov function (7) here for convenience. Recall that $L$ is the length of the input sequence and $N$ is the order of the LSSL internate state, e.g. $A \in \mathbb{R}^{N \times N}$.

$$\mathcal{K}_L(A, B, C) = \left(CA^i B\right)_{i \in [L]} \in \mathbb{R}^L = (CB, CAB, \dots, CA^{L-1}B)$$

**Remark E.1.** *We call* (7) *the* Krylov function *following the notation of [17], since it can be written $\mathcal{K}(A, B)^T C$ where $\mathcal{K}(A, B)$ is the Krylov matrix defined in* (8). *Alternative naming suggestions are welcome.*

### E.1.1 The Algorithm

We follow the similar problem of [17, Lemma 6.6] but track the dependence on $L$ and the log factors more precisely, and optimize it in the case of stronger structure than quasiseparability, which holds in our setting (particularly Theorem 11).

The first step is to observe that the Krylov function $\mathcal{K}_L(A, B, C)$ is actually the coefficient vector of $C(I - Ax)^{-1}B \pmod{x^L}$ as a polynomial in $x$. (Note that $Ax$ means simply multiplying every entry in $A$ by a scalar variable $x$.) This follows from expanding the power series $(I - Ax)^{-1} = I + Ax + A^2x^2 + \dots$. Thus we first compute $C(I - Ax)^{-1}B$, which is a rational function of degree at most $N$ in the numerator and denominator (which can be seen by the standard adjoint formula for the matrix inverse).

The second step is simply inverting the denominator of this rational function $\pmod{x^L}$ and multiplying by the numerator, both of which are operations that need $L \log(L)$ time by standard results for polynomial arithmetic [51].

For the remainder of this section, we focus on computing the first part. We make two notational changes: First, we transpose $\mathbf{C}$ to make it have the same shape as $\mathbf{B}$. We consider the more general setting where $\mathbf{B}$ and $\mathbf{C}$ have multiple columns; this can be viewed as handling a "batch" problem with several queries for $\mathbf{B}, \mathbf{C}$ at the same time.

**Lemma E.2.** *Let $\mathbf{A}$ be a $q$-quasiseparable matrix. Then*

$$\mathbf{C}^T(\mathbf{I} - \mathbf{A}x)^{-1}\mathbf{B} \qquad \text{where } \mathbf{A} \in \mathbb{R}^{N \times N} \qquad \mathbf{B}, \mathbf{C} \in \mathbb{R}^{N \times k}$$

*is a $k \times k$ matrix of rational functions of degree at most $N$, which can be computed in $O(q^3 \log^4 N)$ operations.*

The main idea is that quasiseparable matrices are recursively "self-similar", in that the principal submatrices are also quasiseparable, which leads to a divide-and-conquer algorithm. In particular, divide $\mathbf{A} = \begin{bmatrix} \mathbf{A}_{00} & \mathbf{A}_{01} \\ \mathbf{A}_{10} & \mathbf{A}_{11} \end{bmatrix}$ into quadrants. Then by Definition 4, $\mathbf{A}_{00}, \mathbf{A}_{11}$ are both $q$-quasiseparable and $\mathbf{A}_{01}, \mathbf{A}_{10}$ are rank $q$. Therefore the strategy is to view $\mathbf{I} - \mathbf{A}x$ as a low-rank perturbation of smaller quasiseparable matrices and reduce the problem to a simpler one.

**Proposition 10** (Binomial Inverse Theorem or Woodbury matrix identity [23, 60])**.** *Over a commutative ring $\mathcal{R}$, let $\mathbf{A} \in \mathcal{R}^{N \times N}$ and $\mathbf{U}, \mathbf{V} \in \mathcal{R}^{N \times p}$. Suppose $\mathbf{A}$ and $\mathbf{A} + \mathbf{U}\mathbf{V}^T$ are invertible. Then $\mathbf{I}_p + \mathbf{V}^T\mathbf{A}^{-1}\mathbf{U} \in \mathcal{R}^{p \times p}$ is invertible and*

$$(\mathbf{A} + \mathbf{U}\mathbf{V}^T)^{-1} = \mathbf{A}^{-1} - \mathbf{A}^{-1}\mathbf{U}(\mathbf{I}_p + \mathbf{V}^T\mathbf{A}^{-1}\mathbf{U})^{-1}\mathbf{V}^T\mathbf{A}^{-1}$$

For our purposes, $\mathcal{R}$ will be the ring of rational functions over $\mathbb{R}$.

*Proof of Lemma E.2.* Since $\mathbf{A}$ is $q$-quasiseparable, we can write $\mathbf{A}_{10} = \mathbf{U}_L\mathbf{V}_L^T$ and $\mathbf{A}_{01} = \mathbf{U}_U\mathbf{V}_U^T$ where $\mathbf{U}_{\cdot}, \mathbf{V}_{\cdot} \in \mathbb{F}^{N \times q}$. Notice that we can write $\mathbf{I} - \mathbf{A}x$ as

$$\mathbf{I} - \mathbf{A}x = \begin{bmatrix} \mathbf{I} - \mathbf{A}_{00}x & \mathbf{0} \\ \mathbf{0} & \mathbf{I} - \mathbf{A}_{11}x \end{bmatrix} + \begin{bmatrix} \mathbf{0} & \mathbf{U}_U \\ \mathbf{U}_L & \mathbf{0} \end{bmatrix}\begin{bmatrix} \mathbf{V}_L & \mathbf{0} \\ \mathbf{0} & \mathbf{V}_U \end{bmatrix}^T x.$$

Suppose we know the expansions of each of

$$\mathbf{M}_1 \in \mathcal{R}^{k \times k} = \mathbf{C}^T\begin{bmatrix} \mathbf{I} - \mathbf{A}_{00}x & \mathbf{0} \\ \mathbf{0} & \mathbf{I} - \mathbf{A}_{11}x \end{bmatrix}^{-1}\mathbf{B} \tag{41}$$

$$\mathbf{M}_2 \in \mathcal{R}^{k \times 2q} = \mathbf{C}^T\begin{bmatrix} \mathbf{I} - \mathbf{A}_{00}x & \mathbf{0} \\ \mathbf{0} & \mathbf{I} - \mathbf{A}_{11}x \end{bmatrix}^{-1}\begin{bmatrix} \mathbf{0} & \mathbf{U}_U \\ \mathbf{U}_L & \mathbf{0} \end{bmatrix} \tag{42}$$

$$\mathbf{M}_3 \in \mathcal{R}^{2q \times 2q} = \begin{bmatrix} \mathbf{V}_L & \mathbf{0} \\ \mathbf{0} & \mathbf{V}_U \end{bmatrix}^T\begin{bmatrix} \mathbf{I} - \mathbf{A}_{00}x & \mathbf{0} \\ \mathbf{0} & \mathbf{I} - \mathbf{A}_{11}x \end{bmatrix}^{-1}\begin{bmatrix} \mathbf{0} & \mathbf{U}_U \\ \mathbf{U}_L & \mathbf{0} \end{bmatrix} \tag{43}$$

$$\mathbf{M}_4 \in \mathcal{R}^{2q \times k} = \begin{bmatrix} \mathbf{V}_L & \mathbf{0} \\ \mathbf{0} & \mathbf{V}_U \end{bmatrix}^T\begin{bmatrix} \mathbf{I} - \mathbf{A}_{00}x & \mathbf{0} \\ \mathbf{0} & \mathbf{I} - \mathbf{A}_{11}x \end{bmatrix}^{-1}\mathbf{B}. \tag{44}$$

By Proposition 10, the desired answer is

$$\mathbf{C}^T(X - \mathbf{A})^{-1}\mathbf{B} = \mathbf{M}_1 - \mathbf{M}_2(\mathbf{I}_{2q} + \mathbf{M}_3)^{-1}\mathbf{M}_4.$$

Then the final result can be computed by inverting $\mathbf{I}_{2t} + \mathbf{M}_3$ ($O(q^3 N \log(N))$ operations), multiplying by $\mathbf{M}_2, \mathbf{M}_4$ ($O((kq^2 + k^2q)N \log(N))$ operations), and subtracting from $\mathbf{M}_1$ ($O(k^2 N \log(N))$ operations). This is a total of $O((q^3 + kq^2 + k^2q)N \log(N))$ operations. Note that when $k = O(q \log N)$, this becomes $O(q^3 N \log^3 N)$; we will use this in the analysis shortly.

To compute $\mathbf{M}_1, \mathbf{M}_2, \mathbf{M}_3, \mathbf{M}_4$, it suffices to compute the following:

$$
\begin{aligned}
&\mathbf{C}_1^T(\mathbf{I} - \mathbf{A}_{00}x)^{-1}\mathbf{B}_0 && \mathbf{C}_1^T(\mathbf{I} - \mathbf{A}_{11}x)^{-1}\mathbf{B}_1 \\
&\mathbf{C}_0^T(\mathbf{I} - \mathbf{A}_{00}x)^{-1}\mathbf{U}_U && \mathbf{C}_1^T(\mathbf{I} - \mathbf{A}_{11}x)^{-1}\mathbf{U}_L \\
&\mathbf{V}_L^T(\mathbf{I} - \mathbf{A}_{00}x)^{-1}\mathbf{U}_U && \mathbf{V}_U^T(\mathbf{I} - \mathbf{A}_{11}x)^{-1}\mathbf{U}_L \\
&\mathbf{V}_L^T(\mathbf{I} - \mathbf{A}_{00}x)^{-1}\mathbf{B}_0 && \mathbf{V}_U^T(\mathbf{I} - \mathbf{A}_{11}x)^{-1}\mathbf{B}_1.
\end{aligned}
\tag{45}
$$

But to compute those, it suffices to compute the following $(k + t) \times (k + t)$ matrices:

$$
\begin{aligned}
&[\mathbf{C}_0 \quad \mathbf{V}_L]^T (\mathbf{I} - \mathbf{A}_{00}x)^{-1} [\mathbf{B}_0 \quad \mathbf{U}_U] \\
&[\mathbf{C}_1 \quad \mathbf{V}_U]^T (\mathbf{I} - \mathbf{A}_{11}x)^{-1} [\mathbf{B}_1 \quad \mathbf{U}_L]
\end{aligned}
\tag{46}
$$

Since $\mathbf{A}_{00}$ and $\mathbf{A}_{11}$ have the same form as $\mathbf{A}$, this is two recursive calls of half the size. Notice that the size of the other input (dimensions of $\mathbf{B}, \mathbf{C}$) is growing, but when the initial input is $k = 1$, it never exceeds $1 + q \log N$ (since they increase by $q$ every time we go down a level). Earlier, we noticed that when $k = O(q \log N)$, the reduction step has complexity $O(q^3 N \log^3(N))$ for any recursive call. The recursion adds an additional $\log N$ multiplicative factor on top of this. $\qquad \square$

**Corollary E.3.** *Suppose that $\mathbf{A}$ is semiseparable instead of quasiseparable, and suppose $q$ is a small constant. Then the cost of Lemma E.2 is $O(N \log^2(N))$ operations.*

This follows from the fact that in the recursion (45) and (46), the $\mathbf{U}, \mathbf{V}$ matrices do not have to be appended if they already exist in $\mathbf{B}, \mathbf{C}$. For intuition, this happens in the case when $\mathbf{A}$ is tridiagonal, so that $U, V$ have the structure $(1, 0, \ldots, 0)$, or the case when the off-diagonal part of $\mathbf{A}$ is all 1 (such as the HiPPO-LegT matrix). The matrices in Appendix D.3 and Appendix E.2 (Theorems 8, 9 and 11) actually satisfy this stronger structure, so Corollary E.3 applies.

Combining everything, this proves Theorem 2 with the exact bound $N \log^2(N) + L \log(L)$ operations. The memory claim follows similarly, and the depth of the algorithm is $\log^2(N) + \log(L)$ from the divide-and-conquer recursions.

### E.1.2 Summary of Computation Speed for LSSLs and other Mechanisms

We provide a summary of complexity requirements for various sequence model mechanisms, including several versions of the LSSL. Note that these are over exact arithmetic as in Theorem 2.

First, the self-attention mechanism is another common sequence model that has an $L^2$ dependence on the length of the sequence, so it is not suitable for the very long sequences we consider here. (We do note that there is an active line of work on reducing this complexity.)

Second, we include additional variants of the LSSL. In Table 7, LSSL-naive denotes learning $A$ and $\Delta t$ for unstructured $A$; LSSL-fixed denotes not learning $A, \Delta t$ (see Appendix B for details); LSSL denotes the learning $A$ and $\Delta t$ for the structured class $A$.

We include brief explanations of these complexities for the LSSL variants.

**LSSL-naive**

- Parameters: $O(HN)$ in the matrices $B, C$ and $O(N^2)$ in the matrix $A$.
- Training: $O(HN^3)$ to invert compute the matrix $\overline{A}$ for all $H$ features. $O(LHN^2)$ to compute the Krylov matrix $C, CA, \ldots$. $O(BL \log(L)HN)$ to multiply by $B$ and convolve with $u$.

Table 7: Complexity of various sequence models in terms of length (**L**), batch size (**B**), and hidden dimension (**H**). Measures are parameter count, training computation, memory requirement, and inference computation for 1 sample and time-step.

| | Convolution | RNN |
|---|---|---|
| Parameters | $LH^2$ | $H^2$ |
| Training | $BLH^2 + L\log(L)(H^2 + BH)$ | $BLH^2$ |
| Memory | $BLH + LH^2$ | $BLH$ |
| Parallel | Yes | No |
| Inference | $LH^2$ | $H^2$ |

| | Attention | LSSL-naive |
|---|---|---|
| Parameters | $H^2$ | $HN + N^2$ |
| Training | $B(L^2H + LH^2)$ | $HN^3 + LHN^2 + BL\log(L)HN$ |
| Memory | $B(L^2 + HL)$ | $HN^2 + LHN + BLH$ |
| Parallel | Yes | Yes |
| Inference | $L^2H + H^2L$ | $HN^2$ |

| | LSSL-fixed | LSSL |
|---|---|---|
| Parameters | $HN$ | $HN$ |
| Training | $BL\log(L)HN$ | $BH(N\log^2 N + L\log L) + BL\log(L)H$ |
| Memory | $LHN + BLH$ | $BHL$ |
| Parallel | Yes | Yes |
| Inference | $HN^2$ | $HN$ |

- Memory: $O(HN^2)$ to store $\overline{A}$. $O(LHN)$ to store the Krylov matrix. $O(BLH)$ to store the inputs/outputs
- Inference: $O(HN^2)$ to for MVM by $\overline{A}$.

**LSSL-fixed**

- Parameters: $O(HN)$ in the matrices $C$.
- Training: $O(BL\log(L)H)$ to convolve with $u$.
- Memory: $O(LHN)$ to store the Krylov matrix (but cached, so no backprop). $O(BLH)$ for inputs/outputs.
- Inference: $O(HN^2)$ to for MVM by $\overline{A}$.

**LSSL**

- Parameters: $O(HN)$ for $A, B, C, \Delta t$.
- Training: $BH \cdot \tilde{O}(N + L)$ to compute Krylov, $O(BL\log(L)H)$ for the convolution.
- Memory: $O(BHL)$ to store Krylov (and inputs/outputs).
- Inference: $O(HN)$ to multiply $x_t[H, N]$ by $\overline{A}[H, N, N]$

### E.2 Further Simplification with Tridiagonal Matrices

The algorithm for Theorem 2 for general quasiseparable matrices is still difficult to implement in practice, and we make a further simplification using a particular subclass of quasiseparable matrices.

**Theorem 11.** *The class of $N \times N$ matrices $\mathcal{S}_N = \{P(D + T^{-1})Q\}$ with diagonal $D, P, Q$ and tridiagonal $T$ includes the original HiPPO-LegS, HiPPO-LegT, and HiPPO-LagT matrices [24].*

Theorem 11 shows that a simple representation involving tridiagonal and diagonal matrices captures all of the original HiPPO matrices. In particular, our LSSL implementation initializes $A$ to be the HiPPO-LegS matrix (Appendix B) and learns within the class defined by Theorem 11.

We note that the matrices in Theorem 11 are all 1-quasiseparable and in particular also contain the HiPPO matrices for Gegenbauer and generalized Laguerre orthogonal polynomials derived in Theorem 9. In fact, the notion of semiseparability, which is closely related to (and actually is the predecessor of) quasiseparability, was originally motivated precisely to capture inverses of tridiagonal matrices. Thus the structured class in Theorem 11 can be viewed as an approximation of 3-quasiseparable matrices (Corollary 4.1) to 1-quasiseparable, which still contains many of the HiPPO families of interest.

*Proof.* We simply show that each of these specific matrices can be represented in the proposed form.

**HiPPO-LegT.**

Let $A$ denote the HiPPO-LegT transition matrix. Up to row/column scaling (i.e. left- and right-multiplication by diagonal $P$ and $Q$), we can write

$$A_{nk} = \begin{cases} (-1)^{n-k} & \text{if } n \leq k \\ 1 & \text{if } n \geq k \end{cases}.$$

The main observation is that

$$A^{-1} = \frac{1}{2} \begin{bmatrix} 1 & 1 & 0 & \ldots & 0 & 0 & 0 \\ -1 & 0 & 1 & \ldots & 0 & 0 & 0 \\ 0 & -1 & 0 & \ldots & 0 & 0 & 0 \\ \vdots & \vdots & \vdots & \ddots & \vdots & \vdots & \vdots \\ 0 & 0 & 0 & \ldots & 0 & 1 & 0 \\ 0 & 0 & 0 & \ldots & -1 & 0 & 1 \\ 0 & 0 & 0 & \ldots & 0 & -1 & 1 \end{bmatrix}$$

**HiPPO-LegS.** The HiPPO-LegS matrix is

$$A_{nk} = - \begin{cases} (2n+1)^{1/2}(2k+1)^{1/2} & \text{if } n > k \\ n+1 & \text{if } n = k \\ 0 & \text{if } n < k \end{cases}.$$

This can be written as $-PA'Q$ where $P = Q = \text{diag}((2n+1)^{\frac{1}{2}})$ and

$$A'_{nk} = \begin{cases} 1 & \text{if } n > k \\ 0 & \text{if } n < k \\ 1 - \frac{n}{2n+1} & \text{if } n = k \end{cases}.$$

Finally, $A' = D + T^{-1}$ where $D = -\text{diag}(\frac{n}{2n+1})$ and $T$ is the matrix with 1 on the main diagonal and $-1$ on the subdiagonal.

**HiPPO-LagT.** The HiPPO-LagT matrix is

$$A_{nk} = - \begin{cases} 1 & \text{if } n > k \\ 0 & \text{if } n < k \\ \frac{1}{2} & \text{if } n = k \end{cases}.$$

This can be written as $-P(D+T^{-1})Q$ where $P = Q = I$, $D = -\frac{1}{2}I$, and $T$ is the same tridiagonal matrix as in the HiPPO-LegS case. $\qquad\square$

### E.3 Implementation Details

In this section we provide several implementation details that are useful for implementing LSSLs in practice.

Recall that one of the main primitives of LSSLs is the matrix-vector multiplication $y = \overline{A}x$ (Section 3.1, Appendix B), where $\overline{A}$ is the state matrix $A$ discretized with step size $\Delta t$ using the bilinear

method (Appendix C.1.3). In Appendix E.3.1, we describe how this MVM can be performed with simpler MVM primitives which we call the "forward difference" and "backward difference".

However, if these MVM primitives are implemented in a specialized way for particular classes of $A$ matrices (i.e., not using atoms in a standard autograd framework), then we also need to calculate several additional gradients by hand. Appendix E.3.2 shows that calculating gradients to $A, \Delta t, x$ during backpropagation can actually be reduced to those same forward/backward difference primitives.

Finally, in the case when $A$ is the structured class of matrices in Theorem 11, Appendix E.2 shows how to efficiently calculate those primitives using a black-box tridiagonal solver. Our code[2] implements all the algorithms in this section, with bindings to the cuSPARSE library for efficient tridiagonal solving on GPU.

### E.3.1 Matrix-vector Multiplication by the Bilinear Discretization

**Bilinear Discretization**  The discrete state-space system is given by (4) and (5), re-written here for convenience

$$x_t = \overline{A}x_{t-1} + \overline{B}u_t$$
$$y_t = Cx_t + Du_t$$

where $\overline{A}$ is a function of $A, \delta_t$ and $\overline{A}$ is a function of $A, B, \delta_t$. In particular, we define $\overline{A}$ to be the matrix discretized using the bilinear method (Appendix C.1.3), and the system can be written explicitly:

$$x_t = \left(I - \frac{\Delta t A}{2}\right)^{-1} \left(\left(I + \frac{\Delta t A}{2}\right) x_{t-1} + \Delta t B u_t\right)$$
$$y_t = Cx_t + Du_t$$

Thus it suffices to compute the maps

$$F(A, \Delta t, x) := (I + \Delta t A)x$$

and

$$B(A, \Delta t, x) := (I + \Delta t A)^{-1}x.$$

We will call these functions the *forward difference* and *backward difference* maps, respectively. (The Euler and backward Euler discretizations (Appendix C.1.2) are also known as the "forward difference" and "backward difference" methods, which in the case of linear systems reduces down to the maps $F$ and $B$.)

### E.3.2 Gradients through the Forward/Backward Difference Primitives

In this section we will let $y = F(A, \Delta t, x)$ or $y = B(A, \Delta t, x)$ denote the computation of interest, $L(y)$ denote a generic loss function, and $dx, dy, \dots$ denote gradients to $x, y, \dots$ (e.g., $dx = \frac{\partial L(y)}{\partial x}$).

**Derivatives of backward difference.**  First we have the standard $\frac{\partial L(y)}{\partial x} = \frac{\partial L(y)}{\partial y}\frac{\partial y}{\partial x} = \frac{\partial L(y)}{\partial y}(I + \Delta t A)^{-1}$. This corresponds to matrix-vector multiplication by $(I + \Delta A)^{-T}$. In other words, it can be computed by the primitive $B(A^T, \Delta t, dy)$.

Similarly, in order to compute $\frac{\partial L(y)}{\partial \Delta t}$ we require $\frac{\partial y}{\partial \Delta t}$. We need the result $\frac{\partial Y^{-1}}{\partial x} = -Y^{-1}\frac{\partial Y}{\partial x}Y^{-1}$ for an invertible matrix $Y$ [41, equation (59)]. Then

$$\frac{\partial y}{\partial \Delta t} = \frac{\partial (I + \Delta t A)^{-1}}{\partial \Delta t}x$$
$$= -(I + \Delta t A)^{-1}\frac{\partial (I + \Delta t A)}{\partial \Delta t}(I + \Delta t A)^{-1}x$$
$$= -(I + \Delta t A)^{-1}A(I + \Delta t A)^{-1}x$$

---

[2]Available at `https://github.com/HazyResearch/state-spaces`

and

$$\frac{\partial L(y)}{\partial \Delta t} = \frac{\partial L(y)}{\partial y}\frac{\partial y}{\partial \Delta t}$$

$$= -\left[\frac{\partial L(y)}{\partial y}(I + \Delta t A)^{-1}\right] A \left[(I + \Delta t A)^{-1}x\right]$$

We can summarize this as follows. Let $y = B(A, \Delta t, x) = (I + \Delta t A)^{-1}x$ and $dy = \partial L(y)/\partial y$ (as a column vector). Then

$$y = B(A, \Delta t, x)$$
$$dx = B(A^T, \Delta t, dy)$$
$$d\Delta t = -dx^T A y.$$

**Derivatives of forward difference.** The forward case is simpler. Let $y = F(A, \Delta t, x) = (I + \Delta t A)x$. Then $\frac{\partial y}{\partial x} = I + \Delta t A$ and $\frac{\partial y}{\partial \Delta t} = Ax$. Thus

$$y = F(A, \Delta t, x)$$
$$dx = (I + \Delta t A)^T dy = F(A^T, \Delta t, dy)$$
$$d\Delta t = dy^T A x.$$

### E.3.3 Computing the Forward/Backward Difference for Tridiagonal Inverse Matrices

Theorem 11 uses the classes of matrices $A = P(D + T^{-1})Q$ for diagonal $D, P, Q$ and tridiagonal $T$. We describe how the forward and backward difference MVMs can be performed efficiently for this class of matrices by reducing to a black-box tridiagonal solver.

**Forward difference.** It is straightforward to compute

$$F(A, \Delta t, x) = (I + \Delta t \cdot P(D + T^{-1})Q)x = x + \Delta t \cdot PDQx + \Delta t \cdot PT^{-1}Qx$$

in terms of multiplication by diagonal matrices $x \mapsto Dx$ and tridiagonal solving $x \mapsto T^{-1}x$.

**Backward difference.** We will explicitly rewrite the inverse of the matrix $G = I + \Delta t \cdot P(D + T^{-1})Q$.

The core observation is to multiply $G$ by a choice selection of matrices to cancel out the $T^{-1}$ term:

$$TP^{-1}GQ^{-1} = TP^{-1}Q^{-1} + \Delta t TD + \Delta t I.$$

Rearranging yields

$$G^{-1} = Q^{-1}(TP^{-1}Q^{-1} + \Delta t TD + \Delta t I)^{-1}TP^{-1}.$$

Now note that the matrix in the middle is tridiagonal. Hence we have reduced MVM by $G^{-1}$, i.e. the backward difference problem, to a series of diagonal and tridiagonal MVMs (easy), and a tridiagonal inverse MVM (a.k.a. a tridiagonal solve).

## F   Additional Experiments and Experiment Details

We provide additional experiments and ablations in Appendix F.1. Appendix F.2 describes our training methodology in more detail for each dataset. The hyperparameters for all reported results are in Table 11.

### F.1   Additional Experiments

**Missing Data on CharacterTrajectories.** Table 8 has results for a setting considered in previous work involving irregularly-sampled time series. LSSL is competitive with the best prior methods, some of which were specialized to handle this setting.

Table 8: Test accuracies for irregularly sampled time series on the CharacterTrajectories dataset. $p\%$ denotes percent of data that was randomly dropped.

| Model | 0% | 30% | 50% | 70% |
|---|---|---|---|---|
| GRU-ODE [16] | - | 92.6 | 86.7 | 89.9 |
| GRU-$\Delta t$ [31] | - | 93.6 | 91.3 | 90.4 |
| GRU-D [9] | - | 94.2 | 90.2 | 91.9 |
| ODE-RNN [45] | - | 95.4 | 96.0 | 95.3 |
| NCDE [31] | - | 98.7 | 98.8 | **98.6** |
| CKCNN [44] | **99.53** | **98.83** | 98.60 | 98.14 |
| LSSL | 99.30 | **98.83** | **98.83** | 98.37 |

Table 9: $A$ and $\Delta t$ ablations on sCIFAR.

| | Learn $\Delta t$ | Fixed $\Delta t$ |
|---|---|---|
| Learn $A$ | 82.70 | 80.34 |
| Fixed $A$ | 80.61 | 80.18 |

Table 10: $A$ and $\Delta t$ ablations on SC-Raw.

| | Learn $\Delta t$ | Fixed $\Delta t$ |
|---|---|---|
| Learn $A$ | 96.07 | 95.20 |
| Fixed $A$ | 91.59 | 90.51 |

$A$ **and $\Delta t$ ablations.** Tables 9 and 10 show results on SpeechCommands-Raw and a smaller model on sCIFAR, ablating that learning either the $A$ or $\Delta t$ parameters provides a consistent performance increase.

Finally, Fig. 2 plots the $\Delta t$ values at the beginning and end of training on the SpeechCommands-Raw dataset, confirming that training $\Delta t$ does noticeably change their values to better model the data. In particular, the $\Delta t$ values spread over time to cover a larger range of timescales.

## F.2 Methodology

We describe our training procedure on each dataset for our model and any relevant baselines.

**General** All models and datasets used the Adam optimizer with a LR decay scheduler that reduced LR by 5x upon validation plateau for 10 or 20 epochs. We fixed the batch size to 50 for the MNIST/CIFAR datasets and 32 for other datasets, reducing if necessary to fit in memory.

For all models, we chose the hyperparameters that achieved the highest validation accuracy/RMSE (values in Table 11).

**Error Bars** We note that the results in Section 5 do not include standard deviations for formatting reasons, since most of the baselines were best results reported in previous papers without error bars. As Section 6 noted, the LSSL was actually quite stable in performance and not particularly sensitive to hyperparameters. We note that for every result in Section 5, the LSSL with error bars was at least one standard deviation above the baseline results.

### F.2.1 Sequential and Permuted MNIST

The model architecture of LSSL(-f) was fixed to the small architecture with 200K parameters (Appendix B). Following [44], we fixed the learning rate scheduler to decay on plateau by with a factor of 0.2, and the number of epochs to 200. We searched hyperparameters over the product of the following learning rate values: $\{0.001, 0.002, 0.004, 0.01\}$, and dropout values: $\{0.1, 0.2\}$.

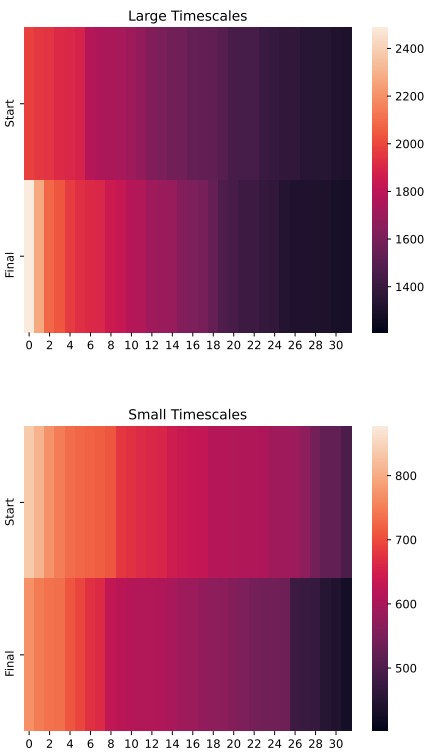

Figure 2: We visualize the 32 largest and smallest $\Delta t$ values at the start and end of training for the first layer of our state-of-the-art LSSL model on the Speech Commands Raw dataset. The plots visualize $\frac{1}{\Delta t}$, which can be interpreted as the timescale at which they operate (Section 2). The plots confirm that LSSL does modify the dt values in order to more appropriately model the speech data.

### F.2.2 Sequential CIFAR

The model architecture of LSSL(-f) was fixed to the large architecture with 2M parameters (Appendix B). We searched over the product of the following learning rate values: $\{0.001, 0.002, 0.004, 0.01, 0.02\}$, and dropout values: $\{0.2, 0.3, 0.4\}$.

### F.2.3 BIDMC Healthcare

The BIDMC tasks aim at predicting three vital signs of a patient, respiratory rate (RR), heart rate (HR), and oxygen saturation (SpO2), based on PPG and ECG signals. The clinical data is provided by the Beth Israel Deaconess Medical Center. The PPG and ECG signals were sampled at 125Hz and have a sequence length of 4000.

For this dataset, we fixed the small LSSL(-f) model (Appendix B). Following [47], we changed the scheduler to a multistep scheduler that decays on fixed epochs, and trained for 500 epochs.

For our methods, we searched over the product of the following learning rate values: $\{0.004, 0.01, 0.02\}$, and dropout values: $\{0.1, 0.2\}$.

**Baseline parameters.** For CKConv, we searched over $\omega_0 \in [10, 50]$ following the guidelines of Romero et al. [44] (best value $\omega_0 = 20$). Since we tuned the sensitive $\omega_0$, we fixed the learning rate to $0.001$ and dropout to $0.1$ which was the default used in [44].

The transformer model we used was a vanilla transformer with a hidden dimension of 256, 8 attention heads, 4 layers, and a feedforward dimension of 1024. We used a learning rate of $0.001$ and a dropout of 0. We tried a few variants, but no transformer model was effective at all.

### F.2.4 CelebA

For these larger datasets, we reduced the size of the order $N$ and did not tie it to $H$. These experiments were computationally heavy and we did not do any tuning (i.e., Table 11 are the only runs). The model size was picked to train in a reasonable amount of time, and the learning rate for the first attribute was picked based on general best hyperparameters for other datasets, and then reduced for subsequent experiments on the other attributes.

**Baseline parameters.** For ResNet-18, we used the standard implementation with a learning rate of $0.001$.

### F.2.5 Speech Commands

For Speech Commands, we use the same dataset and preprocessing code from Kidger et al. [31], Romero et al. [44]. We consider the two settings from Kidger et al. [31]: SC-Raw uses very long time-series raw speech signals of 16000 timesteps each, while SC-MFCC uses standard MFCC features of 161 timesteps.

For our models trained over the raw data, we searched over the product of the following learning rate values: $\{0.002, 0.004, 0.01\}$, and dropout values: $\{0.1, 0.2\}$. For our models trained over the MFCC features, we searched over the product of the following learning rate values: $\{0.0001, 0.001, 0.002, 0.004, 0.01\}$, and dropout values: $\{0.1, 0.2, 0.3, 0.4\}$.

**Baseline parameters.** To get more results for the strongest baselines on very long sequences in the literature, we ran the UniCORNN [47] baseline on both Raw and MFCC variants, and the Neural Rough Differential Equations [37] baseline on the Raw variant.

For UniCORNN trained over the raw data, we searched over multiple hyperparameters. Specifically, we searched over alpha: $\{0, 10, 20, 30, 40\}$, $\Delta t$ values: $\{0.00001, 0.0001, 0.001, 0.01\}$, and learning rate values: $\{0.0001, 0.0004, 0.001, 0.004\}$. However, since the method was not able to generalize to the validation set for any hyperparameter combination, we used the authors' reported hyperparameters for the Eigenworms dataset as it also contains very long sequences ($\approx 18000$). In particular, we used a learning rate of $0.02$, hidden dimension of $256$, 3 layers with dt values $[0.0000281, 0.0343, 0.0343]$, dropout of $0.1$, and alpha of $0$.

For UniCORNN trained over the MFCC features, we used the authors' reported hyperparameters for the MNIST dataset (again due to similarly sized sequence lengths), and further tuned the learning rate over the values: $\{0.0001, 0.001, 0.005, 0.01, 0.02\}$, $\Delta t$ values: $\{0.01, 0.1\}$, and alpha values: $\{10, 20, 30\}$.

The best model used a learning rate of $0.02$, hidden dimension of $256$, 3 layers with dt values of $0.19$, dropout of $0.1$, and alpha of $30.65$.

For NRDE on SC-Raw, we used depth 2, step size 4, hidden dimension 32, and 3 layers. Our results were better than unofficial numbers reported in correspondence with the authors, so we did not tune further.

### F.2.6 Convergence Speed (Table 5)

The convergence table compared against logs directly from the corresponding baseline's SoTA models [44, 47], which were either released publicly or found in direct correspondence with the authors. To generate the wall clock numbers, we ran the baseline models on the same hardware as our models and extrapolated to the target epoch.

## F.3 Hyperparameters

Best hyperparameters for all datasets are reported in Table 11.

Table 11: The values of the best hyperparameters found for each dataset.

| Dataset | Hyperparameters | | | | | | | |
|---|---|---|---|---|---|---|---|---|
| | Learning Rate | Dropout | Batch Size | Epochs | Depth | Hidden Size $H$ | Order $N$ | Channels $M$ |
| sMNIST | 0.004 | 0.2 | 50 | 200 | 6 | 128 | 128 | 1 |
| pMNIST | 0.001 | 0.2 | 50 | 200 | 6 | 128 | 128 | 1 |
| sCIFAR | 0.02 | 0.3 | 50 | 200 | 4 | 256 | 256 | 4 |
| BIDMC-RR | 0.004 | 0.1 | 32 | 500 | 6 | 128 | 128 | 1 |
| BIDMC-HR | 0.01 | 0.2 | 32 | 500 | 6 | 128 | 128 | 1 |
| BIDMC-SpO2 | 0.01 | 0.1 | 32 | 500 | 6 | 128 | 128 | 1 |
| SC Raw | 0.01 | 0.2 | 16 | 50 | 4 | 256 | 128 | 2 |
| SC MFCC | 0.004 | 0.4 | 32 | 100 | 6 | 128 | 128 | 1 |
| sCelebA-Att. | 0.002 | 0.1 | 32 | 200 | 3 | 256 | 128 | 4 |
| sCelebA-MSO | 0.002 | 0.1 | 32 | 200 | 3 | 256 | 128 | 4 |
| sCelebA-Smil. | 0.01 | 0.1 | 32 | 200 | 3 | 256 | 128 | 4 |
| sCelebA-WL | 0.002 | 0.1 | 32 | 200 | 3 | 256 | 128 | 4 |