# OpenReview forum: "Combining Recurrent, Convolutional, and Continuous-time Models with Linear State Space Layers"
_NeurIPS.cc/2021/Conference — NeurIPS 2021 Poster_

### Official Review · Reviewer_LpSP · 2021-07-15

**Rating:** 7
**Confidence:** 5

**Summary:**

The authors proposed structured learnable linear state-space layers as a way to unify recurrence, convolutions, and continuous-time representations. The performance of SLLSSL is evaluated on a set of sequential benchmarks against a couple of modern and classic baseline deep models.

**Limitations And Societal Impact:**

The authors discussed partially the limitations of their approach.
I do not see any potential negative societal impact from this work.


**Main Review:**

remarks:
+ The experimental framework is properly designed and shows the benefits of the proposed SLLSSL approach.
+ The theoretical results are insightful; proofs are provided in Appendix and the code is provided in full. Very much appreciated!
+ Incorporating convolution operators in LSSL is novel.

Concerns:

-	I believe that the approach proposed here has limited novelty. In particular, the bilinear transformation which is core to the method proposed was historically introduced as dynamic causal models [1] back in 2003, and more recently as Liquid time-constant networks (LTCs) [2]. I believe authors should clearly differentiate their framework from DCMs and LTCs while including them in their experimentation. In particular, 1) I suggest authors formally state what is new in LSSL compared to LTCs? 2) The time-gating discovery of this paper is very similar to that of the liquid time-constant proposed in LTCs. Please comment. 3) The fact that authors find 5x fewer parameters than the previous SoTA was observed for networks comprised of LTCs. 4) The expressivity of the models is thoroughly discussed in LTCs whereas LSSL does not provide formal evidence. What is new here in this regard? 5) How would LTC networks equipped with HIPPO similar to the SLLSSL models perform in the experimental setup provided in the paper.

-	All ODE-based models provably suffer from gradient issues [3] ergo learning long-term dependencies (similar to LSSL) and naturally require modules to tackle this shortcoming. The authors use HIPPO in this case. But there are many fundamental approaches proposed for solving the long-term dependency learning capabilities of RNNs and CT-RNNs that are missing in this paper. In particular, orthogonal models [4] Lipschitz RNNs [5], and mixed memory architectures such as ODE-LSTMs [3] are encouraged to be used as baselines in all experiments concerning recurrent networks.

-	[less critical] In the Intro, the authors draw a solid line between RNN models and continuous-time models which I find inaccurate. RNNs are a discretized version of CT networks. In line 65, the Authors mention the limitations of RNNs in remembering long dependencies, but in fact, all ODE-based RNNs also suffer from this limitation [3], because CT models are where RNNs are originally coming from.  The state-space layer is indeed a continuous-time network, not the other way around.


[1] Friston, Karl J., Lee Harrison, and Will Penny. "Dynamic causal modelling." Neuroimage 19.4 (2003): 1273-1302.
[2] Hasani, Ramin, et al. "Liquid time-constant networks." arXiv preprint arXiv:2006.04439 (2020).
[3] Lechner, Mathias, and Ramin Hasani. "Learning long-term dependencies in irregularly-sampled time series." arXiv preprint arXiv:2006.04418 (2020). AAAI 2021
[4] Lezcano-Casado, Mario, and David Martınez-Rubio. "Cheap orthogonal constraints in neural networks: A simple parametrization of the orthogonal and unitary group." International Conference on Machine Learning. PMLR, 2019.
[5] Erichson, N. Benjamin, et al. "Lipschitz Recurrent Neural Networks." International Conference on Learning Representations. 2020.

**Time Spent Reviewing:**

25

---

> ### Author Response · Authors · 2021-08-07
> **Additional Experiments**
>
> The reviewer has suggested several related models as baselines: the expRNN [Lezcano-Casado 2019], Lipschitz RNN [Erichson 2021], ODE-LSTM [Lechner 2020], and LTC [Hasani 2021]. We first show that **these baselines perform poorly on the long sequences used in our work**, whether due to computational or optimization issues, and then make direct comparisons that the **LSSL also outperforms these baselines on shorter tasks** used in these prior works.
>
> ## Performance on Long Sequences
>
> We tested the proposed baselines on our longer datasets, the BIDMC healthcare regression (length 4000) and Speech Commands classification (length 16000).
>
> ### expRNN
>
> Comparison to expRNN on BIDMC is already in Table 3 of the submission. These were the RMSE scores on respiratory rate (RR) and heart rate (HR):
> - **LSSL: 0.35 (RR), 0.43 (HR)**
> - **expRNN: 1.57 (RR), 1.87 (HR)**
>
> We additionally ran expRNN on the SC dataset, where it scored **27.9%** accuracy after 20 epochs, compared to **95.9%** accuracy for LSSL.
>
> ### Lipschitz RNN
>
> On the BIDMC HR dataset, Lipschitz RNN is 8x slower per epoch compared to LSSL and made no progress over random guessing after 50 epochs (**RMSE 10.6**), compared to the LSSL which outperformed all baselines after 10 epochs (**RMSE 0.88**)
>
> ### ODE-LSTM, LTC
> These models based on ODE solvers are **computationally intractable** on long sequences.
>
> We ran the ODE-LSTM on BIDMC, where it is **over 1000x slower than LSSL** (~12 hours vs 30 seconds per epoch). LTC is even slower, as acknowledged in the paper.
> We found similar computation issues with the continuous-time RNN baselines in the submission (Table 5).
>
> ## Performance on Short Sequences
>
> ### expRNN
>
> The expRNN paper reported results on sMNIST and pMNIST [Lezcano-Casado 2019, Table 1], where it has accuracy 98.7% and 96.6%, compared to 99.5% and 98.8% of LSSL (Table 2).
>
>
> ### Lipschitz RNN
>
> The original paper [Erichson 2021] included a common dataset with us, sequential and permuted MNIST. The Lipschitz RNN scored 99.4% and 96.3% respectively, compared to our 99.5% and 98.8%.
>
> We have additionally run the Lipschitz RNN using their code on the harder dataset sequential CIFAR, where the Lipschitz RNN scored **58.3%** compared to our **84.6%** (Table 2).
>
> We additionally note that these Lipschitz RNN experiments downsampled the input by 10x to be length ~100 instead of ~1000 as in our baselines.
>
>
> ### ODE-LSTM
>
> The ODE-LSTM paper [Lechner 2020] had no common datasets with us, so we ran our model on two datasets from [Lechner 2020] in their exact same experimental setting. The following experiments used the same **small LSSL model as in Table 2 and 3 with no tuning** (specified in Appendix A and D).
>
> #### MSE on the Walker2D task for autoregressive prediction (length 64):
> - **LSSL: 0.597 ± 0.015**
> - **ODE-LSTM: 0.883 ± 0.014**
> - Other continuous-time baselines (ODE-RNN, CT-RNN, Augmented LSTM, CT-GRU, RNN-Decay, Bi-directional RNN, GRU-D, PhasedLSTM, GRU-ODE, CT-LSTM): 1.904 ± 0.061 to 1.014 ± 0.014
>
> #### Accuracy on the ET-sMNIST task for irregular classification (length 256):
> - **LSSL: 98.15 ± 0.10**
> - **ODE-LSTM: 95.73 ± 0.24**
> - Other continuous-time baselines (ODE-RNN, CT-RNN, Augmented LSTM, CT-GRU, RNN-Decay, Bi-directional RNN, GRU-D, PhasedLSTM, GRU-ODE, CT-LSTM): 72.41 ± 1.69 to 94.84 ± 0.17
>
>
> ### Liquid Time-Constant Networks
>
> The LTC paper [Hasani 2021] had one common dataset with us, sequential MNIST:
> - LSSL: **99.52 ± 0.01** (Table 2)
> - LTC: **97.57 ± 0.18**
>
> We note that all other datasets considered in [Hasani 2021] had **sequence length 32**, compared to our experiments which had sequences of length **1000, 4000, 16000, and 38000**.
>
> We ran one additional experiment from [Hasani 2021], the Half-Cheetah autoregressive prediction task (length 32), with MSE scores listed below
> - **LSSL: 2.071 ± 0.030**
> - **LTC: 2.308 ± 0.015**
> - Other baselines (LSTM, CT-RNN, Neural ODE, CT-GRU): between 2.500 ± 0.140 and 3.805 ± 0.313.
>
>
> ### Final Remarks
>
> We acknowledge that some of these prior works may have had different goals than us. For example, the ODE-LSTM excels on irregularly sampled time series instead of long time series. While the LSSL also has preliminary results in that regime (shown in Table 9 in the submission; prior work [Gu 2020, Table 2]; and the additional experiments on Walker2d and ET-SMNIST here), our main motivations were on different aspects of CT models. We believe that ideas from all of these works can be used to improve continuous-time models at large, and are excited about the future of this area.

---

> ### Author Response · Authors · 2021-08-07
> **Liquid-Time Constant Networks (LTC)**
>
> We thank the reviewer for pointing us to the recent paper on LTCs [Hasani 2021]. Indeed, LTCs theoretically share several similarities to LSSLs in their ODE formulation, and are an interesting and relevant model in the CT family. Ultimately, LSSLs and LTCs are different models with different theoretical, computational, and empirical properties. We believe that there was a simple misunderstanding over the relationship between LSSL and DCM (and LTC) based on terminology, and address this first.
>
> We respond to specific points in more detail below, and include a section in our revised paper discussing the LTC and other CT models.
>
> ------
>
> > bilinear transformation which is core to the method proposed was historically introduced as dynamic causal models back in 2003, and more recently as Liquid time-constant networks (LTCs)
>
> The reviewer's claim that the LSSL has limited novelty compared to the DCM (and LTC) is based on the premise that they all use a "bilinear transformation". We believe this is a misconception, and claim that **the LSSL and DCM are fundamentally different, coincidentally sharing the word "bilinear" with different etymologies**.
>
> * The DCM [Friston 2003] uses a *bilinear state equation* of the form $x'(t) = A(t)x(t) + B(t)x(t) u(t) + C(t)u(t)$. Here the word "bilinear" refers to this as being an 2nd-order Taylor approximation to a *non-linear* ODE. More precisely, this bilinear state equation is a [bilinear map](https://en.wikipedia.org/wiki/Bilinear_map) (in the linear algebraic sense) on the input $u$ and state $x$, since they interact multiplicatively.
> * The LSSL is based on the [state space representation](https://en.wikipedia.org/wiki/State-space_representation) using a *linear state equation* $x'(t) = Ax(t) + Bu(t)$, which is a fundamentally different ODE with no non-linear terms. We point out that **state spaces have a rich history in control engineering stemming from the mid 1900s, far pre-dating DCMs**.
> * Linear state equations can be numerically approximated using various methods (analogous to the classic Euler's method [Line 114]). The [bilinear transform](https://en.wikipedia.org/wiki/Bilinear_transform) for LTI systems is one such method, originally from [Arnold Tustin, 1947]. Here the name "bilinear" refers to the [Mobius transformation](https://en.wikipedia.org/wiki/M%C3%B6bius_transformation) (which are also called bilinear transformations) and is a fundamentally different usage of the word. Other discretization methods would work just as well, such as the [zero-order hold](https://www.wikiwand.com/en/Zero-order_hold) method used by [Voelker 2019].
> * We point out that although the equations for the LSSL's linear state equation may look similar to the DCM's bilinear state equation, the missing bilinear term $x(t) u(t)$ as well as the coefficients $A, B$ being constant gives it fundamentally different properties. Many of the special properties of LSSLs, such as their connection to convolutions, arise from them being a [linear time-invariant (LTI) system](https://en.wikipedia.org/wiki/Linear_time-invariant_system), and does not apply to non-linear ODEs such as the DCM.
>
> In summary, the LSSL leverages the "bilinear transform" (a.k.a. Tustin's method [Tustin 1947]) which turns a *linear state equation* into a *linear recurrence*. It is unrelated to the DCM's [Friston 2003] "bilinear dynamical system" (a.k.a. 2nd order Taylor series) which is a *bilinear state equation* approximating a *non-linear ODE*. The word "bilinear" in the former refers to the Mobius approximation, while the word "bilinear" in the latter refers to the linear algebraic concept.
>
>
> ------
>
> > I suggest authors formally state what is new in LSSL compared to LTCs?
>
> Our above response about the bilinear transform showed that LSSLs and LTCs are not related. We elaborate in more detail about some differences:
>
> - **LSSLs and LTC are simply defined by different ODEs** (see above response). LSSLs are based on the rich theory of state spaces and have many properties, including their connection to convolutions, that are not known for LTCs.
>
> - LSSLs are able to address long-range dependencies by generalizing the recent HiPPO theory of continuous-time memorization [Gu 2020], which is unrelated to LTCs [Hasani 2021].
>
> - Computationally, the linearity of LSSLs allow **parallelizable training with the convolutional view** (Section 3.2) using standard backpropagation. By contrast, LTC is more similar to conventional neural ODEs and is trained using the adjoint method or a new BPTT method, which the authors acknowledge is slow in practice [Hasani 2021, Sec 8].
>
> - Empirically, LSSLs are designed for **long sequences** while LTCs have different advantages such as state stability.
>
>
>
> > The time-gating discovery of this paper is very similar to that of the liquid time-constant proposed in LTCs. Please comment.
>
> Indeed similar observations linking gating mechanisms of RNNs to CT models have been made in previous work, for example in [Tallec 2018, Section 1] and [Gu 2020, Section 3.2] as well as the reviewer’s references. Our theoretical results on LSSLs and RNNs (Section 3.2, Appendix B.2) build on this line of work to provide more perspectives and understanding. We will add a reference that LTCs and other works have made related observations about gating.
>
> Concretely, the contribution of Lemma 3.1 is showing that the precise formulation of the ubiquitous **sigmoid** gating mechanism arises exactly from applying a standard state space discretization to the type of damped ODE appearing in LSSLs, LTCs, and more. In the RNN literature, the sigmoid function is usually motivated as an arbitrary function with range (0,1), and it was not previously observed that the exact formula actually arises in a principled way.
>
> > ​The fact that authors find 5x fewer parameters than the previous SoTA was observed for networks comprised of LTCs.
>
> We agree that LTCs are also an interesting CT model that can achieve strong empirical performance with few parameters. We point out that LTCs were evaluated on very different tasks than LSSLs [Hasani 2021], so these observations about parameter count are not directly comparable.
>
>
>
> > The expressivity of the models is thoroughly discussed in LTCs whereas LSSL does not provide formal evidence. What is new here in this regard?
>
> **LTCs and LSSLs prove different types of expressivity results** (Section 3.2, Appendix B.2); while the former directly proves a universal approximation theorem, the latter focuses more on the conceptual connection to existing models (RNNs and convolutions).
>
> **We hypothesize that showing universality of LSSLs should also be possible.** For example, this could proceed by first showing that LSSLs are universal approximators for convolutions using standard methods for state-space systems (Section 3.2), and then showing that convolutions are universal approximators. Ideas from LTCs may help here, and we leave this as an interesting question for future work.
>
> Finally, we note that **LSSLs satisfy related approximation guarantees** that were proven in prior work. For example, a precise approximation bound of the HiPPO ODE was shown in terms of the smoothness of the input function [Gu 2020, Proposition 6].
>
>
> > How would LTC networks equipped with HIPPO similar to the SLLSSL models perform in the experimental setup provided in the paper.
>
> HiPPO defines a specific linear ODE of the form equation (1) that goes hand-in-hand with LSSL. In fact, this was one of the primary motivations of the LSSL model. However, **it is not clear how HiPPO could be combined with other continuous-time models aside from LSSL**, which is an interesting future direction.
>
>
> ### References
>
> - Erichson et al, 2021. “Lipschitz Recurrent Neural Networks.” ICLR 2021.
> - Gu et al, 2020. “HiPPO: Recurrent Memory with Optimal Polynomial Projections.” NeurIPS 2020.
> - Hasani et al, 2021. “Liquid Time-Constant Networks.”  AAAI 2021.
> - Lechner and Hasani, 2020. "Learning long-term dependencies in irregularly-sampled time series."
> - Lezcano-Casado and Martinez-Rubio, 2019. “Cheap orthogonal constraints in neural networks: A simple parametrization of the orthogonal and unitary group.” ICML 2019.
> - Tallec and Ollivier, 2018. “Can Recurrent Neural Networks Warp Time?”. ICLR 2018.
> - Voelker et al, 2019. “Legendre Memory Units: Continuous-Time Representation in Recurrent Neural Networks.” NeurIPS 2019.

---

> ### Author Response · Authors · 2021-08-07
> **Comparison to Continuous-Time Models**
>
> The LSSL is indeed a continuous-time (CT) model that builds upon a rich line of recent work, and we appreciate the reviewer’s suggestions which allow us to include a richer discussion and more baselines in the revised paper. Despite the similarities, the LSSL is ultimately a new model with different properties, leading to some theoretical and empirical advantages. As just one example, LSSLs are trained in parallel using convolutions and are **2-3 orders of magnitude faster compared to standard CT or RNN models** trained using the adjoint method or BPTT. Extensive empirical comparisons between LSSL and CT baselines are provided in the submission (Tables 5, 9) and additional experiments below.
>
> > The state-space layer is indeed a continuous-time network, not the other way around.
>
> > RNNs are a discretized version of CT networks.
>
> We fully agree with these statements. In the Intro we claim that the LSSL is an example of a continuous-time network that has properties of both RNNs and convolutions [Line 54], not that they generalize CT models. We will also add more discussion on the history of the CT to RNN connection as discussed in [Hasani 2021, Lechner 2020].
>
> > the Authors mention the limitations of RNNs in remembering long dependencies, but in fact, all ODE-based RNNs also suffer from this limitation, because CT models are where RNNs are originally coming from
>
> > All ODE-based models provably suffer from gradient issues [Lechner 2020] ergo learning long-term dependencies (similar to LSSL) and naturally require modules to tackle this shortcoming.
>
> We disagree that all ODE-based RNNs struggle with remembering long dependencies. As the reviewer notes, RNNs come from CT models, hence the optimization properties of the RNN are inherited from the underlying ODE. Moreover, there are mathematically derived ODEs that can remember for hundreds of thousands of time steps (e.g. LMU [Voelker 2019], HiPPO), which is why the LSSL succeeds at long sequences. We point out that this does not contradict [Lechner 2020, Theorem 2] which is about a specific parameterized ODE, while for example HiPPO [Gu 2020, Proposition 5] is about a different non-trainable ODE.
>
> Our empirical results are a direct counterpoint to the reviewer's claims: the LSSL is a simple ODE-based model that does not suffer from gradient issues and excels at long-term dependencies.

---

> ### Author Response · Authors · 2021-08-07
> **Response to Reviewer LpSP**
>
> We are encouraged that the reviewer found our experiments compelling and the theory insightful. We appreciate the reviewer’s detailed comments, baseline and experimental suggestions, and overall expertise on continuous-time (CT) networks, which has helped improve the context of our work. We provide some general comments about the theoretical and empirical contributions of our paper, and then respond in detail to the reviewer’s technical comments and experimental suggestions.
>
> ### Conceptual Contributions
>
> The reviewer’s primary concern is the novelty around the CT to RNN connection. While our method is certainly related to previous CT/RNN works, it is clearly distinguished in many ways. Compared to conventional CT/RNN methods, it is:
> - based on a different underlying ODE,
> - orders of magnitude faster based on new computational viewpoints and algorithms,
> - grounded in recent theory on continuous-time memorization, allowing it to address realistic time series of thousands of steps.
>
> We contrast the LSSL to prior works such as LTC in detail below. **We point out that the reviewer's primary concern around novelty compared to the DCM and LTC may be based on an accidental misunderstanding of the technical term "bilinear"**, which we elaborate on below.
>
> Additionally, while the connections between CT and RNNs have indeed been studied in the past, **a major theme of this work is to also draw the connection between CT and convolutional models** - another major class of models that is usually treated separately from CTs and RNNs. This connection is made through a link to state-space models, another important system that has been well-studied in signal processing. We hope that our technical contributions will advance the understanding of how these fundamental models (CT, RNNs, CNNs, state spaces) relate to each other.
>
> ### Empirical Contributions
>
> A central empirical contribution of our work is showing that our model is the first CT/recurrent model that succeeds in settings (such as healthcare and speech) involving **very long sequences** of 1000 to 40000 time steps. Although the reviewer has suggested several additional CT baselines [Hasani 2021, Lechner 2020, Lezcano-Casado 2019, Erichson 2021], all of them evaluated exclusively on short sequences of a few dozen to hundred steps. For example, the primary LTC baseline suggested by the reviewer explicitly mentions that it is not designed for long sequences [Hasani 2021, Sec 8]. We would also like to note that the submission already includes many recent CT models as baselines (Table 5, Table 9).
>
> Below, we have added additional experiments with direct comparisons to all of the proposed baselines, where the **LSSL still surpasses the performance of these baselines on the short tasks** (e.g. length 32 to 256) considered in these previous works.
>
> ---
>
> In summary, the **LSSL is clearly distinguished from previous continuous-time models** in the following ways, among others:
> * Novel application of a fundamental linear ODE (the state space equation), compared to standard CT models which learn a parameterized nonlinear ODE
> * Multiple conceptual viewpoints (recurrent, convolutional) allowing computation 100x or more faster than other CT models
> * Generalization of recent theory on continuous-time memorization (HiPPO), allowing handling extremely long sequences that have not previously been possible with CT/recurrent models.

---

> > ### Comment · Reviewer_LpSP · 2021-08-27
> > **Response**
> >
> > I thank the authors for their elaborated response to my concerns. I respond to their arguments and clarifications in this single message.
> >
> > ### Coceptual Contributions:
> >
> > I believe that the improvements in speed and accuracy achieved by LSSL in sequence modeling are direct results of the solid engineering frameworks used, namely the HIPPO memorization scheme + optimization of the codebase in C++, and the linearization of operations. On the conceptual level, however, the work has limited impact. In particular, the authors counted the following points as conceptual contribution differences to other CT models:
> >
> > 1) " based on a different underlying ODE "
> >
> > I disagree. The ODE proposed here is fundamentally a linear system with fixed parameters. It is indeed a particular linear subclass of DCMs and LTCs. See more details in **(a)**.
> >
> > 2) " orders of magnitude faster based on new computational viewpoints and algorithms,"
> >
> > The gain in speed is a direct result of the downstream engineering tricks used, namely: HIPPO, optimization in C++, and linearization of operations. Moreover, the computational advantages gained are not conceptual distinctions of the proposed model compared to other CTs. I argue that with the exact same engineering steps we can also gain speed up for any CT model.
> >
> > 3) " grounded in recent theory on continuous-time memorization, allowing it to address realistic time series of thousands of steps. "
> >
> > Exactly. Again an advantage that the authors mention is gained thanks to the continuous-time memorization (HIPPO).
> >
> > 4) "a major theme of this work is to also draw the connection between CT and convolutional models"
> >
> > I fully agree and as a matter of fact, in my original response have appreciated this contribution.
> >
> > **(a) Bilinear Approximation**
> >
> > Although the bilinear approximation term is different in these notions, their resulting dynamical systems are very similar. If we relax the Taylor approximation conditions from a DCM, in fact, both equations become similar, obviously with the distinction of one (LSSL) being linear and one (DCM and LTCs) being nonlinear. Therefore, in my opinion, LSSL will be best introduced as a linear version of the more general framework such as DCMs and liquid networks.
> >
> > **(b) On fundamental differences of LSSL with CT models and LTCs**
> >
> > - I believe that the formulation of LSSL is simply a linearized version of LTCs, based on the above arguments.
> > - There is a direct connection between state-space modeling and the type of dynamic causal model you are working with here. As the authors also mentioned, there are many ways one could approximate dynamics. When it comes down to function approximation the representation of the resulting system is what matters. That is, in the case of LSSL, its representation is a subset of DCMs and LTCs.
> > - The authors say: "We point out that although the equations for the LSSL's linear state equation may look similar to the DCM's bilinear state equation, the missing bilinear term as well as the coefficients being constant gives it fundamentally different properties." I fully agree, but this is exactly where I would argue that fundamentally and theoretically from the expressivity point of view, LSSL has limitations in the range of functions it can approximate given the same number of parameters similar to an LTC network. I will comment on the additional experimental results of the paper in the next section of my response.
> >
> > **(c) On the theoretical claims related to Time-Gating**
> >
> > Authors say: "Concretely, the contribution of Lemma 3.1 is showing that the precise formulation of the ubiquitous sigmoid gating mechanism arises exactly from applying a standard state space discretization to the type of damped ODE appearing in LSSLs, LTCs, and more. In the RNN literature, the sigmoid function is usually motivated as an arbitrary function with range (0,1), and it was not previously observed that the exact formula actually arises in a principled way."
> >
> > I disagree. Time-gating is not a consequence of state-space discretization. It is simply an emerging property of damped ODEs where the damping factor is input-dependent. The sigmoid function is not defined as an arbitrary function with a range (0,1). It is defined as a positive, monotonically increasing, and bounded function.
> >
> > **(d) On the approximation capabilities of LSSL vs others** Authors totally misunderstood this point. Universal approximation property is certainly easy to derive. The more rigorous measure of the approximation capability of a deep network, in general, is the expressivity measures such as the number of linear regions [Montufar et al 2014] or the trajectory length measure [Raghu et al. 2017]. In order to show LSSL's expressive power in a fundamental way, you can use these measures to discuss the expressivity of the LSSL models. Although, based on the current form of the LSSL formulation, I suspect that their expressivity limit is very similar to that of linearized CT-RNNs (Funahashi et al. 1993) which was proven to be limited compared to LTCs.
> >
> > -------------------------------
> > ### Experimental Contributions
> >
> > I appreciate the additional experiments on ODE-LSTMs which indeed makes the experimental parts of the paper to be more conclusive. However, the results on the other methods are not fair as these models are not incorporated on the same engineering framework (Hippo + computational tricks). More in the following:
> >
> > *The very long-term sequential data modeling capability of LSSL* I totally agree and believe that building networks based on the Hippo framework + the underlying engineering tricks are the core reasons why we see improved performance. That is why I asked the authors to explicitly equip the parent family of LSSL (LTCs) with the HIPPO framework and then compare them against each other. Unfortunately, the authors did not include any evidence of such an experiment for this and argued that it is not clear how to do this.
> >
> > Overall, based on these arguments which are not addressed, I still believe that the paper is of borderline impact and would like to keep my score as is.

---

> > > ### Author Response · Authors · 2021-08-28
> > > **Further Response to Reviewer LpSP**
> > >
> > > ## Miscellaneous
> > >
> > > > “The sigmoid function is defined as a positive, monotonically increasing, and bounded function.”
> > >
> > > We agree. We reiterate that previous works choose the sigmoid as an arbitrary function satisfying these generic properties. On the other hand, Lemma 3.1 shows something much stronger: the **exact formula** for the sigmoid appears by applying a particular discretization to the fundamental damped ODE $x’(t) = -x(t) + f(x(t))$.
> > >
> > > ----------
> > >
> > > ## Expressivity
> > > We respond to the claims that the LSSL is a special case of the LTC and has limited expressivity.
> > >
> > > > “If we relax the Taylor approximation conditions from a DCM, in fact, both equations become similar, obviously with the distinction of one (LSSL) being linear and one (DCM and LTCs) being nonlinear.”
> > >
> > > > “the formulation of LSSL is simply a linearized version of LTCs”
> > >
> > >
> > > We contrast the equation of the LTC with the LSSL:
> > >
> > > **[LTC]** [Hasani 2021, eq. (1)]
> > >
> > > $$ x’(t) = -\left[ \frac{1}{\tau} + f(x(t), u(t), t, \theta) \right] x(t) + f(x(t), u(t), t, \theta) A $$
> > >
> > > **[LSSL]** [eq. (1), (2)]
> > >
> > > $$ x’(t) = Ax(t) + Bu(t) $$ $$ y(t) = Cx(t) + Dx(t) $$
> > >
> > > The reviewer argues that with enough approximations, the former equation can be turned into the latter equation. Even if this is hypothetically possible, we believe that it is not a productive comparison as it can be argued about many models. The reviewer acknowledges that the DCM is formally defined as a different model, and has agreed with the statement that:
> > >
> > > > “the LSSL's … missing bilinear term as well as constant coefficients gives it fundamentally different properties.”
> > >
> > > To reiterate, the LSSL is a *linear, time-invariant* system compared to the DCM and LTC which are *non-linear, time-varying* systems. It is precisely these differences in the representation that are useful, for example leading to the LSSL’s [connection to convolutions](https://en.wikipedia.org/wiki/Linear_time-invariant_system) and in turn to the LSSL’s parallelizable computation. **These properties do not hold for other CT models**,
> > > and it is exactly these differences that lead to the LSSL’s improved performance over previous CT models.
> > >
> > > > “The more rigorous measure of the approximation capability of a deep network, in general, is the expressivity measures such as the number of linear regions [Montufar et al 2014] or the trajectory length measure [Raghu et al. 2017].”
> > >
> > > The reviewer mentions additional expressivity measures, beyond the ones shown in the paper, that would be useful. We agree that these are interesting directions for further research and are excited about the open problems continuing these lines of work. We highlight that our paper contains numerous theoretical results on expressivity, which are designed to help understand our model’s empirical strengths by connecting to well-established existing models (recurrence and convolutions).
> > >
> > > We would additionally like to point out that while understanding the expressivity of models is important, we believe that **the goal of designing models is not to *maximize* expressivity, but to express the relevant properties for their purpose**. As one example from this work, the reviewer agrees that reparameterizing state spaces using our generalized HiPPO theory (Theorem 1), while technically reducing expressivity, is actually a core factor for the LSSL’s performance (ablation [Line 312]). As another simple analogy, CNNs are a restricted form of MLPs, which is exactly why they excel at the tasks they were designed for. Similarly, LSSLs are a specific form of continuous-time model designed with specific goals in mind. Overall, we believe that we have introduced a simple and theoretically principled model that empirically excels across a wide range of sequence modeling tasks, particularly when long dependencies are involved.

---

> > > ### Author Response · Authors · 2021-08-30
> > > **Further Response to Reviewer LpSP**
> > >
> > > We thank the reviewer for responding to our comments. The reviewer’s new concerns fall into three main categories:
> > >
> > > - **[LTC]** The reviewer claims that the LSSL’s underlying ODE, the state space representation [Kalman 1960], is a special case of the LTC [Hasani 2021] and has limited conceptual novelty.
> > >
> > > - **[Engineering]** The reviewer re-frames the LSSL’s conceptual contributions, as well as improved empirical results over all baselines, as a collection of “engineering tricks”.
> > >
> > > - **[Experiments]** The reviewer argues that our empirical results are invalid on the basis that incorporating our own proposed theoretical contributions (e.g. generalized HiPPO theory and algorithms, Theorems 1 + 2) into our model is “unfair” to other methods.
> > >
> > > We respond to these claims below.
> > >
> > > --------------
> > >
> > > ## LTC
> > >
> > > The reviewer’s first main objection is that the LSSL is a special case of the LTC (and DCM). We reiterate that the LSSL is motivated from linear state spaces (Intro), a much older [e.g. Kalman, 1960] and more fundamental system than the DCM [Friston et al., 2003] or LTC [Hasani et al., 2021].
> > >
> > > > “LSSL will be best introduced as a linear version of the more general framework such as DCMs and liquid networks.”
> > >
> > > The LSSL is simply the deep learning version of the [linear state space representation](https://en.wikipedia.org/wiki/State-space_representation). **The reviewer argues that state spaces, which are a foundational topic in control theory and have been studied for decades, should instead be introduced as a special case of LTCs (Ramin Hasani et al., AAAI 2021).** We do not believe that this is the best way to motivate our work.
> > >
> > > We respond to more specific technical statements in the last section of this response, [Expressivity].
> > >
> > > --------------
> > >
> > > ## Engineering
> > >
> > > In response to our additional experiments on the baselines that the reviewer requested, the reviewer claims that our results are not fair and has labelled this work’s theoretical and empirical contributions as an “engineering framework”. We strongly disagree with this framing.
> > >
> > >
> > > > “I argue that with the exact same engineering steps we can also gain speed up for any CT model.”
> > >
> > > This claim is made without substantiation. Our speedup is directly tied to ideas such as parallelization from the convolutional view of LSSLs (Intro; Figure 1; Line 51; Line 295). The reviewer has acknowledged that these properties are a conceptual contribution of this work that does not hold for other CT models, contradicting this statement.
> > >
> > > We additionally would like to note that even if the reviewer’s argument is true, it would actually make our work’s contributions even stronger. If this work hypothetically only proposed new “engineering steps” that can be applied to all CT models, **improving their speed by 100x-1000x and increasing their accuracy on all tasks** as in our proposed model, this would constitute a major contribution toward the field.
> > >
> > >
> > > > “The gain in speed is a direct result of the downstream engineering tricks used, namely: HIPPO, optimization in C++, and linearization of operations.”
> > >
> > > As mentioned above, our work clearly states that our speed results are primarily a consequence of the convolutional view of the model, as well as new algorithms (Theorem 2). Additionally, the reviewer’s examples of “engineering tricks” are factually inaccurate, which we explain below.
> > >
> > > ### C++
> > > **Our model does not use special C++ code. Our entire method is implemented in PyTorch, which is available in the supplementary material.**
> > >
> > > The reviewer may be referring to [Line 1120], where we provide a CUDA binding to a [core linear algebra operation](https://docs.nvidia.com/cuda/cusparse/index.html#gtsv) that currently does not have a PyTorch interface but is available in other frameworks such as [Scipy](https://docs.scipy.org/doc/scipy/reference/generated/scipy.linalg.solve_banded.html)​​ and [Tensorflow](https://www.tensorflow.org/api_docs/python/tf/linalg/tridiagonal_solve). This is adding an interface to a core operation, not a “C++ optimization”.
> > >
> > > To make this argument completely clear, we note the following:
> > > - If we had implemented our method in Tensorflow instead of PyTorch, we would not have had to write a CUDA interface at all.
> > > - We note that the MLSSL, which does not train its state matrix $A$, only calls the above operation once and then caches it (Appendix A). Therefore without calling this CUDA interface at all, our MLSSL method -- which outperforms all baselines on all our tasks -- would simply take slightly longer to initialize, and the rest of the train and evaluation loops would remain the same speed as reported in the paper (Table 6).
> > >
> > >
> > > ### HiPPO
> > > **The HiPPO theory is a mathematical, not engineering, framework.**
> > > This positioning is clear in prior work [HiPPO, Gu et al. 2020] and in our submission.
> > >
> > > In more detail, our main proposed method, the SLLSSL, involves reparameterizing the state matrix $A$ of the LSSL with a novel representation based on generalizing the HiPPO theory. This is grounded in a mathematical framework, and the reviewer’s claim that this is an instance of “engineering” is misleading. To draw analogies:
> > >
> > > - CNNs, which are a reparameterization of MLPs using convolutions, are not considered an engineering trick. Rather, CNNs have improved inductive bias based on the theory of shift-equivariant kernels.
> > >
> > > - ExpRNNs, which are a reparameterization of RNNs using orthogonal matrices, are not considered an engineering trick. Rather, ExpRNNs mitigate vanishing gradients based on the theory of orthogonal matrices.
> > >
> > > Our generalization of HiPPO is analogous. The SLLSSL is a reparameterization of state spaces, which improves long-range dependencies based on the theory of continuous-time memorization.
> > >
> > >
> > > ### Linearity
> > > **Using a linear instead of quadratic (bilinear) model is a different mathematical representation, not engineering.**
> > >
> > > We elaborate on this point in the final section [Expressivity] of this response.
> > >
> > >
> > >
> > > -------------------------------
> > >
> > > ## Experiments
> > >
> > > > “However, the results on the other methods are not fair as these models are not incorporated on the same engineering framework (Hippo + computational tricks)”
> > >
> > > > “I believe that building networks based on the Hippo framework + the underlying engineering tricks are the core reasons why we see improved performance”
> > >
> > > The reviewer discounts our experiments based on two main arguments:
> > >
> > > - The reviewer claims that our method has access to unspecified “computational tricks” and “engineering tricks”. These claims have not been substantiated.
> > >
> > > - The reviewer agrees that our generalized HiPPO theory and algorithms, which was a major theoretical contribution of our work (Theorem 1 + Theorem 2 + Appendix C), is a core factor in our method’s strong empirical performance. However, **the reviewer claims that validating our proposed theory is “engineering” and “not fair” to other models.** We do not understand the logic of this objection.
> > >
> > > > “That is why I asked the authors to explicitly equip the parent family of LSSL (LTCs) with the HIPPO framework and then compare them against each other. Unfortunately, the authors did not include any evidence of such an experiment for this and argued that it is not clear how to do this.”
> > >
> > > While it is an interesting direction to explore how our proposed theory can be combined with other CT models, the reviewer has not given an indication of how this might be accomplished. We reiterate that HiPPO involves a very specific equation which is also exactly equation (1) of the LSSL, allowing them to be combined. **This equation does not appear in the LTC, so it is not at all clear how to combine the LTC with HiPPO**. Since no further details are provided, we are unable to address the reviewer’s insistence on this experiment.
> > >
> > > Finally, we reiterate that the generalized HiPPO parameterization is itself an important novelty of this work. We point out that *even if* our proposed component could be combined with other CT models to achieve better results, that in itself would underscore the contributions of our work.

---

> > > > ### Comment · Reviewer_LpSP · 2021-09-03
> > > > **Final Remarks**
> > > >
> > > > Dear Authors,
> > > >
> > > > I would like to thank you for commenting on the concerns raised.
> > > >
> > > > I took some time and worked with the codebase provided for LSSL vs the HiPPo code. As I mentioned before, in my opinion, the fundamental novelty and impact of this work are in the solid engineering framework the authors designed which I appreciate even more now that I got hands-on myself. I agree that even if we consider this point, the contributions of the paper can be beneficial for the RNN and the continuous-depth modeling community. In this light, I would like to vote for the acceptance of the paper and will increase my score by 2+ points.
> > > >
> > > > Please include a detailed discussion of what we discussed during the rebuttal phase, in the camera-ready version.
> > > >
> > > > The expressivity of these state-space models is an open topic that could be investigated in future work.
> > > >
> > > > Also, I believe that there is a strong connection between the state-space models, dynamic causal models, and liquid models that have to be investigated in the future. To this end, a large portion of the debate we had here can be discussed publically once the paper is out within the community.

---

### Official Review · Reviewer_Ueff · 2021-07-16

**Rating:** 8
**Confidence:** 3

**Summary:**

This paper suggests learning a proxy to continuous linear dynamical system. The paper discusses the tradeoffs and considerations and shows that the computations can be made efficiently. The paper demonstrates comprehensive empirical evaluation for the suggested method.


**Ethical Concerns:**

The paper does not entail any ethical concerns.

**Limitations And Societal Impact:**

The empirical evaluation is lacking more details in order for a fair comparison between the methods - namely a clear experimental protocol and reporting standard deviation.
There are no social concerns arising from this submission.

**Main Review:**

**Originality** - the paper suggests a novel reparameterization for RNNs. The work clearly extends previous work such as HiPPO [1]. Related work is discussed thoroughly, some times to briefly, specifically, relation to prior work in control theory requires a more comprehensive discussion.

**Quality** - The theoretical claims are well support. The empirical evaluation shows remarkable results compared to the baselines considered.

**Clarity** - Overall, the paper is written well. The ideas presented may very well be the basis for future work.
My problem here is that the bottom line is not clearly stated, rather it’s implicit and a reader must go back and forth to understand the big picture. I think an implementation subsection may help make things clearer.

**Significance** - The subject of improving recurrent networks for long time dependency is a very important topic. This paper suggests a new scheme, improving on existing ones significantly. In addition, the theoretical analysis may also lead to other parameterizations for continuous linear dynamical systems.

Additional comments:
1. Lines 58-59 - this connection is an important part of this work, it should be included at least in the appendix, also in order to implement an arbitrary convolution with a linear dynamical system, the dimensionality of the RNN must be larger than that of the 1d convolution.
2. Lines 113-116 - something seems wrong here (perhaps one of the $\Delta t$ is a typo), if you set $\alpha=0$ you get $x(t+\Delta t)=(I+\Delta t A)x(t)+\Delta tB u(t)\Delta t$ which can be written as $x(t+\Delta t)=x(t)+\Delta t(Ax(t)+ \Delta t Bu(t))$, but $x’(t)=Ax(t)+Bx(t)$.
3. Lines 119-120 - this is confusing, it is clear the paper continues with $\alpha=\frac{1}{2}$, A and B are not a function of $\Delta t$, why denote them as such?
4. Lines 186-188 - this statement is misleading, the overall functions that can be expressed with deep linear RNNs is still linear in the inputs, the statement implies otherwise.
5. Theorem 1 - the citation of [8] is in italic.
6. Corollary 4.1 - quasiseparable should be defined prior to the corollary.
7. Section 5.1 - the margins are so narrow here, reporting std is a must.
8. Line 310 - shouldn’t $\Delta t$ be a scalar? why is it O(H) parameters?
9. The learning setup is not clear from the main paper, what is the eventual optimization problem that is being minimized, how is $\Delta t$ parameterized in the learning setup. It is clear that the paper discusses special structures for A which are favorable in terms of computation but a reader is left a bit puzzled as to what it actually done in practice. Is it the GBT in equation (3) with \alpha=1/2?


[1] - Hippo: Recurrent memory with optimal polynomial projections, Albert Gu, Tri Dao, Stefano Ermon, Atri Rudra, and Christopher Ré.

**Time Spent Reviewing:**

7

---

> ### Author Response · Authors · 2021-08-07
> **Detailed Responses**
>
> 1. We will elaborate on the details in the Appendix. We also agree with the reviewer’s comment about dimensionality; see also the response to Reviewer ZFwG.
>
> 2. The second $\Delta t$ is indeed a typo. We caught this and noted it in line 574 of the Supplemental, and it has been fixed in the main body.
>
> 3. Indeed $A$ and $B$ are not functions of $\Delta t$, while the discretized matrix is a function of both $A$ and $\Delta t$. This discretized matrix is denoted $A^{(\Delta t)}$ to show that it is a function of both $A$ and $\Delta t$. More precisely, it is defined to be the matrix so that when plugging $\alpha = ½$ into equation (3) results in equation (4). We will add an explicit expression for $A^{(\Delta t)}$ in the complete model description.
>
> 4. We define deep linear RNNs to allow non-linearities in the depth direction, so that the overall output of the last layer can be non-linear in the input.
>
> 5. Thanks for catching this!
>
> 6. Space permitting, we will define quasiseparable before Corollary 4.1. It is currently in Definition 4 in Appendix C.3
>
> 7. See previous response about error bars
>
> 8. Since $\Delta t$ is a scalar, and we stack this $H$ times independently to form an LSSL (Figure 1, first line; Section 3.1, lines 142-147), there are $H$ parameters.
>
> 9. These details about $A$, $\Delta t$, and other model implementation details are in Appendix A. As discussed in the response to all reviewers, we will move essential model details to the main body.

---

> ### Author Response · Authors · 2021-08-07
> **Response to Reviewer Ueff**
>
> We are encouraged that the reviewer appreciated our novel ideas and strong empirical results, and thank the reviewer for their constructive feedback. We first respond to the reviewer’s main remarks on experimental protocol and error bars, and then respond in detail to the comments and questions.
>
> > Model Implementation
>
> We agree that important end-to-end model details should be moved from Appendix A to the main body for clarity, which is done in the revised paper. Conceptually, the end-to-end model is similar to a 1-D CNN with the convolutions replaced by LSSL layers, and the rest of the learning setup is exactly as in previous works on our datasets (all of which were used in prior work). See also response to Reviewer ZFwG for ablations on model architecture and tuning.
>
> > Experimental Protocol
>
> For space reasons, the detailed experimental protocol is in Appendix D. We hope that the above description of the end-to-end model also clarifies the learning setup and protocol. Code reproducing the experiments is provided in the supplemental and will be released.
>
> > Standard Deviation
>
> We note that “margins are narrow” only on the MNIST datasets which have been extensively evaluated in the literature. Here we provide the full mean/std results for SLLSSL corresponding to Tables 2, 3, and 5, which are all substantially better than the baselines, accounting for error bars:
> - sMNIST: 99.52 ± 0.01
> - pMNIST: 98.74 ± 0.04
> - sCIFAR: 84.52 ± 0.21
> - BIDMC RR: .356 ± .004
> - BIDMC HR: .489 ± .057
> - BIDMC SpO2: .147 ± .008
> - Speech Commands Raw: 95.54 ± 0.33
>
> The reason std is not reported in the original tables is for consistency and formatting considerations: **our results compare directly to best numbers reported in the previous papers, many of which did not include error bars**. As mentioned in Section 6, Line 329-336, the LSSL is extremely stable and consistent compared to baselines. We will add standard deviations to our results in the Appendix with a note in the main body.

---

> > ### Comment · Reviewer_Ueff · 2021-09-10
> > **Update following rebuttal**
> >
> > Thank you for the detailed response and the experiments conducted.
> >
> > I have read all the reviews and the authors' responses. I choose to keep my original score of strong accept.
> >
> > If accepted, I would suggest for the camera ready version to address the concerns raised by other reviewers, specifically,
> > consider adding ablation experiments to examine the effect of efficient implementations (HiPPO framework) on performance.

---

### Official Review · Reviewer_1sLL · 2021-07-16

**Rating:** 7
**Confidence:** 2

**Summary:**

This paper proposes a sequence-to-sequence model whose main building block is a linear layer called LSSL based on a standard dynamical system in control theory, including an internal ODE. LSSL is shown to implement both a convolutional and recurrent network, and, conversely, to be able, when stacked, to approximate any convolution and recurrent network with gating mechanisms, making the proposed model expressive. By endowing LSSL with the preexisting memory mechanism HiPPO, the resulting model MLSSL, and its counterpart SLLSSL that learns this operator, become computationally tractable and avoid gradient vanishing issues. The models are empirically validated on a number of regression and classification tasks showing their computational and performance advantages.

**Limitations And Societal Impact:**

Yes.

**Main Review:**

## Contribution

As described in the submission, the proposed model seems novel, computationally and numerically efficient, and well motivated by theoretical considerations. The extent of the contributions is significant and may be of broad interest for the NeurIPS community. The versatile abilities of MLSSL and SLLSSL and their thorough analysis further strengthen the contribution.

Furthermore, the claims of the paper are validated by a strong set of experiments with a provided code that seems complete and runnable, showing the clear advantage of MLSSL and SLLSSL to tackle long sequences by establishing state-of-the-art results on challenging tasks with reduced number of parameters and computational cost.

Unfortunately, this paper is too long and dense, with a long appendix, to adequately review given the short allocated reviewing time and high number of papers to review. I appreciate the efforts of the authors to provide a strong and well motivated paper, but I am not sure whether the NeurIPS reviewing system can assimilate such papers. Nonetheless, I would have some questions for the authors.

### Learning $\Delta t$ and Matching the Input and Output Times

The main issue of the paper as I understand it is how the learned timesteps are handled with respect to the input time series temporality. In Lemma 3.1, the learned $\Delta t$ most likely does not correspond to the actual temporal discrepancy between $x_t$ and $x_{t+1}$ (i.e., $\Delta t \neq 1$). This decorrelates the step size $\Delta t$ and the temporality of $x$ from the temporality of the input $u$, at least when following the dynamical system interpretation of Equation (3). Could the authors clarify this point, and in particular explain how this does not partly invalidate the dynamical viewpoint of LSSL?

### Sequence to Sequence Tasks

To the best of my understanding, all considered tasks in the experiments of the paper deal with classification and regression to scalar values. This is unfortunate, given that the proposed architecture is a sequence-to-sequence model. Could any sequence-to-sequence task may be considered?

### Generalization Vs. Approximation

The claim of lines 57-58 that LSSL generalizes convolutions and CNNs should be corrected, as it is shown in Section 3.2 that it can only approximate them, which is substantially different.


## Writing

This submission can be difficult to read for the uninitiated reader given the information density and high number of introduced notions. Nonetheless, the claims and results presented by the authors are clear and well explained throughout the paper, especially as the submission is fully self-contained thanks to a thorough appendix.



## Overall Sentiment

While the submission externally seems solid, I am unable to commit to a strong acceptation recommendation due to the aforementioned inherent time constraints of the NeurIPS reviewing system. Therefore, and partly because of the concerns expressed above, I am only suggesting a "Marginally above the acceptance threshold", but I am strongly willing to change my score following the authors' response and discussion phase with the other reviewers.


***

## Post-Rebuttal Update

I acknowledge the authors' response and thank them for their answers. The responses to all reviewers are thorough and seem to tackle the raised concerns, especially regarding readability and positioning. As far as I am concerned, my concerns are appropriately addressed. In particular, I believe that the solutions proposed by the authors will improve the quality of the submission.

Therefore, within the already exposed limits of my appreciation, I think that this paper can be accepted, thus raising my score to "Good paper, accept" but keeping my low confidence rating.

**Second update:** Following the authors' convincing responses and further investigation, I still think that this paper should be accepted but with a higher confidence. I consequently choose to raise my confidence score.

**Time Spent Reviewing:**

6

---

> ### Author Response · Authors · 2021-08-07
> **Response to Reviewer 1sLL**
>
> We sincerely appreciate the reviewer’s time and thoughtful review, which raised insightful points that have helped us improve the exposition in the revised paper. Despite the constraints imposed by the NeurIPS system, we believe that the reviewer’s summary, comments, and questions demonstrate understanding of the main points of this work. We respond to each detailed comment below.
>
>
>
> > Learning $\Delta t$
>
> An input sequence does not necessarily have an intrinsic timescale $\Delta t$, and it is instead up to the model to choose a resolution for the sequence, motivating the importance of learning the timescale. Conceptually, the model simply sees a discrete sequence of numbers and chooses to process it “as if” it were sampled at a rate of $\Delta t$.
>
> For a more intuitive interpretation, consider the special case when A corresponds to a uniform measure (in the literature known as the LMU [Voelker et al. 2019] or HiPPO-LegT [Gu et al. 2020] matrix). Then for a fixed $\Delta t$, equation (3) is simply memorizing the input within sliding windows of $\frac{1}{\Delta t}$ elements, and equation (4) extracts features from this window. Thus one interpretation is that the SLLSSL automatically learns convolution filters with a **learnable kernel width** $\frac{1}{\Delta t}$.
>
> We will include this simpler interpretation of the timescale $\Delta t$ in the revised paper.
>
> > Sequence to Sequence Tasks
>
> We agree that sequence to sequence tasks are a particularly suitable domain for the LSSL, where its dual interpretations as recurrent or convolutional can be most effective. Our updated experiments include examples of sequence to sequence tasks previously considered in the literature, where LSSLs outperform all baselines by 10-20% reduction in RMSE (see “Half-Cheetah” and “Walker2d” experiments in the response to Reviewer LpSP).
>
> We are excited about future applications of LSSLs in areas that require sequence to sequence models such as audio generation, where the theoretical and computational advantages of LSSLs may shine.
>
> > Generalization vs. Approximation
>
> We agree with the reviewer’s more precise phrasing of this claim. Colloquially, we meant that “LSSLs generalize convolutions” with a limiting argument, in the same sense as “neural networks are universal approximators”. More precisely, LSSLs do generalize all convolutions in the limit as the state size N goes to infinity; for any finite N they approximate convolutions. We will make this wording more precise.

---

### Official Review · Reviewer_ZFwG · 2021-07-19

**Rating:** 6
**Confidence:** 3

**Summary:**

Explored a linear state space dynamical system as a sequence transformation paradigm'
Explored and discussed the ability of this representation to explain some models like linear convolution and linear RNN
Presented mathematical analysis and empirical experiments for the proposed model

**Limitations And Societal Impact:**

I did not find any.

**Main Review:**

Clarity and Quality:
I think the idea is very interesting and authors have a done descent job in presenting details. However, my main concern with this paper is readability. The 9 pages submitted as main paper was not sufficient for me to understand the whole content and answer some of my basic questions like:
1. Dimension of y(t):
In the model definition y is one dimensional, which is not necessarily that appealing for many deep neural architectures where intermediate dimensions might have different dimensions. However, in appendix A.4, this is generalized to M > 1.
2. What is an example of coefficients A, B, C and D?
The difficulty of training A has been mentioned multiple times in the paper but A is introduced for first time in appendix A.2, line 521.
3. Same argument with \delta t. While it plays really important role in equations and discussions, there is not much details about it until reading appendices.
4. Reading the first 9 pages and experiments, I did not have any clue how the stack of LSSL look like.
Section A.4 was needed. And after that I am guessing the class of RNNs discussed here are the ones with non-linearity on depth and not on time. If thats right, I am not sure if the sentencing of the theorems necessarily reflect this point?
5. Authors seems have used many existing heuristic techniques like residual connections, pre-norm, post-norm, ... And I am wondering if using these will have any impact on the formulation? For example, can we use residual layer and still analyze our model with linear left-to-right system in Eq 1 and 2?
6. Experiment design is not reflecting the advantages of this model.
- I enjoyed reading the arguments about modeling long sequences. However, all experiments are some empirical performance comparison and since the architectural details are not clear, it is not clear to me the gains are because this model is more capable of modeling longer sequences or maybe benefited from some architectural detail choices? I was hoping to see some advantage of using LSSL, one can always train aLSTM network and tune all parameters to get state-of-the-art, what LSSL provides that make it better than the simple tuning ?
- For CelebA task, I dont understand the reasoning to choose ResNet-18 model which is 10X larger that MLSSL model  as baseline. Why not just find two architecture with and without LSSL assumption and show the effectiveness of LSSL?
7. Through the paper, authors use speech as an example case with long sequences. In practical speech recognition, model do not use raw audio as input, instead a window of signal is used. There are many successful methods using raw signal or FFT of window of signal as input (so line 276 is not completely correct), but the observation has been that these models at the end learn similar filters like Mel filters. Also they perform very similar to standard Mel features thus not beneficial for practical use cases.


**Time Spent Reviewing:**

12 hours

---

> ### Author Response · Authors · 2021-08-07
> **Additional Ablations**
>
> **The reviewer was concerned whether techniques such as residual connections and normalization layers are the main contributors to our strong empirical results.** We note that these techniques have been used for years (residuals - [He 2016]; LayerNorm - [Ba 2016]) and are incredibly standard; deep models often perform poorly without them, and most of the strongest baselines we compared against used these components [3, 4, 9, 29, 31].
>
> In addition to the ablations in the submission (see response 6a), we ran additional ablations on datasets from Tables 2 and 3 with residuals and normalization completely removed. As expected, the performance degrades, but the **pure LSSL still outperforms all baselines (which use these components)**.
>
> sequential CIFAR (best baseline: 74.4% accuracy)
> - SLLSSL (no residual): 83.4%
> - MLSSL (no residual): 82.8%
> - SLLSSL (no residual or normalization): 77.1%
> - MLSSL (no residual or normalization): 76.1%
>
> BIDMC SpO2 (best baseline: 0.87 RMSE)
> - SLLSSL (no residual): 0.24
> - MLSSL (no residual): 0.31
> - SLLSSL (no residual or normalization): 0.44
> - MLSSL (no residual or normalization): 0.53
>
> We additionally remark that these results were found using the **same hyperparameters as in the paper, without tuning** (Table 12). These ablations confirm that our empirical performance is driven by the core LSSL module as predicted by the theory.

---

> ### Author Response · Authors · 2021-08-07
> **Detailed Responses**
>
> We respond in detail to the reviewer’s questions and comments below. Citation numbers are from the submitted paper (supplementary version).
>
> (1) Indeed, $y(t)$ is 1-dimensional, which is the main limitation of prior work such as  LMU [41] and HiPPO [18]. LSSL overcomes this limitation by computing many states at once, allowing a larger intermediate dimension to be used as in standard deep models. This intermediate dimension was denoted by H, defined in Figure 1, Line 142-147, and elsewhere. This was usually set to 128 or 256 in our experiments (Table 12), comparable to standard Deep NNs.
>
> (2) A is defined to be a HiPPO matrix [Line 224, Section A.2]; B, C, and D can be viewed as standard learnable parameters [Line 144]. Since examples of A were already defined in prior work, for space reasons we chose to defer examples and details to Appendix A to be self-contained. We will add a note in the main body for clarity.
>
> (3) $\Delta t$ was specified to be a parameter [Line 144] which can be interpreted in relation to the length of the dependencies [Line 121-126]. See also the response to Reviewer 1sLL, which provides a more intuitive interpretation of \Delta t that will be added to the main body. With additional space, we will also move the ablations on $A$ and $\Delta t$ from Appendix D.1 into the main body.
>
> (4a) For space reasons, the architecture of LSSL was deliberately excluded from the main body because (i) it is orthogonal to the conceptual contributions of the paper, and (ii) the model is not sensitive to the choice of these architectural components. (See also response 5 and 6a)
>
> (4b) LSSLs are indeed a recurrent model that is linear in time and non-linear in depth; Lemmas 3.1 and 3.2 show that they are as expressive as standard non-linear RNNs. More precisely, LSSLs and popular RNNs are all approximators of a particular ODE (Appendix B.2).
>
> (5) As motivated in [Line 537, Line 552], these choices of residual and norm reflect the standard type of architecture used in modern deep neural networks. The overall model architecture is roughly equivalent to **taking a standard 1-D CNN and replacing the convolutional layers with LSSL layers** (equations 4 and 5).
>
> Unlike standard convolutions however, the LSSL carries a continuous-time interpretation and can also be computed recurrently (Figure 1 and Line 149). Thus adding components such as residuals do not change the “left-to-right” or “temporal” interpretation in equations (1)+(2) or (4)+(5).
>
> (6a) The architecture of the model was chosen to follow standard deep neural networks, and many of the baselines use similar architectures (e.g. [29], [9], [31]).
>
> > **I was hoping to see some advantage of using LSSL, one can always train a LSTM network and tune all parameters to get state-of-the-art, what LSSL provides that make it better than the simple tuning ?**
>
> We disagree that simple tuning can get SoTA on our datasets. For example, the LSTM has been extensively validated in previous work:
> (i) independent LSTM baselines have been reported on sequential MNIST and CIFAR in [18,39,17,31] (and many more), where our model strongly outperforms it (e.g. 20% gap on sCIFAR).
> (ii) It is well known that the LSTM struggles on sequences of just 200-500 time steps [2,3,5,18,25,31,41], while we showed that the LSSL works for up to 40000 time steps.
>
> As discussed in Section 6, our method involved less tuning than the baselines. Our experimental protocols have been detailed in Appendix D.2.
>
> > it is not clear to me the gains are because this model is more capable of modeling longer sequences
>
> * Outside of strong empirical results, the performance of LSSL is supported by ablations and theory. By ablating the choice of the A matrix (Section 5.4), we validate that the **performance of LSSL is not driven by the architecture, but by the theoretical properties of HiPPO** used in defining equation (1).
> * Other experiments validating the HiPPO theory have been confirmed in previous work, for example [17, Section 4.3] and [41, Section 3.1], showing that properly choosing A in equation (1) allows memorizing sequences of length 100000 or longer.
> * Finally, our new ablations below also confirm that the architecture (residuals and layernorm) are not the primary contributor to LSSL’s empirical performance.
>
> (6b) The motivation of this experiment is that LSSL strongly outperforms other sequence models on very long sequences (validated by Table 5 on Speech Commands), and thus we wanted to upper-bound its performance by a specialized domain-specific model. We have now additionally trained the strongest baseline, the CKConv [29], on CelebA, as an additional point of comparison. The result is that the CKConv is 7% behind ResNet, compared to only 2% for LSSL, further validating the effectiveness of LSSLs compared to previous methods designed for long sequences.
>
> (7) The reviewer asks about the motivation of our end-to-end speech experiments, since it is often better to train models using pre-defined features instead of the raw signal.
>
> * The Speech Commands dataset was primarily used as a benchmark for models on very long sequences, following prior work [29].
>
> * The reviewer mentions that raw signals in speech are not used in practice because standard hand-crafted features are sufficient. This is exactly what makes the results exciting - Table 5 demonstrates that LSSLs can potentially overcome this limitation and **learn better features than Mel filters**. We remark that using end-to-end models to outperform hand-crafted features is one of the main goals of deep learning itself.
>
> * There are related applications that necessitate models to handle raw signal, such as audio generation. In future work, we are excited about applications of LSSLs in such domains where their unique flexibility as a continuous-time, recurrent, and convolutional model may shine.

---

> ### Author Response · Authors · 2021-08-07
> **Response to Reviewer ZFwG**
>
> We are glad the reviewer found the ideas in the paper interesting and appreciate their constructive clarification questions, which have helped us to improve the draft. The reviewer’s main concerns were (1) clarity of model details in the main body, (2) questions about the model architecture and (3) validity of the SoTA results, in particular with respect to tuning. In brief:
>
> 1) In the expanded responses below, we have provided references to descriptions of model details from the original submission. We thank the reviewer for pointing out details that may be confusing, and we will highlight them better in the main body of the revised paper. We also mention that our full submitted code will be publicly released.
>
> 2) The reviewer is **concerned that components such as residuals and layernorm affect the formulation of our proposed LSSL module**. We note that our deep architecture is very standard -- appearing in many of our baselines -- and does not impact the conceptual contributions of this paper. The submission includes theory and ablations that validate that our empirical results come from LSSL and not the deep architecture (see detailed response 6a). At the reviewer’s suggestion, we have additionally run more ablations to substantiate this. In summary, a **pure LSSL network with no residuals or normalization layers still outperforms all baselines on our datasets** (see “Additional Ablations” response).
>
> 3) We disagree with the reviewer’s claim that **“simple tuning of baselines such as LSTMs would make them achieve SoTA”**. We emphasize that all experiments in the paper compared directly against previously reported results in their exact experimental settings [Line 257], where these baselines have been tuned extensively by previous works. Our method actually requires less tuning than baselines [Line 329-336], which is further substantiated by our additional experiments where LSSL outperforms several more methods on 3 new tasks with no tuning (see response to Reviewer LpSP). On the LSTM claim in particular, see response 6a.

---

### Author Response · Authors · 2021-08-07
**Revision Summary - Thanks to all reviewers!**

We thank all reviewers for their time and thoughtful feedback. Overall, reviewers spoke positively of our model’s theoretical contributions touching on continuous-time, recurrent, convolutional, and state-space models, and our strong empirical results on very long time series. Reviewers noted that the extent of the technical contributions is significant (1sLL), may well be the basis for future work (Ueff), and that the theoretical results particularly around convolutions are insightful (LpSP). Reviewers were also impressed by the properly designed experimental framework (LpSP) which showed comprehensive and “remarkable” results (1sLL, Ueff), where the proposed method substantially outperforms baselines on long sequences (e.g., by 20% accuracy on speech data of length 16000).

The reviewers’ main concerns were on clarity of model details and implementation (ZFwG, 1sLL, Ueff), and discussion of related work (Ueff, LpSP). Their feedback has helped improve our updated manuscript, which includes:

* Moving essential model implementation details from Appendix A and C to the main body (space permitted). We also note that full code reproducing the experiments was provided in the supplementary material, and will be released publicly.

* Adding more related work discussion, particularly on previous continuous-time (CT) and RNN models (LpSP), as well as relevant control theory (Ueff). Our response to Reviewer LpSP includes a substantial discussion of the LSSL compared to previous CT models.

* Additional experiments, in particular (i) ablating the deep architecture to validate that the core LSSL module is the main contributor to the state-of-the-art results (Reviewer ZFwG), and (ii) 4 additional baseline methods and 3 diverse tasks (shorter length, seq-to-seq, and irregularly sampled tasks) showing that LSSL outperforms previous CT models in their best performing regimes (Reviewer 1sLL, LpSP).

We have responded individually to each reviewer in the threads. Our responses are organized so that the first post has the main takeaways, with detailed replies following.

---

### Decision · Program_Chairs · 2021-09-27

**Decision:**

Accept (Poster)

**Comment:**

The authors propose a continuous time model combining recurrent and convolutional structures. Overall, reviewers are supportive of the paper. The main remaining concerns, after discussion, are mostly with respect to the presentation. It was felt that the paper is dense, heavily relying on the appendix, and could be more clearly communicated in the main text. There was also a perception that the main contributions were in engineering. While the authors push back this notion, there is no need for this to be perceived as a drawback, and could be reasonable to highlight engineering contributions in revisions. There were also some concerns that it was hard to derive clear takeaways from some of the experiments (details in the reviews). Please try to be receptive of reviewer comments in preparing revisions. But generally, this is great work!